# PROCESS-DRIVEN AUTOFORMALIZATION IN LEAN 4

## ABSTRACT

Autoformalization, the conversion of natural language mathematics into formal languages, offers significant potential for advancing mathematical reasoning. However, existing efforts are limited to formal languages with substantial online corpora and struggle to keep pace with rapidly evolving languages like Lean 4. To bridge this gap, we propose a large-scale dataset **Form**alization for **L**ean **4** (**FORML4**) designed to comprehensively evaluate the autoformalization capabilities of large language models (LLMs), encompassing both statements and proofs in natural and formal languages. Additionally, we introduce the **P**rocess-**D**riven **A**utoformalization (**PDA**) framework that leverages the precise feedback from Lean 4 compilers to enhance autoformalization. Extensive experiments demonstrate that PDA improves autoformalization, enabling higher compiler accuracy and human-evaluation scores using less filtered training data. Moreover, when fine-tuned with data containing detailed process information, PDA exhibits enhanced data utilization, resulting in more substantial improvements in autoformalization for Lean 4.

## 1 INTRODUCTION

Autoformalization is the automatic conversion of natural language mathematics into formal languages (Wang et al., 2018; Szegedy, 2020). It reduces the high cost of formalization and bridges the gap between automated mathematical reasoning research and the vast body of natural language mathematical knowledge (Wu et al., 2022; Jiang et al., 2023c).

Recent advancements in large language models (LLMs) showed promising capabilities for various tasks (Achiam et al., 2023; Anthropic, 2024; Meta, 2024), opening up possibilities for LLM-based autoformalization. While researchers have explored using few-shot prompting (Wu et al., 2022; Gadgil et al., 2022) or training LLMs on large-scale datasets containing both informal and formal data (Azerbayev et al., 2023a;b; Jiang et al., 2023a; Ying et al., 2024c;a), existing efforts are limited to formal languages with a substantial online corpus, e.g., Lean 3 (de Moura et al., 2015).

Recently, due to the improved performance and advanced compilation features, the community has pivoted towards Lean 4 (de Moura & Ullrich, 2021), a next-generation theorem prover and programming language. This transition has created a pressing need for comprehensive datasets and models tailored specifically to Lean 4 (Ullrich & de Moura, 2022b;a; Nawrocki et al., 2023). Meanwhile, the rapid evolution of Lean 4 poses significant challenges for autoformalization efforts due to its complex syntax and extensive lemma corpora. This underscores the need for methods that focus on the semantic aspects of mathematical theorems, an area previously underexplored due to difficulties in automated assessment (Lu et al., 2024b). Addressing these semantic elements could enhance autoformalization techniques to better adapt to Lean 4's ongoing development.

To address key gaps in autoformalization for Lean 4, we introduce **Form**alization for **L**ean **4** (**FORML4**), an extensive dataset for training and evaluating LLMs' autoformalization capabilities. FORML4 is derived from Mathlib 4 theorems, automatically informalized, and then rigorously quality-checked manually. In addition, we propose a **P**rocess-**D**riven **A**utoformalization (**PDA**) framework for iterative performance improvement and automated assessment. As illustrated in Figure 1, PDA begins with training an autoformalization model on FORML4. The model's output is then processed by the Lean 4 Compiler, generating automated feedback. This feedback generates process-level annotations for the autoformalization output, utilized to train a process-supervised verifier (PSV). The autoformalization model is then fine-tuned based on the verifier's feedback. This iterative cycle enables mutual improvement between autoformalization and verifier models.

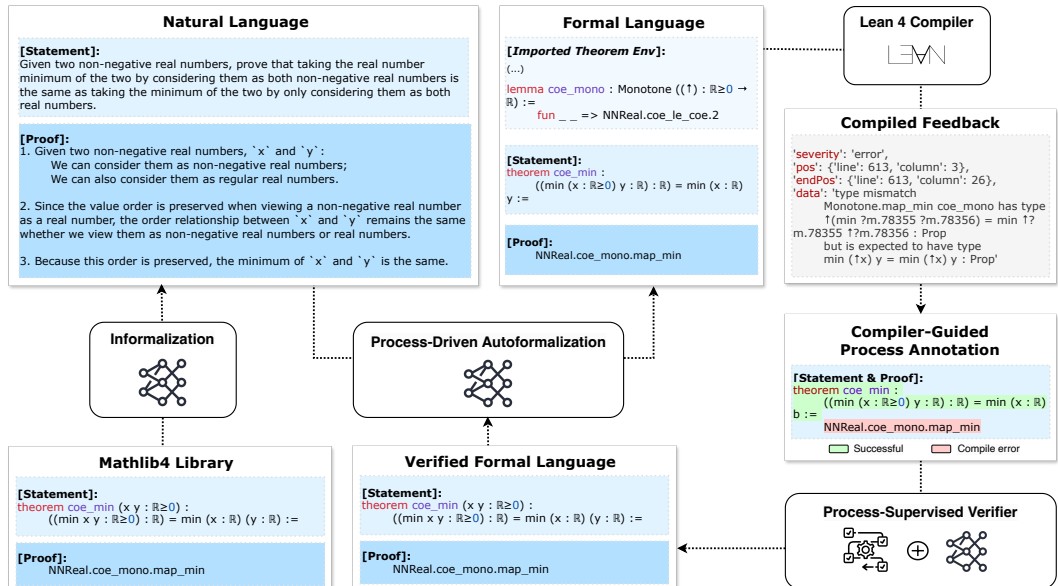

Figure 1: An overview of PDA trained on FORML4. Note that the goal of PDA is statement autoformalization, and does not include the translation of proof per se (Jiang et al., 2023a). The reason for including proof steps throughout our framework is to *enable the compiler to better assess the semantic and logical aspects of autoformalized statements by compiling statements and proof steps together*. As illustrated, while the statement passes the compiler as grammatically correct, an error is detected in the proof step, indicating an incorrect autoformalization. This process-level feedback helps PDA refine the autoformalized statement effectively.

The unique strength of FORML4 lies in its inclusion of *both statements and their corresponding proofs in natural and formal languages*. This approach enables a comprehensive evaluation of model autoformalization outputs, contrasting with existing datasets (Jiang et al., 2023a; Ying et al., 2024a), which focus solely on statements. There are three key reasons for appending proofs to theorem statements in FORML4, each contributing to improved **data quality**, **evaluation granularity**, and **process-driven enhancement**. First, including proofs provides valuable context that aids in the generation of higher-quality statements during the dataset construction phase of FORML4. This context also serves as a prompt, potentially enhancing the performance of autoformalization models.

More importantly, including proof in FORML4 empowers the PDA framework to use the compiler feedback of combined statement and proof steps as the proxy for evaluating the quality of statement autoformalization. Formal languages offer syntactic rigidity that allows for automatic assessment by compilers, eliminating ambiguity in formal language generation (Yang et al., 2023a). By utilizing both statements and proofs, FORML4 facilitates comprehensive feedback from the Lean 4 compiler[1], enabling strict assessments of syntax and semantic integrity in reasoning logic. Lastly, we can leverage the precise feedback provided by Lean 4 compilers to improve autoformalization. Building on FORML4, our PDA is distinct from existing informal mathematical reasoning methods that rely heavily on human or machine annotation (Lightman et al., 2024; Wang et al., 2023a).

Extensive experiments demonstrate that PDA significantly enhances autoformalization in Lean 4, achieving better results with less training data. When fine-tuned with higher-quality data, PDA utilizes this information effectively, leading to further improvements. Our key contributions are:

- We construct an extensive pioneer dataset FORML4 for evaluating autoformalization in Lean 4, encompassing the complete process from natural language questions to formal proofs.

- We propose a process-driven framework PDA that leverages formal languages to provide process feedback on reasoning, enhancing the autoformalization capabilities of LLMs.

- We conduct a comprehensive study featuring robust quantitative and qualitative analysis, along with human evaluation. We fully open-source FORML4 and PDA to facilitate research.

---

[1]Details of the Lean 4 compiler are provided in Appendix I.3.

## 2 RELATED WORK

**Autoformalization with LLMs** Autoformalization is the task of automatically converting informal theorems and proofs into machine-verifiable formats (Wang et al., 2018; Szegedy, 2020). Early approaches employed neural machine translation methods to translate texts into the Mizar language (Wang et al., 2020). Recent advancements in LLMs have opened up new possibilities for autoformalization. Researchers have explored using few-shot prompting to enable LLMs to translate mathematical problems into formal formats, including Isabelle and Lean (Wu et al., 2022; Gadgil et al., 2022). Other studies have adopted a more structured approach to this task. Notably, the DSP system (Jiang et al., 2023c) utilizes LLMs to draft informal proofs and map them into formal sketches, with automated theorem-proving systems employed to fill in the missing details in the proof sketch. Additionally, a line of research has focused on training LLMs on large-scale datasets containing both informal and formal mathematical data to evaluate their performance in autoformalization (Azerbayev et al., 2023a;b; Jiang et al., 2023a; Ying et al., 2024c). Unlike existing efforts that often neglect the detailed compilation information available in ITPs, our proposed method utilizes process feedback from the Lean 4 compiler to further improve the autoformalization abilities of LLMs.

**Process and Outcome Supervision** Recent efforts explore enhancing the reasoning capabilities of LLMs by using verifiers to select the best answer from multiple candidates. There are two main types of verifiers: the Outcome-Supervised Verifier (OSV) and the Process-Supervised Verifier (PSV). OSV is supervised with a signal based on the final answer (Cobbe et al., 2021; Yu et al., 2023a), while PSV is with detailed feedback which requires evaluating individual reasoning steps (Uesato et al., 2022; Li et al., 2023; Lightman et al., 2024; Ma et al., 2023). Despite the time-consuming annotation cost, PSV offers several advantages that make it preferable to OSV. PSV can provide fine-grained feedback by pinpointing the location of errors, which is valuable for reinforcement learning and automatic correction (Lightman et al., 2024; Wu et al., 2023). To alleviate the extensive human annotation, recent efforts (Wang et al., 2023a; 2024) propose a machine annotation framework using Monte Carlo Tree Search (Coulom, 2006; Silver et al., 2016). This annotation process demands a lot of computing resources, potentially imposing a limitation on the usage. PDA leverages formal languages that can naturally provide precise feedback on the reasoning process, enabling automatic process annotation without substantial human or machine annotation costs.

## 3 FORML4: DATASET CONSTRUCTION

The rapid development of Lean 4 (de Moura & Ullrich, 2021) necessitates a benchmark to assess LLMs' autoformalization capabilities. Existing datasets (Jiang et al., 2023a; Ying et al., 2024a) aims to create benchmarks by informalizing formal theorems from existing libraries. However, they rely on zero-shot instructions to collect natural language statements from GPT-4 without quality checks or rigorous post-processing. Additionally, it focuses solely on translating theorems, overlooking the benefits of using proofs as context, which could enhance both the dataset quality and the evaluation of autoformalization performance.

Instead, we implement a deliberate informalization framework to curate a high-quality autoformalization dataset FORML4 for training and evaluation. FORML4 incorporates proof steps alongside statement translation, leveraging formal proof generation as an auxiliary task. Proof steps could enhance formalized statement quality by providing additional context for model reasoning (Huang et al., 2024b). FORML4 encompasses formal-informal pairs of proof steps along with statements, enabling a comprehensive assessment of an LLM's autoformalization capabilities. Additionally, FORML4 is constructed using a fine-grained pipeline and rigorous quality checks to ensure high translation quality. In this section, we will introduce the data source of FORML4 (Section 3.1), the informalization approach (Section 3.2), the curation process (Section 3.3), and comparisons with existing datasets (Section 3.4).

### 3.1 DATA SOURCE

**Statement and Proof Extraction** We start by extracting formal statements and proofs from Lean 4 theorems in Mathlib 4[2], one of the most extensive formal mathematics libraries available. This

---
[2]https://github.com/leanprover-community/mathlib4

Table 1: Statistics of FORML4. The test sets do not necessarily require Lean 4 ground truth statements and proofs, since the autoformalized output can be verified by the compiler. The real test set only contains natural language queries and answers, without any corresponding Lean 4 statements.

| Dataset | Size | Lean 4 | | | | Natural Language | | | |
|---|---|---|---|---|---|---|---|---|---|
| | | # Chars, State. & Proof | | | | # Chars, Q & A | | | |
| | | Mean | Median | Min | Max | Mean | Median | Min | Max |
| Training | 14,250 | 147 | 116 | 39 | 5507 | 192 | 166 | 30 | 1485 |
| Random Test | 950 | 152 | 116 | 43 | 3170 | 188 | 166 | 35 | 836 |
| Basic Test | 970 | 133 | 96 | 41 | 2716 | 146 | 135 | 33 | 529 |
| Real Test | 967 | - | - | - | - | 1269 | 1151 | 134 | 4909 |

process is adapted from the implementation of LeanDojo[3] (Yang et al., 2023a) to search for and extract theorems from Mathlib 4. However, unlike LeanDojo focuses on extracting theorem names and tactics[4] for theorem proving, we extract the complete content of both the statement and the proof, aiming to provide comprehensive content for improved autoformalization.

**Datasets Split** We randomly sample theorems (including their statements and proofs) from the extracted pool of Mathlib 4, and split them to create a **training set** and a **random test set** for training and evaluating LLMs. An example is provided in Appendix Table 8. In addition, for a more domain-general comprehensive evaluation of a model's autoformalization performance, we further include a **basic test set** and a **real test set** whose domains differ from the training set. The basic test set is extracted from Mathlib 4, but it exclusively focuses on the proof for fundamental concepts in a mathematical topic[5]. It assesses the model's ability to autoformalize basic theorems with minimal reliance on prior knowledge or established lemmas. The real test set is constructed by collecting natural language math questions and answers from NuminaMath, a high-quality collection of natural language mathematics problems ranging from high school exercises to international mathematics olympiad problems (LI et al., 2024). We transform each question into a natural language statement by appending the ground-truth answer and a request to prove the answer is true. By not relying solely on formal mathematical theorems and proof from Mathlib 4, we extend our evaluation domains to real-world settings. More details of test sets can be found in Appendix O.2.

## 3.2 INFORMALIZATION

To obtain natural language data for the extracted formal theorems, we employ a two-step process: 1) We utilize a LLM to translate formal mathematical statements into natural language (i.e., formalization) 2) Next, we generate new informalized versions by first explaining the formalized proof and then providing a step-by-step proof in natural language. This process avoids verbatim mentions of Lean 4 functions. Our construction pipeline was further augmented with the following techniques, to elicit high-quality informalization output from LLMs.

**Statement and Proof Conversion:** We instruct the model to convert all components of the formal content – both statements and proofs – into natural language. While this is computationally heavier and more challenging as it requires the model to understand the syntax of Lean 4 and the logical reasoning steps within each proof, the inclusion of proof steps has several benefits in both dataset construction and evaluation: (1) during informalization, the provided proof steps could potentially add informative context to the preceded formal theorem statement in the prompt, hence improving informalization quality (Liu et al., 2023a); (2) in autoformalization, the existence of proof steps also enables us to examine autoformalization performance by assessing the validity of the formalized combination of theorem statements and proof using a compiler, increasing the difficulty and granularity of autoformalization evaluation. This is supported both in our human evaluation results (Table 6) and in previous research (Huang et al., 2024b).

---

[3]https://github.com/lean-dojo/LeanDojo/blob/main/scripts/generate-benchmark-lean4.ipynb

[4]Tactics are commands or instructions that describe how to construct such a proof.

[5]For example, such theorems typically appear in files like `mathlib4/Mathlib/Geometry/Euclidean/Basic.lean`, which establish core geometrical concepts and prove simple results about real inner product spaces and Euclidean affine spaces.

It is important to note that the role of included proof steps is to serve as an auxiliary tool to aid (1) dataset quality and (2) evaluation in **statement autoformalization** which is the central goal of the current work, rather than statement-and-proof autoformalization. Therefore, the quality of FORML4 and our evaluation framework is independent of whether the natural-language proof is perfectly aligned with the formal proof.

**Decomposition Strategy:** To address the complexity of informalizing both statements and proofs, we implement a decompositional prompting strategy inspired by task decomposition approaches in scalable oversight research (Christiano et al., 2018; Wu et al., 2021). Our strategy breaks down the informalization process into sequential subtasks: translating the formal statement, explaining each proof step, and then constructing a natural language proof. This approach effectively differentiates between explaining Lean 4 terms and creating an independent natural language proof, crucial for meaningful autoformalization evaluation. The strategy is augmented with few-shot examples to align the model output with our expectations. Please check Appendix C for the detailed rationale and Appendix D for the complete prompt template.

### 3.3 CURATION PROCESS

**Preprocessing:** Before informalization, we conducted several preprocessing steps on the extracted theorems to enhance the quality of our formalization output. These steps include retaining specific commands, filtering certain samples, and removing unsuitable entries. More details on our preprocessing approach can be found in Appendix O.1.

**Model Selection:** To ensure high-quality LLM-based informalization, we evaluated two state-of-the-art LLMs in formal mathematical reasoning: GPT-4 and Gemini-Pro-1.5. Based on a comparative study involving human annotators, Gemini-Pro-1.5 consistently outperformed GPT-4, achieving higher scores in informalization success (80% vs. 70%) and being preferred in 80% of samples. Given its superior performance, we employed Gemini-Pro-1.5 for the informalization process in constructing FORML4. For detailed evaluation methodology and results, see Appendix K and Appendix E.

**Post-processing:** Based on the obtained informalized data, we conduct a filtering process to further guarantee PDA to have high-quality training and testing data for auto-formalization. More details are listed in Appendix O.3. In FORML4, we further provide a "Theorem Environment" that includes each theorem's full dependencies and premises, facilitating easier compilation. Specifically, one only needs to concatenate the "Theorem Environment" with the autoformalized result to verify the latter, eliminating the need to delve into the details of Mathlib. This approach simplifies the compilation process in autoformalization evaluation later.

**Human Verification:** We first manually verify the informalized dataset where four Lean 4 experts evaluated 60 samples: 20 from the basic test set and 40 from the random test/train set. The average success rate was 72%, indicating relatively high-quality informalization performance. Please check Appendix F for detailed verification results and discussions.

To further validate the dataset quality of FormL4, we additionally verified three comparable datasets that are constructed using LLM-based methods and in similar magnitude of sizes: FORML4, MMA (Jiang et al., 2023a), and Lean Workbook (Ying et al., 2024a), extracting 30 samples from each dataset. FORML4 achieves the highest verification accuracy of 73.33%, consistent with the previous verification result of 72%. This validates FORML4's quality and effectiveness of our carefully implemented informalization pipeline. Full comparison details are in Appendix Q.

Notably, the split stats between the basic test set (0.875) and the random test set (0.575) show a significant discrepancy in the human-verified informalization success rate (p = 0.0099), suggesting that informalization difficulty increases with formal theorem complexity.

**Dataset Statistics:** Table 1 displays the final data statistics of FORML4, including the size of each subset and the length of statement and proof in characters for both Lean 4 and natural language.

### 3.4 COMPARING FORML4 WITH EXISTING AUTOFORMALIZATION DATASETS

Table 2 compares FORML4 with existing autoformalization datasets, highlighting its unique features. As the largest dataset designed for iterative, process-driven autoformalization training, FORML4 includes both statements and proofs, employing an LLM-based informalization method that departs

Table 2: Comparison of FORML4 with existing autoformalization datasets.

| Characteristic | FORML4 | MMA (Jiang et al., 2023a) | Lean Workbook (Ying et al., 2024a) | ProofNet (Azerbayev et al., 2023a) | Minif2f (Zheng et al., 2022a) | FIMO (Liu et al., 2023b) |
|---|---|---|---|---|---|---|
| Source Language | Formal | Formal | Natural | Natural | Natural | Natural |
| Size | 17k | 332k | 57k | 371 | 488 | 149 |
| Includes Proofs | ✓ | ✗ | ✗ | ✓ | ✗ | ✗ |
| Uses Lean 4 | ✓ | ✓ | ✓ | ✗ | ✗ | ✗ |
| **Construction Method** | | | | | | |
| Direction | Informalization | Informalization | Formalization | Formalization | Formalization | Formalization |
| LLM-based | ✓ | ✓ | ✓ | ✗ | ✗ | ✓ |
| Human-Verified | ✓ | ✗ | ✓ | ✓ | ✓ | ✓ |
| **Primary Usage** | | | | | | |
| Training | ✓ | ✓ | ✓ | ✗ | ✗ | ✗ |
| Benchmarking | ✓ | ✓ | ✓ | ✓ | ✓ | ✓ |
| Process-Driven Feedback | ✓ | ✗ | ✗ | ✓ | ✗ | ✗ |

from traditional formalization approaches. It ensures high-quality data through rigorous inspection and human verification, while enabling fully automated training using Lean 4 compiler feedback, unlike datasets requiring human intervention. Moreover, FORML4 covers a broader spectrum of mathematical complexities, making it suitable for advanced autoformalization tasks. For a detailed analysis of each characteristic in the comparison, please refer to Appendix L.

# 4 METHOD: PROCESS-DRIVEN AUTOFORMALIZATION

This section presents our approach to enhancing the autoformalization capabilities of LLMs using process feedback. We establish a baseline by fine-tuning an LLM on the FORML4 training set. Then we further introduce a Process-Supervised Verifier (PSV) that incorporates Lean 4 compiler feedback during training (Section 4.1). Finally, we propose a continuous improvement methodology that iteratively refines both autoformalization and verification models, guided by the objective evaluation of the Lean 4 compiler (Section 4.2).

## 4.1 VERIFICATION MODEL

We propose to train the verifier by leveraging the granular, process-level feedback provided by the Lean 4 compiler. This method diverges from previous approaches (Wu et al., 2022) that rely solely on binary compilation outcomes. Instead, we employ a more nuanced strategy that assigns labels to each step in the training data based on the "first error location" principle introduced by Uesato et al. (2022). Our labeling strategy is as follows: steps preceding the first compiler-detected error are labeled as "correct", while subsequent steps are labeled as "incorrect". This approach allows us to incorporate rich, step-wise information throughout the compilation process, in contrast to traditional result-centered methods that use rejected sampling or apply binary outcomes to train reward or verifier models. The parameters and variables used in our verifier models are summarized in Table 3. To evaluate the efficacy of our process-supervised training, we compare two models:

Table 3: Parameters and variables used in verifier models.

| Symbol | Description |
|---|---|
| $q$ | Question |
| $S = \{S_1, \ldots, S_n\}$ | Set of samples |
| $S_i^{(1:t)}$ | Subsequence of steps up to the $t^{th}$ step of sample $S_i$ |
| $Y = \{Y_1, \ldots, Y_n\}$ | Label set for the samples |
| $y_i \in \{0, 1\}$ | Outcome-level label across all steps based on final compilation outcome |
| $y_i^t \in \{0, 1\}$ | Step-level label for the $t^{th}$ solution step within the $i^{th}$ sample $S_i$. |
| $n$ | Total number of samples |
| $m_i$ | Number of steps in $S_i$ |
| $r_i^t = f_\theta(q; S_i^{(1:t)})$ | Predicted probability of correct class at step $t$ |
| $\theta$ | Model parameters |

**1. Outcome-Supervised Verifier (OSV):** This model is trained using step-level loss with a uniform label based on the final compilation outcome. Following Lightman et al. (2024) and Wang et al.

(2023a), we train the OSV model using cross-entropy loss:

$$\mathcal{L}_{\text{OSV}}(q, S, Y, \theta) = -\frac{1}{n} \sum_{i=1}^{n} \frac{1}{m_i} \sum_{t=1}^{m_i} \left[ y_i \log(r_i^t) + (1 - y_i) \log(1 - r_i^t) \right],$$

**2. Process-Supervised Verifier (PSV):** This model is trained using the "first error location" labeling strategy with step-level loss. The loss function is structurally similar to that of the OSV model, but it uses step-wise labels $y_i^t$ based on the "first error location" strategy:

$$\mathcal{L}_{\text{PSV}}(q, S, Y, \theta) = -\frac{1}{n} \sum_{i=1}^{n} \frac{1}{m_i} \sum_{t=1}^{m_i} \left[ y_i^t \log(r_i^t) + (1 - y_i^t) \log(1 - r_i^t) \right],$$

To ensure a fair comparison between PSV and OSV, both models are trained within a standard language modeling framework. We introduce two special tokens to represent the "correct" and "incorrect" labels during training. By leveraging the process feedback from the Lean 4 compiler, we hypothesize that our method is more suitable and efficient for the task of autoformalization, as it captures the nuanced progression of the proof construction process rather than relying solely on the outcome. The comparative performance analysis of these models is presented in Table 5.

### 4.2 FURTHER ENHANCEMENT WITH BACK-PROPAGATED PROCESS FEEDBACK

An iterative refinement strategy is designed to leverage feedback from the Lean 4 compiler to continuously improve both the autoformalizer and verifier. This process comprises three key steps:

**Step 1: Autoformalizer Improvement** The verifier evaluates the autoformalizer's outputs, assigning labels based on their estimated likelihood of successful compilation. This filtering process ensures that subsequent training phases focus on the most promising solutions. The autoformalizer is then fine-tuned using the verifier's labels, effectively leveraging the outputs that PSV evaluates correctly. This approach enhances the autoformalizer's learning efficiency and output quality.

**Step 2: Lean 4 Process Feedback Integration** The enhanced autoformalizer, when applied to the training dataset, demonstrates an improved rate of successful compilations. These outputs are then processed by the Lean 4 compiler, which provides detailed process feedback through syntax checking and reasoning verification.

**Step 3: Verifier Enhancement** We further fine-tune the verifier using the high-quality data (with an increased proportion of positive examples) generated by the enhanced autoformalizer. This fine-tuning incorporates process-level supervision derived from the Lean 4 compiler's feedback, allowing the verifier to learn from a more nuanced and accurate representation of the compilation process.

The cyclical nature of this process, with feedback from the Lean 4 compiler at its core, offers significant advantages. It provides an objective measure of progress, mitigating the potential for bias arising from isolated interactions between the autoformalizer and verifier.

## 5 EXPERIMENTS

To systematically validate the enhancement of autoformalization performance, we use a multi-faceted evaluation approach: Firstly, Lean 4 compiler feedback of combined statement and proof is introduced as the proxy for **automatically evaluating statement autoformalization** (Section 5.2). The stricter requirements for successfully compiling both the statements and proof can potentially encompass both the semantic and logical validation in the autoformalized statements, represents a significant departure from prior statement-only compiling approaches which only assess syntactic validity. Secondly, we conducted extensive **human evaluation** (Section 5.4) to authentically assess autoformalization performances, comparing enhanced models with baselines. Our human evaluation showed a strong correlation with compiler results, validating our automated evaluation approach.

### 5.1 LLMS AS AUTOFORMALIZERS

We assess the autoformalization capabilities of both open-sourced and proprietary LLMs on FORML4 test sets. The results in Table 4, underscore the challenges that current LLMs, including GPT-4, face

in Lean 4 autoformalization tasks. The low-performance results obtained from greedy decoding underscore the need for method improvements in this domain. Additional details on pass@k, the querying prompt, and performance analysis are provided in Appendix P.

## 5.2 AUTOFORMALIZATION ENHANCEMENT

This section presents our process-driven autoformalization framework and its experimental results. We begin by describing our experimental setup, followed by the performance of our enhanced autoformalizer, and conclude with the results of our further enhanced verifier model.

### 5.2.1 EXPERIMENTAL SETUP

We establish three key components for our own experiments:

1) **Finetuned Baseline Autoformalizer (BA):** We train Mistral-v0.3-7B (Jiang et al., 2023b) on FORML4 as a baseline. To improve its performance in real-world scenarios, we further fine-tune it on successfully compiled outputs from GSM8K (Cobbe et al., 2021) and MATH (Hendrycks et al., 2021).

2) **Verifier Models:** We develop two types of verifiers: Process-Supervised Verifier (PSV) i.e., fine-tuned using step-level feedback from the Lean 4 compiler, and Outcome-Supervised Verifier (OSV) i.e., fine-tuned based on single final compilation signal.

3) **Evaluation Metrics:** i) Multiple Choice (**MP1**): Ability to select a successfully compiling candidate from multiple candidates. ii) Precision (**Prec.**): Fraction of selected samples that compile successfully. iii) **Recall**: Fraction of successfully compiled samples selected by the verifier.

### 5.2.2 ENHANCED AUTOFORMALIZER PERFORMANCE

We compare four autoformalizer models: 1) Baseline Autoformalizer (**Baseline**) 2) Rejective Sampling Fine-tuned (**RFT**) Autoformalizer (Yuan et al., 2023; Wu et al., 2022) 3) Verifier-Enhanced Autoformalizer (**VEA**) 4) Combined RFT and Verifier-Enhanced Autoformalizer (**RFT+VEA**). Results are presented in Table 4, and our analysis reveals three key findings:

**Effectiveness of Finetuning on FORML4**: Even our baseline model, which is finetuned on the FORML4 training data, significantly outperforms both open-source and closed-source LLMs across all test sets. This dramatic improvement indicates the effectiveness of our dataset and training approach in enhancing autoformalization performance.

**Complementary Strengths of RFT and VEA**: RFT significantly improves autoformalization across all test sets but is time-consuming due to its reliance on the Lean 4 compiler. In contrast, VEA offers a more time-efficient approach by using predictive labels from our trained verifier, though it may not match RFT's data quality. This trade-off between performance and efficiency suggests that these methods could be valuable in different scenarios, depending on the specific requirements of the task.

**Synergistic Benefits of Combined Approach**: The RFT+VEA model, which combines the strengths of both methods, shows the best performance across all test sets. This finding is particularly noteworthy, as it demonstrates that the verifier, despite being trained using feedback from the Lean 4 compiler, can contribute additional value when combined with direct compiler feedback for filtering training data. We propose this is due to the limitations of compilation alone in ensuring semantic alignment between formal and informal statements Lu et al. (2024b). The Lean 4 compiler can only validate the formal proof's correctness, not its semantic correspondence to the original natural language. In contrast, our verifier can take both the formal statement and the informal statement with proof as input, and the superior performance of RFT+VEA suggests a potential solution to the long-standing challenge of ensuring semantic alignment between formal and informal statements in autoformalization. The success of the combined RFT+VEA approach further underscores the potential for iterative improvements in autoformalization techniques.

## 5.3 FURTHER ENHANCED VERIFIER PERFORMANCE

We further enhance our verifier models using high-quality training data generated by the RFT+VEA autoformalizer. We compare outcome-supervision and process-supervision training methods as

Table 4: Performance of various LLMs on FORML4 in terms of greedy scores. We include both open-source and closed-source LLMs, as well as models finetuned on FORML4 training data. Reported results indicate the percentage of successfully compiled outputs over all the generated ones (%).

| Model | Test Sets | | |
|---|---|---|---|
| | Random Test | Basic Test | Real Test |
| **Closed-Source LLMs** | | | |
| GPT-3.5-Turbo (Achiam et al., 2023) | 0.43 | 0.31 | 5.23 |
| GPT-4-Turbo (OpenAI, 2023) | 0.52 | 1.51 | 5.35 |
| GPT-4o (OpenAI, 2023) | 1.38 | 1.53 | 5.85 |
| **Open-Source LLMs** | | | |
| DeepSeek-Math-Base-7B (Shao et al., 2024) | 0.21 | 0.38 | 0.03 |
| DeepSeek-Math-Instruct-7B (Shao et al., 2024) | 0.59 | 1.21 | 0.35 |
| LLEMMA-7B (Azerbayev et al., 2023b) | 0.03 | 0.20 | 0.02 |
| LLEMMA-34B (Azerbayev et al., 2023b) | 0.02 | 0.03 | 0.02 |
| InternLM-Math-7B (Ying et al., 2024b) | 0.03 | 0.22 | 1.13 |
| InternLM-Math-20B (Ying et al., 2024b) | 0.02 | 0.03 | 0.24 |
| Mistral-Instruct-v0.3-7B (Jiang et al., 2023b) | 0.30 | 0.48 | 0.33 |
| **Finetuned with FORML4** | | | |
| Baseline | 21.89 | 28.76 | 23.72 |
| RFT | 26.21 | 34.12 | 26.14 |
| VEA (Ours) | 25.87 | 33.95 | 25.91 |
| RFT + VEA (Ours) | 27.43 | 35.67 | 26.87 |

discussed in Section 4.1. "PSV+" indicates further fine-tuning under process-supervision, building upon "PSV," while "OSV+" signifies additional refinement from "OSV" with outcome-supervision.

Results for the verifier models comparison are presented in Table 5. It is important to note that autoformalized outputs are generated by the RFT+VEA model described in Section 5.2.2. A more detailed evaluation of the RFT+VEA model and further information on how we enhance verifier models are presented in Appendix J.

Table 5: Comparative performance of the enhanced verifier models.

| Dataset | OSV | | | OSV + | | | PSV | | | PSV + | | |
|---|---|---|---|---|---|---|---|---|---|---|---|---|
| | MP1 | Prec. | Recall | MP1 | Prec. | Recall | MP1 | Prec. | Recall | MP1 | Prec. | Recall |
| Basic | 34.13 | 75.22 | 80.19 | 39.08 | 81.17 | 85.24 | 36.11 | 76.25 | 81.18 | 41.09 | 82.21 | 87.26 |
| Random | 27.32 | 79.05 | 81.73 | 31.33 | 80.31 | 83.72 | 30.34 | 81.06 | 84.71 | 33.31 | 81.32 | 85.74 |
| Real | 28.14 | 75.23 | 78.33 | 35.12 | 81.22 | 80.31 | 30.13 | 76.21 | 79.32 | 37.11 | 83.22 | 81.33 |

**Improved Performance with High-Quality Data**: As demonstrated in Table 5, both the OSV+ and PSV+ models show improvements across all three evaluation metrics (MP1, precision, and recall) compared to their predecessors—OSV and PSV. This improvement is consistent across all datasets, substantiating the premise that fine-tuning with higher-quality data enhances both outcome-supervision and process-supervision training methods.

**Superior Efficacy of Process-Supervised Fine-tuning**: The results reveal that PSV+ consistently outperforms OSV+ across all metrics and datasets. In the Basic dataset, the PSV+ MP1 score is 41.09 compared to OSV+'s 39.08. Similarly, for the Real dataset, PSV+ achieves an MP1 score of 37.11, higher than OSV+'s 35.12. Additionally, PSV+ shows slightly superior precision and recall rates across all datasets, such as the 83.22% precision in the Real dataset, compared to 81.22% for OSV+. This suggests that process-based supervision leverages the training data more effectively, leading to better overall performance enhancements.

Table 5 demonstrates the potential for iterative training interaction among the autoformalizer, verifier, and Lean 4 compiler. The iterative improvement over the autoformalizer and verifier, supervised by the Lean 4 compiler, can be a promising direction for future work.

## 5.4 HUMAN EVALUATION ON AUTOFORMALIZER PERFORMANCES

Table 6: An overview of the human evaluation results of an autoformalization model. Significance tests are conducted using ANOVA and indicated by '*' in the table (*: p<0.05; **: p<0.01; ***: p<0.001).

| Variable | Overall | | Proof Validity** | | Model | | Dataset Split*** | | |
|---|---|---|---|---|---|---|---|---|---|
| | Avg | Fleiss' K | True | False | Baseline | Enhanced | Basic Test | Random Test | Real Test |
| **Evaluation Score** | 0.62 | 0.48 | 0.75 | 0.50 | 0.78 | 0.80 | 0.85 | 0.73 | 0.30 |

To accurately investigate the autoformalization performances of our PDA model in different settings, we conduct an extensive post-hoc human evaluation on the autoformalizers' output about whether the natural-language statements are successfully translated into formal statements.

**Goal** Human experts provide the most accurate evaluations of the semantic alignment between natural and formal languages, a task that the automated compiler struggles with, even when supplemented with additional proof steps, as observed in Lu et al. (2024b).

**Factorial Design** In particular, we investigate in detail whether the following variable changes will impact model autoformalization performances:

**Proof Validity**: whether the autoformalized sample can pass the Lean 4 compiler with both statement and proof. If false, it means that the statement along with the proof cannot pass the Lean 4 compiler, indicating that there is a logical fallacy either inside the statement itself or within proof steps. We group the sampled output so that half (30 samples) are labeled false in proof validity, and the other true. **PDA Enhancement**: whether the autoformalized sample is outputted by a baseline autoformalizer or a RFT + VEA enhanced autoformalizer in Table 12. **Test Set Categories**: Since the test sets vary in difficulty level and question types, we include the dataset split factor by extracting test sets in the closely identical proportional distribution as the full-size PDA test sets: random (20 samples): basic (20 samples): real (20 samples) $\approx 1 : 1 : 1$.

Based on the assigned factors in the evaluation samples, we investigate the following hypotheses to analytically support the validity of PDA method:

(i) Those whose proof validity is true achieve significantly better autoformalization performances. This will support our argument about PDA in using process-level compiler feedback from statement+proof to better indicate the semantic and logical validity of autoformalized statements. (ii) The enhanced autoformalizer achieves significantly better autoformalization performances. This can further support the validity of our enhancement approach to improve not only compiling successes but also human-evaluated semantic alignment. (iii) The autoformalization performance is higher on the basic and real test sets due to their lower difficulty and complexity.

**Results** As suggested in 5.4, our factor grouping statistics generally support the three hypotheses. Specifically, Proof Validity (p = 0.002992) and Dataset Split (p = 0.000002) show high significance in ANOVA results, supporting our first and third hypotheses. Regarding the comparison between the baseline and enhanced autoformalizer model, though the statistical significance is not obtained, we still find the higher evaluation score in the enhanced model consistent with our expectations.

## 6 CONCLUSION

In the current study, we introduce a new benchmark FORML4 specifically designed to assess the autoformalization capabilities of LLMs in Lean 4, and propose a processs-driven autoformalization (PDA) training pipeline with iterative process-level feedback. Unlike existing datasets, which focus on translating questions into statements, FORML4 focuses on extracting each statement's proof steps, enabling a more comprehensive, fine-grained, and effective evaluation of autoformalized statements. Importantly, PDA leverages the precise feedback naturally provided by the Lean 4 compiler to improve autoformalization, significantly enhancing performance and enabling more effective utilization of high-quality training data. For future work, we plan to extend our benchmark and apply our method to more formal languages such as Isabelle, HOL Light, Agda, and Coq.

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

## A    Future Directions

Our experiments demonstrate the effectiveness of our process-driven autoformalization framework:

1) The combined RFT+VEA approach leverages the strengths of both rejective sampling and verifier-based filtering, leading to superior autoformalization outcomes.

2) Process-supervised fine-tuning (PSV and PSV+) consistently outperforms outcome-supervised methods, indicating its ability to more effectively leverage training data.

3) The iterative improvement cycle between the autoformalizer, verifier, and Lean 4 compiler shows promise for further advancements in autoformalization.

Future work can focus on refining the process-supervision techniques and exploring more sophisticated ways to combine different enhancement methods. Additionally, investigating ways to reduce the time complexity of RFT while maintaining its data quality could lead to even more efficient autoformalization systems.

## B    More Related Works

**Formal Mathematics**    Formal languages, such as Isabelle (Wenzel et al., 2008), Lean (de Moura et al., 2015), HOL Light (Harrison, 1996), and Coq (Barras et al., 1997), have become integral tools in modern mathematics verification systems. These interactive theorem provers (ITPs) function as programming languages, allowing users to input statements and proofs in a formal language for automatic correctness verification. Among these ITPs, Lean 4 (de Moura & Ullrich, 2021) stands out for its recent advancements, offering full extensibility and addressing previous limitations (Ullrich & de Moura, 2019; 2022b;a; Nawrocki et al., 2023). However, keeping up with Lean 4's rapid development, including its evolving syntax, semantics, library, and other aspects, remains a challenge, even for human experts and powerful LLMs like GPT-4 (Achiam et al., 2023). To bridge this gap, we introduce FORML4 for training and testing autoformalization of LLM for Lean 4. Unlike the existing Lean 4 dataset MMA (Jiang et al., 2023a), which focuses on translating questions to statements, FORML4 provides a "complete" autoformalization from natural language questions and answers to statements and proofs in Lean 4. This more challenging task requires understanding Lean 4's syntax and the reasoning steps in each proof, enabling valuable feedback from the Lean 4 compiler on both syntax and reasoning verification.

**Formal Datasets**    The field of formal datasets has seen significant progress in extracting and cleaning theorems and proofs from established formal libraries and verification projects. Several datasets have been developed for popular proof assistants, focusing on extracting information from existing formalizations. For Coq, notable datasets include Gamepad (Huang et al., 2019), CoqGym (Yang & Deng, 2019), and PRISM (Reichel et al., 2023). For Isabelle, datasets like IsarStep (Li et al., 2021) and Magnushammer (Mikula et al., 2023) leverage the Archive of Formal Proofs and Isabelle Standard Library. Similarly, LeanStep (Han et al., 2022), LeanDojo (Yang et al., 2023b), and MLFMF (Bauer et al., 2023) utilize the mathlib library in Lean. LeanDojo, in particular, extracts

over 98,000 theorems and proofs with 130,000 premises from Mathlib. Beyond extracting data from existing projects, several works have focused on manually annotating or formalizing problems expressed in natural language. miniF2F (Zheng et al., 2022b) stands out by manually formalizing 488 Olympiad-level problems across four proof systems, equally splitting them into validation and test sets. FIMO (Liu et al., 2023a) and ProofNet (Azerbayev et al., 2023a) formalize theorem statements from IMO and undergraduate-level problems in Lean. For domain-specific problems, TRIGO (Xiong et al., 2023b) focuses on formalizing trigonometric reduction problems. UniGeo (Chen et al., 2022) and FormalGeo (Zhang et al., 2023) annotate proof steps for geometry proving problems. These datasets provide valuable resources for researchers working on automated theorem proving, proof verification, and natural language processing in the context of formal mathematics.

**Improving Reasoning Abilities of LLMs**   To enhance the reasoning capabilities of LLMs, prior research primarily focuses on specific prompting techniques. Existing efforts include few-shot prompting with intermediate steps augmented demonstrations (Wei et al., 2022; Wang et al., 2023b; Xiong et al., 2023a) or zero-shot prompting with specific instructions (Kojima et al., 2022; Yasunaga et al., 2023). Although these methods have shown promising results, their effectiveness is often constrained by their task-specific nature and the labour-intensive process of designing prompts, leading to inconsistent outcomes across different tasks (Ye & Durrett, 2022; Zhou et al., 2023). Another strategy to facilitate reasoning involves instruction tuning or knowledge distillation, which elicits reasoning paths from LLMs without explicit prompting (Mukherjee et al., 2023; Gunasekar et al., 2023; Lu et al., 2023; 2024c). These approaches typically involve resource-intensive fine-tuning over LLMs and require a large set of examples annotated with chain-of-thoughts (CoT). To address these challenges, verification techniques have emerged as a promising solution (Uesato et al., 2022; Lightman et al., 2024). Verification models are trained to evaluate and potentially correct the reasoning process generated by LLMs. This approach aims to mitigate the risk of relying solely on the top-1 result, which may not always be reliable (Wang et al., 2023a; Lu et al., 2024a).

**Learning From Feedback**   Improving LLMs through learning from feedback has become a prevalent strategy, notably through reinforcement learning from human feedback, which seeks to align LLMs with human values by refining their outputs based on feedback (Ouyang et al., 2022; Bai et al., 2022). However, this method faces challenges such as high costs due to manual labor and a lack of real-time feedback capabilities. An alternative strategy involves using self-correcting LLMs, which rely on automated feedback to iteratively adapt and understand the consequences of their actions without heavy reliance on human intervention. This feedback can be derived from inside sources such as the model itself (Madaan et al., 2023; Shinn et al., 2023) or generation logits (Yao et al., 2024), and outside sources such as tools (Gou et al., 2023; Huang et al., 2024a), knowledge bases (Gao et al., 2023; Yu et al., 2023b), or evaluation metrics (Jung et al., 2022; Welleck et al., 2023). Our method leverages formal languages that can naturally provide precise feedback on the reasoning process, enabling automatic process annotation without substantial human or machine annotation costs.

## C   DETAILED DECOMPOSITION STRATEGY

Our decomposition strategy for informalization involves instructing the model to perform the following subtasks sequentially:

1. Translate the formal statement into a natural-language problem.
2. Explain the meaning of each step of the formal proof in natural language, based on the definition of the employed lemma or tactics.
3. Write a step-by-step proof of the problem in natural language without verbatim mention of any Lean 4 function.

For FORML4 construction, we extract only the translated natural-language problem and the step-by-step proof from the model output to form the natural-language data.

The strategy of explaining each tactic step before writing the natural-language proof serves two crucial purposes: 1. It creates a reasoning buffer for the model. 2. It effectively differentiates between "listing and explaining each Lean 4 term from the formal proof in natural language" and "proving the problem statement step by step in natural language", with the latter being our intended goal.

Our empirical observations indicate that a naive instruction prompt without decomposition often leads to ambiguity. Models tend to write natural-language proof steps by explaining each term in the formal proof steps (and even in the formal statement), regardless of verbal emphasis on the distinction. This approach would render autoformalization evaluation meaningless, as the formal content would already exist in the input.

In contrast, decomposing our complex goal into separate subtasks effectively addresses this issue, as verified by human expert evaluators. The decomposition strategy ensures that the resulting natural language proof is genuinely independent of the formal proof structure, making it suitable for autoformalization tasks.

We further enhance the strategy by adding few-shot examples to better align the model with our expected format and goal. The complete prompt template, including these examples, can be found in Appendix D.

## D    PROMPT FOR INFORMALIZATION

Below is the few-shot prompt template for querying an LLM to perform formalization. The few-shot examples are carefully curated to ensure the semantical equivalence, logical validity, and readability of natural language translations.

---

Given a statement and its proof written in Lean 4's syntax, please translate them into the semantically equivalent natural language that a human reader can independently understand without knowing any concepts in Lean 4. The translation should accurately convey the same logical structure and content as the original statement and proof.

You need to explain the theorem and proof in the most intuitive terms possible, but also maintain the fidelity of the original mathematical reasoning. To do so, first translate the theorem statement into a natural language problem so that it does not contain any function in lean 4 (write after "# Problem:"). Then for the proof, you can explain each step of the proof in natural language based on the meaning of the lemma or tactic that is used (write after "# Explanation:"). Lastly and most importantly, write the step-by-step proof for the problem in natural language without mentioning verbatim any function in Lean 4 (write after "# Proof:").
—

Follow the format below.
# Theorem: (theorem and proof in lean 4, to be translated)
# Problem: (theorem in natural language)
# Explanation: (proof in natural language, explaining the functions in lean 4)
# Proof: (proof in natural language, understandable by any human reader without the knowledge of lean 4 functions)
—

Here are some examples:
**# Theorem:** $theorem\ eq\_zero\_iff\_even\ \{n : \mathbb{N}\} : (n : ZMod\ 2) = 0 \iff Even\ n := (CharP.cast\_eq\_zero\_iff\ (ZMod\ 2)\ 2\ n).trans\ even\_iff\_two\_dvd.symm$
**# Problem:** Prove that for any natural number $n$, $n$ is even if and only if $n$ is congruent to 0 modulo 2.
**# Explanation:** The proof uses the following chain of reasoning:
1. `CharP.cast_eq_zero_iff (ZMod 2) 2 n`: This lemma states that for any natural number $n$, $n$ is congruent to 0 modulo 2 if and only if the remainder when $n$ is divided by 2 is 0.
2. `even_iff_two_dvd.symm`: This lemma states that a number is even if and only if it is divisible by 2.
3. `.trans`: This tactic combines the two lemmas by showing that if $n$ is congruent to 0 modulo 2, then the remainder when $n$ is divided by 2 is 0, and therefore $n$ is divisible by 2, which means $n$ is even.
**# Proof:** We need to prove both directions of the "if and only if" statement.
**Direction 1: If $n$ is even, then $n$ is congruent to 0 modulo 2.**
If $n$ is even, then by definition, $n$ is divisible by 2. This means that the remainder when $n$ is

divided by 2 is 0. Therefore, $n$ is congruent to 0 modulo 2.
**Direction 2: If $n$ is congruent to 0 modulo 2, then $n$ is even.**
If $n$ is congruent to 0 modulo 2, then the remainder when $n$ is divided by 2 is 0. This implies that $n$ is divisible by 2. Hence, $n$ is even.
Since we have proven both directions, we have shown that a natural number $n$ is even if and only if $n$ is congruent to 0 modulo 2.

# Theorem: $theorem\ forall\_mem\_comm\ \{\alpha\ \beta\}\ [Membership\ \alpha\ \beta]\ \{s\ :\ \beta\}\ \{p\ :\ \alpha \to \alpha \to Prop\}$ :
$(\forall a\,(\_ : a \in s)\, b\,(\_ : b \in s),\ p\,a\,b) \iff \forall a\,b,\ a \in s \to b \in s \to p\,a\,b :=$
$forall\_cond\_comm$
# Problem: Prove that for any set $s$, a property $p$ holds for all elements $a$ and $b$ in $s$ if and only if, for every pair of elements $a$ and $b$ in the set $s$, the property $p$ holds between them.
# Explanation:
1. The original statement involves checking whether a property $p$ holds for elements $a$ and $b$ in a set $s$.
2. The left-hand side of the equivalence states that for every $a$ in $s$, for every $b$ in $s$, the property $p(a, b)$ holds.
3. The right-hand side of the equivalence restates this, but in a more traditional way, using implications. It says that for every $a$ and $b$, if $a \in s$ and $b \in s$, then $p(a, b)$ holds.
4. The tactic `forall_cond_comm` helps translate between these two forms, essentially commuting the logical structure of the quantifiers and conditions.
# Proof:
We need to show that these two forms are logically equivalent.
**First direction (left to right)**:
Suppose we are given that for all elements $a \in s$, for all $b \in s$, the property $p(a, b)$ holds. This directly means that, for any $a$ and $b$, if both $a$ and $b$ are in the set $s$, then $p(a, b)$ is true. Therefore, if $a \in s$ and $b \in s$, we know that $p(a, b)$ holds by the original assumption.
**Second direction (right to left)**:
Now assume that for every pair of elements $a$ and $b$, if $a \in s$ and $b \in s$, then $p(a, b)$ holds. This means that for any $a \in s$, we can take any $b \in s$, and the property $p(a, b)$ must hold. Thus, the condition on $p$ is satisfied for all such pairs within $s$.
Since both directions of the equivalence are proven, the two forms of the statement are logically equivalent. Therefore, the property $p$ holds for all pairs of elements in the set $s$ if and only if, for each $a \in s$ and $b \in s$, the property $p(a, b)$ holds.

# Theorem: $theorem\ asq\_pos\ :\ 0\ <\ a * a :=$
$le\_trans\ (le\_of\_lt\ a1)$
$(by\ have := @Nat.mul\_le\_mul\_left\ 1\ a\ a\ (le\_of\_lt\ a1);\ rwa\ [mul\_one]\ at\ this)$
# Problem: Prove that the square of any natural number is greater than 0.
# Explanation: The proof uses the following steps:
1. `a1`: This refers to the fact that $a$ is greater than 0, which is a premise of the theorem.
2. `le_of_lt a1`: This lemma states that if $a$ is greater than 0, then 0 is less than or equal to $a$.
3. `Nat.mul_le_mul_left 1 a a (le_of_lt a1)`: This lemma states that if 1 is less than or equal to $a$ (which is true because $a$ is a natural number and greater than 0), and 0 is less than or equal to $a$ (from step 2), then 1 multiplied by $a$ is less than or equal to $a$ multiplied by $a$.
4. `rwa [mul_one] at this`: This tactic rewrites the goal using the lemma `mul_one`, which states that 1 multiplied by any number is equal to that number. This simplifies the goal from $1 * a \le a * a$ to $a \le a * a$.
5. `le_trans`: This tactic combines the inequality $0 \le a$ (from step 2) with the inequality $a \le a * a$ (from step 4) to conclude that $0 < a * a$.
# Proof:
1. We know that $a$ is a natural number greater than 0.
2. Since $a$ is greater than 0, we can say that $0 \le a$.
3. We can also say that 1 is less than or equal to $a$ because $a$ is a natural number and greater

than 0.

4. Multiplying both sides of the inequality $0 \leq a$ by $a$, we get $0 * a \leq a * a$.

5. Since multiplying any number by 0 results in 0, the inequality becomes $0 \leq a * a$.

6. We also know that multiplying any number by 1 doesn't change its value, so $1 * a$ is the same as $a$.

7. Combining this with the fact that $1 \leq a$, we get $a \leq a * a$.

8. Since $0 \leq a$ and $a \leq a * a$, we can conclude that $0 < a * a$.

9. Therefore, the square of any natural number is greater than 0.

**# Theorem**: {Theorem}
**# Problem**:

## E  HUMAN EVALUATION: COMPARATIVE MODEL SELECTION

### E.1  ANNOTATION PROTOCOL FOR INFORMALIZATION MODEL COMPARISON

#### E.1.1  INTRODUCTION

The protocol provides guidance for evaluating the quality of informalization of two sampled models (gpt4-o and gemini-pro-1.5). Two models are tasked to translate theorem statements and their proof written in Lean 4 syntax to natural language (i.e., informalization), so that the natural language problem statement and proof can be understood by readers without any Lean 4 knowledge.

Given a theorem and proof, the models are prompted to respond following the format below:

- **Theorem:** (the given theorem and proof in Lean 4, to be translated)

- **Problem:** (translated theorem statement in natural language)

- **Explanation:** (proof in natural language, explaining the functions in Lean 4)

- **Proof:** (proof in natural language, understandable by any human reader without the knowledge of Lean 4 functions)

#### E.1.2  FILE STRUCTURE

You are given two model output .json files (sample size = 10). In the file, each sample contains five items:

- **"nl":** (past informalized output. Ignore)

- **"formal":** formal statement and proof in Lean 4 (i.e., Theorem)

- **"gemini_output" / gpt4o_output:** complete model output

- **"nl_problem":** extracted from model output (i.e., Problem)

- **"nl_explanation":** extracted from model output (i.e., Explanation)

- **"nl_proof":** extracted from model output (i.e., Proof)

Among them, your annotations focus on the quality of "nl_problem" and "nl_proof" per sample.

#### E.1.3  TASK

For both model output .json files, you need to annotate three items:

1. **Informalization Success (T/F):** whether the translation from **"formal"** statement to **"nl_problem"** is semantically equivalent. The natural-language translation should accurately convey the same logical structure and content as the original statement in Lean 4.

2. **Informal Proof Correctness (T/F):** whether the informalized proof **"nl_proof"** successfully proves the problem statement **"nl_problem"**, and can be independently understood without prior knowledge of Lean 4.

3. **Model Preference (T/F):** Compare the informalization output (i.e., **"nl_problem"** + **"nl_proof"**) between gemini and gpt4o, choose which one is preferable based on the criteria described below. Label T if preferred, F if not.

### E.1.4   QUALITY CRITERIA

The ideal informalized output should meet the following criteria:

1. **Semantically Equivalent to the Lean 4 Theorem and Proof:** (informalization success = T)

2. **Intuitive Terms without Lean 4 Functions:** Both problem statement and proof use intuitive terms without Lean 4 functions mentioned, proves the intended theorem successfully, and can be independently understood without prior knowledge of Lean 4. (informal proof correctness = T)

Check the instruction and demo examples in the fewshot prompt for reference of an ideal informalization case. You can use tools like https://jsoneditoronline.org/ to compare two model output files more easily.

### E.2   ANNOTATION RESULTS FOR INFORMALIZATION MODEL COMPARISON

Table 7: Comparison of Model Evaluation in Three Metrics

| Model | Metric | Average True Rate | Inter-Rater Agreement |
|---|---|---|---|
| **Gemini** | Informalization Success | 80.0% | 80.0% |
| | Informal Proof Correctness | 85.0% | 70.0% |
| | Model Preference | 60.0% | 60.0% |
| **GPT4o** | Informalization Success | 72.5% | 45.0% |
| | Informal Proof Correctness | 62.5% | 45.0% |
| | Model Preference | 22.5% | 55.0% |

## F   HUMAN EVALUATION: PDA DATASET QUALITY

After obtaining the final dataset, we perform a more extensive manual verification on the informalized dataset, compared to the preliminary one in the model selection stage. Because the core goal of FORML4 is to train and evaluate statement autoformalization, the human verification task only includes annotating the informalization success specific to statement translation. We recruited a different group of four human experts in Lean 4 than in the model selection stage to perform manual quality checks on 60 samples. Among them, 20 samples come from the basic test set, and 40 from the random test/train set.

The average success rate evaluated by human experts is 0.72, indicating a relatively high-quality informalization performance. The intra-rater standard deviation of 0.44 suggests moderate variability in individual assessments while inter-rater Fleiss' Kappa is 0.3730, showing fair agreement among four raters, highlighting a reasonable level of consensus in evaluations.

The split stats between the basic test set (0.875) and the random test set (0.575) show a significant discrepancy in the human-verified informalization success rate ($p = 0.0099$), suggesting that informalization difficulty increases with formal theorem complexity.

Notably, all four human evaluators comment on the same two challenges during the annotation task:

1. The incompatibility of certain theorem statements for informalization due to their topics or settings. In practice, our Lean 4 experts observe that it is sometimes *infeasible* to perfectly

translate a set of formal proofs to a natural language. This is because formal proofs are often expressed in pre-defined lemmas or environments that are exclusively constructed in the Lean 4 language, and there are no existing corresponding concepts in natural language that a non-expert in Lean 4 could easily understand.

2. Individual subjectivity in determining the condition constraints that need to be specified in natural language (Azerbayev et al., 2023a; Ying et al., 2024a).

As emphasized in past autoformalization research, such challenges are due to the highly parallel gap between formal and natural language, with the former requiring precision and syntactic rigidity while the latter suffering from ambiguity and reliance on contexts (Liu et al., 2023a; Jiang et al., 2023a). As the formal theorem complexity rises, it likely widens such a gap that the informalization difficulty also increases. This is reflected in the split stats between the basic test set (0.875) and the random test set (0.575) show a significant discrepancy in the human-verified informalization success rate (p = 0.0099).

## G    CASE VISUALIZATION IN COMPARISION WITH EXISTING DATASETS

In addition to the summarized comparison of dataset features in 2, below we also provide a visualization comparison through a data example with the same statement in both our FORML4 training set and existing training sets (Jiang et al., 2023a; Ying et al., 2024a). As shown in Table 8, our FORML4 incorporates both the informal statement and its proof as input for our autoformalization process, making it a complete autoformalization task. In contrast, the MMA, one of the existing datasets, requires the model to output only the statement, without the proof.

Our task requires the model to not only understand the basic Lean 4 syntax rules but also comprehend the logical relationships present in the proof process, such as dependencies illustrated in the example. When compiling our output examples using the Lean 4 compiler, we require a complete theorem output. Therefore, the feedback from the Lean 4 compiler is more comprehensive, providing syntax checking for both statements and proofs, coupled with reasoning checking to validate the proofs.

This comprehensive feedback is crucial for guiding the enhancement of autoformalization within our framework, as described in Section 5.2. The 'tactic' feedback indicates that our example successfully verifies the goal of proving that the cosine of the angle $\pi$ (pi), when measured in radians, is equal to -1. In the MMA case, due to the absence of a proof, the Lean 4 compiler can only return a warning that the theorem is incomplete.

In summary, the feedback from the Lean 4 compiler provides syntax checking and reasoning verification for both statements and proofs, which is essential for improving autoformalization in our framework. In contrast, the feedback from the existing dataset is limited to syntax checking of statements, lacking the depth of reasoning verification.

## H    HUMAN EVALUATION: AUTOFORMALIZATION PERFORMANCE EVALUATION

The same four annotators for the FORML4 informalization verification task are asked to cross-evaluate 60 autoformalized samples. Each sample is annotated twice. It takes each annotator approximately 5 minutes to complete evaluating a sample.

## I    EXPERIMENTAL DETAILS

### I.1    TRAINING SETTINGS

Our experiments were conducted in a computing environment equipped with 8 NVIDIA A100 GPUs, each having 40GB of memory. All models underwent fine-tuning in a full-parameter setting. We employed the AdamW optimizer for model training over 2 epochs, with a batch size of 128. The learning rate was set at $5 \times 10^{-6}$, incorporating a 3% learning rate warmup period. Below, we present a comprehensive overview of the training hyperparameters utilized. These parameters were consistently applied across training autoformalizer models in our experiments in Table 9.

Table 8: Comparison of one data example from FORML4 and existing datasets.

| Aspect | PDA | MMA |
|---|---|---|
| Input | *Statement and proof in natural language:*

# Statement: The statement we're examining asserts that the cosine of the angle $\pi$ (pi), when measured in radians, is equal to -1. This is a fundamental result in trigonometry, capturing a key property of the cosine function on the unit circle.

# Proof: The proof provided in the Lean 4 syntax is brief and relies on two key elements: the 'cos_coe' lemma and the 'Real.cos_pi' fact.

*Translate the statement and proof in natural language to Lean:* | *Statement in natural language:*

# Statement: The cosine of pi, when pi is considered as an angle, equals -1.







*Translate the statement in natural language to Lean:* |
| Output | ```theorem cos_coe_pi : cos
    (π : Angle) = −1 :=
by rw [cos_coe,
    Real.cos_pi]``` | ```theorem cos_coe_pi : cos
    (π : Angle) = −1 :=``` |
| Feedback | ```"tactic": "rw [cos_coe,
    Real.cos_pi]",
"proofState": 99,
"goals": "⊢ cos ↑π= −1"``` | ```"severity": "warning",
"proofState": 0,
"data": "declaration uses
    'sorry'"}],``` |

Table 9: Autoformalizer training hyperparameters.

| Hyperparameter | Global Batch Size | LR | Epo. | Max Length | Weight Decay | Warmup Ratio |
|---|---|---|---|---|---|---|
| Value | 128 | $5 \times 10^{-6}$ | 2 | 2048 | 0 | 0.03 |

For training verifier, the setting is as shown in Table 10.

Table 10: Verifier training hyperparameters.

| Hyperparameter | Global Batch Size | LR | Epo. | Max Length | Weight Decay | Warmup Ratio |
|---|---|---|---|---|---|---|
| Value | 512 | $2 \times 10^{-6}$ | 1 | 2048 | 0 | 0.03 |

## I.2 GENERATION SETTINGS

In this section, we specify the settings used for model generation to ensure reproducibility across all experiments, including baseline models and variations enhanced with verifiers.

For the generation of results using the "greedy" strategy, we set the temperature parameter to 0.0 and 0.7 for the "pass@k" strategy. To present unbiased results for "pass@k", we follow the calculation

method outlined in (Chen et al., 2021). Specifically, we generate $n = 20$ samples for each instance, evaluate the number of correct samples passing unit tests, and then calculate the unbiased estimator for pass@k.

It's important to note that all generation scripts are based on the vLLM framework (Kwon et al., 2023) for efficient inference of LLMs.

### I.3 LEAN 4 COMPILATION

In this section, we outline the specific versions of libraries utilized and the details about the compilation process in Lean 4 in our experiments.

**Lean 4 Compiler:** The Lean 4 Compiler is a critical component of the Lean 4 programming language. This tool enables users to craft effective proof automation tactics within the Lean environment and transform them into optimized C code. The Lean 4 Compiler in our scope is referred to as the tool available at `https://github.com/leanprover-community/repl`. This particular resource provides a read-eval-print loop (REPL) designed for Lean 4, which supports user interaction through JSON formatted input and output streams (stdin and stdout, respectively). Our compilation projection is therefore founded on REPL. We also developed a multiprocessing framework to streamline the compilation of Lean 4, which is attached in the supplementary material.

**Standard library:** We acknowledge that Lean 4 is still in active development, as are its associated libraries such as mathlib and others. To maintain consistency and reproducibility, we fixed our Lean 4 version from the official website. We specify the versions and sources of required libraries as shown in Table 11.

Table 11: Library versions and sources of Lean 4.

| Name | URL | Revision | Input Revision |
|---|---|---|---|
| mathlib | https://github.com/leanprover-community/mathlib4 | 3cecb82 | 3cecb82 |
| std | https://github.com/leanprover/std4 | e5306c3b | main |
| Qq | https://github.com/leanprover-community/quote4 | fd76083 | master |
| aesop | https://github.com/leanprover-community/aesop | 8be30c2 | master |
| proofwidgets | https://github.com/leanprover-community/ProofWidgets4 | fb65c47 | v0.0.30 |
| Cli | https://github.com/leanprover/lean4-cli | be8fa79 | main |
| importGraph | https://github.com/leanprover-community/import-graph.git | 61a7918 | main |

**Running Time:** Lean 4's compilation times are a bottleneck. The compilation duration varies depending on factors such as theorem complexity, dependencies on relevant lemmas or theorems, etc. Compiling 1k examples requires around 10 minutes. This duration is notably longer than the generation time for a large language model, which typically takes only 1-2 minutes to generate output on 1k samples.

## J COMPREHENSIVE EVALUATION OF THE ENHANCED AUTOFORMALIZER

We leverage our enhanced autoformalizer to generate high-quality training data, supervised by the Lean 4 compiler, to further refine the verifier model. This process involves the following steps:

1. **Data Generation:** We employ the RFT+Verifier enhanced autoformalizer to produce samples from the FORML4, MATH, and GSM8K training sets.

2. **Compilation Testing:** Each generated sample undergoes testing via the Lean 4 compiler to ascertain compilation success and extract detailed compilation information.

3. **Verifier Fine-tuning:** We further refine the Process-Supervised Verifier (PSV) model using this high-quality data, incorporating step-level process supervision derived from the compiler's feedback.

To assess the efficacy of our refined verifier, we first evaluate the comprehensive performance of the RFT+Verifier enhanced Autoformalizer (RFT + VEA) model. This evaluation employs both greedy decoding and pass@k sampling methods, as detailed in Appendix P.2. Table 12 presents these results.

Table 12: Comprehensive performance of the enhanced autoformalizer.

| Model | Dataset | Greedy | Pass@1 | Pass@5 |
|---|---|---|---|---|
| | Basic | 35.67 | 33.14 | 43.11 |
| RFT + VEA | Random | 27.43 | 26.47 | 36.19 |
| | Real | 23.72 | 22.29 | 40.33 |

## K  MODEL SELECTION PROCESS

**Model Selection Process:** Our model selection process involved a rigorous comparative evaluation of GPT-4 and Gemini-Pro-1.5. We sampled 10 inputs from the extracted formal theorems and recruited four human annotators to cross-evaluate the informalization outputs of both models. The evaluation was based on three key metrics:

1. Success of statement informalization: Assessing whether the translated natural-language statement is logically accurate and semantically equivalent to the formal statement.

2. Informalized proof correctness: Evaluating whether the translated natural-language proof is logically valid to prove the statement.

3. Model preference: Determining whether the translation output of one model is preferred over the other, with only one 'True' value allowed per sample.

Both models demonstrated satisfactory performance in informalization success (Gemini-Pro-1.5: 80%; GPT-4: 70%) and informalized proof correctness (both at 80%). However, Gemini-Pro-1.5 consistently achieved higher scores with a high interrater agreement rate of 0.77. Moreover, when tasked to cross-compare the model outputs based on their statement and proof generation, annotators preferred Gemini-Pro-1.5 in 80% of the samples.

The detailed annotation protocol and comprehensive results of this evaluation process are provided in Appendix E.

## L  DETAILED DATASET COMPARISON ANALYSIS

Table 2 compares FORML4 with existing autoformalization datasets. Here, we provide a detailed explanation and analysis for each characteristic:

**Source Language:** FORML4 and MMA use formal language as their source, while others use natural language. This approach aligns with empirical findings by Jiang et al. (2023a) suggesting that informalization is generally easier than formalization, potentially leading to higher-quality datasets.

**Size:** With more than 17k entries, FORML4 is significantly larger than most datasets except MMA and Lean Workbook. This size allows for more comprehensive training and evaluation of autoformalization models.

**Includes Proofs:** FORML4 and ProofNet are the only datasets that include proofs along with statements. This feature is crucial for training process-driven autoformalizers and enables a more holistic approach to mathematical reasoning.

**Uses Lean 4:** FORML4, MMA, and Lean Workbook use Lean 4, a modern theorem prover. This choice ensures compatibility with current formal verification tools and practices.

**Construction Method** (1) Direction: FORML4 and MMA use informalization, while others use formalization. The informalization approach may lead to more natural-sounding informal statements and potentially easier dataset creation. (2) LLM-based: FORML4, MMA, Lean Workbook, and FIMO use LLMs in their construction, leveraging recent advances in AI to create large-scale datasets

efficiently. (3) Human-Verified: All datasets except MMA incorporate human verification, ensuring higher data quality. FORML4's rigorous verification process, including task decomposition and data inspection, sets it apart.

**Primary Usage** (1) Training: FORML4, MMA, and Lean Workbook are suitable for training, unlike smaller datasets like ProofNet, miniF2F, and FIMO, which are primarily for benchmarking. (2) Benchmarking: All datasets can be used for benchmarking, allowing for comprehensive evaluation of autoformalization models across different dataset characteristics. (3) Process-Driven Feedback: FORML4 and ProofNet uniquely offer process-driven feedback, crucial for training iterative autoformalizers. FORML4's approach is fully automated, using the formal language compiler to process proof steps and provide annotated feedback.

## M   ANALYSIS FOR TRAINING AND TEST DATA IN FORML4

To showcase the connection between the training data provided by FORML4 and the test sets, we conduct standard supervised fine-tuning on the Mistral-7B (Jiang et al., 2023b) model using the training data provided by FORML4, with training hyperparameters detailed in Appendix I.1. We compare it with a model trained on 5k sampled training data provided by FORML4. Their autoformalization performance on our three test sets is listed in Table 13.

Table 13: Comparison of models trained on different data sizes.

| Model | Basic | Random | Real |
|---|---|---|---|
| Mistral | 0.12 | 0.00 | 0.21 |
| Mistral (5K) | 20.12 | 16.19 | 2.82 |
| Mistral (Full) | 28.87 | 21.47 | 5.34 |

We demonstrate the following insights:

**Training Data Always Matters**: Our study reveals a strong correlation between the test and training data provided in our FORML4. By enlarging the training dataset from 5k to full 14.51k samples, we observe a notable improvement in the compilation rate on three test sets. This indicates that increasing the training data size positively impacts the model's performance on the test sets, as shown in Table 13.

**Real Test is Still Challenging**: Despite the improvements observed in all test sets, there remains substantial room for enhancement in the real test set, i.e., the natural language-based benchmark as shown in Table 13. This discrepancy can be attributed to two primary factors: i. Out-of-Distribution Test Domains: The real test set represents OOD test domains compared to the two Mathlib Lean 4 test sets, i.e., Random and Basic. Consequently, models fine-tuned solely on the Mathlib Lean 4 training set may struggle to generalize effectively to these benchmarks. ii. Lack of Dependency on Pre-Defined Lemmas or Basic Terms: Unlike Mathlib Lean 4 test sets, the real test set often lacks dependencies on pre-defined lemmas or basic terms.

Additionally, we evaluate the autoformalization efficiency on two Math Reasoning benchmarks, i.e., GSM8K (Cobbe et al., 2021) and MATH (Hendrycks et al., 2021) in Table 14 in Appendix N. We note that the SFT model exhibits different performance on the real test sets compared to the baseline model listed in Table 4. This is because this section aims to explore the connection between the training and test sets provided by FORML4. Therefore, the two SFT models in this section do not undergo further rejective sampling fine-tuned on the MATH and GSM8K datasets, as described in the Section 5.2.1.

## N   AUTOFORMALIZATION ON REAL-WORLD MATHEMATICAL REASONINGS

In this section, We list the results of using the SFT model trained in Appendix M, to perform autoformalization based on questions and answers in GSM8K and MATH training sets. The results are presented in the following Table 14.

Table 14: Comparison of the SFT model's autoformalization performance, measured by compilation rate (%), on the GSM8K and MATH training sets.

| Model | MATH | GSM8K |
|---|---|---|
| Mistral | 0.0 % | 0.0 % |
| Mistral (5K) | 0.55 % | 3.28 % |
| Mistral (Full) | 0.65 % | 8.16 % |

The results in Table 14 demonstrate that despite fine-tuning with training sets provided by FORML4, the model's performance on autoformalization tasks for GSM8K and MATH was still not satisfactory. To address this weakness, we employed Mistral (Full) to conduct the autoformalization task on training sets from GSM8K and MATH. For each example, we generated 10 samples with a temperature of 0.7. The outputs that were successfully compiled by the Lean 4 compiler were then used to further fine-tune a final **baseline** model utilized in Section 5.2.2.

## O   DATA CONSTRUCTION DETAILS

### O.1   DATA PREPROCESSING

Firstly, we retain the "#align" command within the proof, which is used by Mathport[6] to connect Lean 3 names to Lean 4 names. This inclusion is intended to facilitate the informalization process for GPT-4 during data construction, as we hypothesize that GPT-4 will better understand the Lean 4 language if there is a connection to the more familiar Lean 3 language.

Secondly, all samples with custom Mathlib 4 lemma (as indicated by the '.mk' suffix) in the theorem statement are removed. This is because such lemmas are custom-defined under the same file of the theorem inside the Mathlib 4 library, hence the model will have no access to its definition, causing inevitable ambiguity or uncertainty in informalization[7]. Altogether 262 samples are filtered, with 236 from the train set, 6 from the basic test set, and 20 from the random test set.

Lastly, 35 samples in the real test set specified to be solved in Python are removed for being unsuitable for autoformalization evaluation.

### O.2   DATASET SPLIT

The basic test and the real test set are the two added test set data in FORML4 for a more domain-inclusive evaluation of autoformalization. They are collected from distinctive sources compared to the random test set or training data and aimed at assessing nuanced domains of autofomalization capability. Below are detailed descriptions of their features, content, and data creation processes.

**Basic Test**   It assesses the model's ability to autoformalize basic theorems with minimal reliance on prior knowledge or established lemmas. These theorems typically appear in files like `Mathlib/Geometry/Euclidean/Basic.lean`, which establish fundamental geometrical concepts and prove simple results about real inner product spaces and Euclidean affine spaces. Conversely, theorems with more intricate proofs or richer geometrical content are usually found in separate files, like `Mathlib/Geometry/Euclidean/Triangle.lean`, and are excluded from the Basic Test.

From all the `Basic.lean` files across various mathematical subjects (like geometry and algebra), we extract roughly 10,000 theorems. After removing the sampled training and random test sets from this pool, we randomly select theorems to create the Basic Test. This ensures that the Basic Test remains entirely exclusive from the training and random test sets.

---

[6]https://github.com/leanprover-community/mathport

[7]We tried tracking and appending the definitions of custom lemmas to the model input as contexts. This did not significantly improve the models' informalization outcomes.

**Real Test**    To evaluate our models' ability to handle real-world scenarios, we constructed a real test set by collecting natural language math questions and answers from LI et al. (2024). This real test set assesses how well our models can automatically formalize natural language expressions, providing a more comprehensive evaluation metric.

Since this set is derived from real math questions, we do not preprocess them using GPT-4 for informalization. It's important to note that this real test set lacks any inherent dependencies on predefined lemmas or basic Lean 4 terms, unlike the environment we typically use for Lean 4 programming. We follow the setting of the Lean 4 version of LeanDojo (Yang et al., 2023a) and employ its predefined theorem environment as shown in `https://github.com/yangky11/miniF2F-lean4/blob/main/MiniF2F/Minif2fImport.lean` for all real test examples.

### O.3    DATA POSTPROCESSING

We apply a post-filtering process to both the training and test sets to uphold the quality of data examples. The exclusion criteria were as follows:

- Instances where the API failed or produced empty content during the informalization stage.
- Cases where the length of the natural-language question or answer did not exceed 400 characters, or the length of the formalized theorem and proof did not exceed 200 characters. This step ensures that each datapoint retains complexity and richness for the autoformalization task.
- Situations where the informalization was evidently incorrect were manually reviewed and removed. It is important to note that this manual check was not applied to the entire dataset.

## P    DETAILS OF AUTOMATED AUTOFORMALIZATION EVALUATION

### P.1    PROMPT

We used a specific instruction prompt for autoformalization with all existing LLMs. The prompt is as follows:

---

Statement and proof in natural language:

\# Statement:

$$Statement$$

\# Proof:

$$proof$$

Translate the statement and proof in natural language to Lean 4:

---

For the instruction-finetuning model, we used the prompt template and inserted our autoformalization prompt into their template to ensure consistent performance.

### P.2    PREPROCESSING AND EVALUATION

The model's response may contain raw text mixed with Lean 4 language, We applied different handling functions to extract the exact Lean 4 language for subsequent compilation. For model responses without any Lean 4 output, we marked them as negative outputs. We employ the metric **pass@k** to evaluate model performance, defined as the condition where at least one autoformalized instance, comprising both the statement and proof, successfully passes the Lean 4 compiler within the model's first k attempts. Additionally, we use the term **greedy** to assess model performance based on whether the output with the highest confidence from the model can pass the Lean 4 compiler.

Table 15: Performance of LLMs on FORML4 in terms of greedy and pass@k scores. We include open-source LLMs that claim integration of formal languages into their pretraining/finetuning. Reported results indicate the percentage of successfully compiled outputs over all the generated ones (%).

| Model | Random Test | | | Basic Test | | | Real Test | | |
|-------|-------------|---------|---------|------------|---------|---------|-----------|---------|---------|
| | Greedy | Pass@1 | Pass@5 | Greedy | Pass@1 | Pass@5 | Greedy | Pass@1 | Pass@5 |
| **Closed-Source LLMs** | | | | | | | | | |
| GPT-3.5-Turbo (Achiam et al., 2023) | 0.43 | 0.34 | 0.75 | 0.31 | 0.02 | 0.68 | 5.23 | 3.92 | 10.21 |
| GPT-4-Turbo (OpenAI, 2023) | 0.52 | 0.44 | 3.48 | 1.51 | 1.18 | 4.45 | 5.35 | 4.83 | 12.32 |
| GPT-4o (OpenAI, 2023) | 1.38 | 1.14 | 3.51 | 1.53 | 1.20 | 5.47 | 5.85 | 5.38 | 13.31 |
| **Open-Source LLMs** | | | | | | | | | |
| DeepSeek-Math-Base-7B (Shao et al., 2024) | 0.21 | 0.25 | 0.96 | 0.38 | 0.22 | 0.86 | 0.03 | 0.02 | 0.04 |
| DeepSeek-Math-Instruct-7B (Shao et al., 2024) | 0.59 | 0.26 | 1.73 | 1.21 | 0.48 | 3.08 | 0.35 | 1.63 | 5.39 |
| LLEMMA-7B (Azerbayev et al., 2023b) | 0.03 | 0.02 | 0.79 | 0.20 | 0.13 | 0.45 | 0.02 | 0.03 | 0.04 |
| LLEMMA-34B (Azerbayev et al., 2023b) | 0.02 | 0.03 | 0.19 | 0.03 | 0.02 | 0.03 | 0.02 | 0.03 | 0.04 |
| InternLM-Math-7B (Ying et al., 2024b) | 0.03 | 0.02 | 0.21 | 0.22 | 0.15 | 0.29 | 1.13 | 1.06 | 3.76 |
| InternLM-Math-20B (Ying et al., 2024b) | 0.02 | 0.03 | 0.03 | 0.03 | 0.02 | 0.03 | 0.24 | 0.72 | 2.39 |
| Mistral-Instruct-v0.3-7B (Jiang et al., 2023b) | 0.30 | 0.23 | 1.90 | 0.48 | 0.80 | 1.86 | 0.33 | 0.53 | 1.96 |

For the generation of results using the "**greedy**" strategy, we set the temperature parameter to 0.0 and 0.7 for the "**pass@k**" strategy. To present unbiased results for "pass@k", we follow the calculation method outlined in (Chen et al., 2021). Specifically, we generate $n = 20$ samples for each instance, evaluate the number of correct samples passing unit tests, and then calculate the unbiased estimator for pass@k. We repeat the experiments 5 times and report the 95% confidence intervals with a precision of ±0.1 to account for variability in the results.

## P.3 DETAILED ANALYSES OF EXISTING LLMS ON FORML4

The emergence of LLMs has fostered advancements in autoformalization tasks, where natural language descriptions are converted into formal, programmable constructs. In this analysis, we examine how various LLMs, benchmarking them across three different tests: Random, Basic, and Real proposed by FORML4.

As shown in Table 15, there is a distinguishing performance divide between closed-source and open-source LLMs. Closed-source models like GPT-4 and GPT-3.5 display substantially higher Greedy and Pass@k scores across all tests compared to open-source LLMs. For instance, GPT-4 achieves a Greedy score of 10.20% in the Real Test, whereas the highest corresponding score for an open-source model (InternLM-Math-7B) is only 1.10%. Focusing on open-source LLMs, DeepSeek-Math-Instruct-7B stands out, particularly in the Random Test with a Greedy score of 0.58% and a Pass@5 score of 1.71%. This model's performance suggests a basic understanding of Lean 4 formalizations, even though it falls behind the scores of closed-source LLMs.

On the other end of the spectrum, LLEMMA-7B and LLEMMA-34B models display negligible results in the Real Test. Their zero scores across all three metrics suggest that these models may not have effectively integrated Lean 4 formalization capabilities into their architectures or training data.

Finally, size seems to play a less significant role in autoformalization tasks, as evidenced by consistently low scores across models of varying sizes, from 7B to 34B parameters. This indicates that simply increasing the model size doesn't necessarily lead to better performance in specialized tasks such as autoformalization in Lean 4.

Despite the progress made by both open-source and closed-source LLMs in the area of autoformalization, our analysis identifies a consistent need for enhancement across the board. While certain closed-source models demonstrate superior performance, the opportunity for improvement remains vast, particularly within the open-source domain. We, therefore, propose FORML4 encompassing both training and testing sets tailored for evaluating and improving autoformalization capabilities.

## Q    DATASET QUALITY AND HUMAN VERIFICATION

The human verification process for FormL4 achieved an average success rate of 0.72, situating it within the context of existing LLM-constructed autoformalization datasets. Table 16 presents a comprehensive comparison of relevant datasets, highlighting key characteristics across different formalization efforts.

Table 16: Comparison of Autoformalization Datasets

| Dataset | Source Lang. | Size | Human Verif. | Accuracy | Verif. Rate (%) |
|---|---|---|---|---|---|
| ProofNet | Lean 3 | 371 | ✓ | 1.00 | 100.00 |
| MiniF2F | Multi | 488 | ✓ | 1.00 | 100.00 |
| FIMO | Lean 3 | 149 | ✓ | 0.61 | 100.00 |
| Lean Workbook | Lean 4 | 57k | ✓ | 0.94 | 0.10 |
| MMA | Lean 4 | 332K | ✗ | – | 0.00 |
| **FormL4** | Lean 4 | 17k | ✓ | 0.72 | 0.30 |

To systematically evaluate our dataset quality, we conducted a comparative verification study across FormL4, MMA, and Lean Workbook. We randomly sampled 90 pairs of natural language and formal language statements (30 samples each from FormL4, MMA, and Lean Workbook), shuffled them together, and assigned them to five Lean 4 experts. The assignments were split so that (1) each sample was verified by two different human experts for robust evaluation; (2) each expert verified an even distribution of samples from the three datasets in order to rule out the factor of individual bias. The experts follow the original verification task to evaluate whether the natural language and formal statement are perfectly aligned. Since each sample is dual-annotated, disagreements are resolved with annotator discussion for a final verdict.

In addition, we observe a large discrepancy in statement complexity between Lean Workbook and FormL4/MMA, as demonstrated in the examples inTable 17. Therefore, our experts also evaluated a new item called autoformalization difficulty from the natural language statement into its corresponding formal statement. The difficulty levels were categorized according to the criteria shown in Table 18.

Table 17: Statement Examples from Lean Workbook, MMA, and `FormL4`

| Source | Natural Language Statement | Formal Statement |
|---|---|---|
| Lean Workbook | For $a, b, c \in \mathbb{R}$ such that $a+b+c = 1$, prove that $ab(3a-1) + ac(3b-1) + bc(3c-1) \geq 0$. | `theorem lean_workbook_plus_14251 : ∀ a b c : ℝ , a + b + c = 1 → a * b * (3 * a - 1) + a * c * (3 * b - 1) + b * c * (3 * c - 1) ≥ 0 :=` |
| MMA | For a summable function $f$ and a constant $a$ from a topological space $M$ that is also a $T_2$ space (Hausdorff), the infinite sum of $f(z) \cdot a$ equals the product of $\sum f(z)$ and $a$. | `theorem tsum_smul_const [T2Space M] (hf : Summable f) (a :  M) : ( ∑′ z, f z • a) = (∑′ z, f z) • a :=` |
| FormL4 | Prove that in category theory, if there exists a kernel pair for a monomorphism $f$ with $a$ as its domain, then $a$ is an isomorphism. | `theorem isIso_of_mono (h : IsKernelPair f a b) [Mono f] :  IsIso a :=` |

Table 19 summarizes the verification study findings, including the distribution of difficulty levels across datasets.

Our verification study reveals several significant findings:

Table 18: Autoformalization Difficulty Level Criteria

| Level | Description |
|---|---|
| Simple (S) | The statement is primarily numeric or equation-based, requiring minimal knowledge of Lean 4 syntax or semantics to translate. |
| Medium (M) | The statement involves mathematical concepts or relations written in natural language, and its autoformalization requires an understanding of common Lean 4 constructs. |
| Advanced (A) | The statement incorporates advanced mathematical concepts (college level or above), and the Lean 4 syntax or lemma required for translation is also advanced. |

Table 19: Cross-Dataset Verification Results

| Dataset | Aligned Ratio | Disagreement | Difficulty Distribution |
|---|---|---|---|
| FormL4 | 73.33% | 26.67% | S:21.67%, M:61.67%, A:16.67% |
| MMA | 66.67% | 33.33% | S:13.33%, M:66.67%, A:20.00% |
| Lean Workbook | 63.33% | 36.67% | S:95.00%, M:5.00%, A:0.00% |

- Among LLM-constructed datasets, FormL4 demonstrates superior verification accuracy (73.33%), aligning with our initial verification result of 72%.

- The difficulty distribution analysis shows that FormL4 and MMA encompass more sophisticated statements (predominantly medium-level complexity), while Lean Workbook predominantly contains elementary statements.

- The observed verification disagreement rates underscore the inherent complexity in evaluating natural language to formal statement alignment, a recognized challenge in the field.

The notable disparity between our verification results and previous studies (e.g., Lean Workbook's reported 93.5% versus our observed 63.3%) can be attributed to two primary factors:

1. Implementation of more stringent verification criteria, mandating precise preservation of mathematical constraints and logical relationships

2. Systematic identification and handling of non-statement instances within source datasets

This comprehensive evaluation substantiates FormL4's quality while acknowledging the broader challenges in autoformalization assessment. The results validate both the effectiveness of our construction pipeline and underscore the necessity of rigorous verification protocols in dataset development.

