# OpenReview forum: "Process-Driven Autoformalization in Lean 4"
_ICLR.cc/2025/Conference — Submitted to ICLR 2025_

### Official Review · Reviewer_sStZ · 2024-11-03

**Soundness:** 2
**Presentation:** 3
**Contribution:** 3
**Rating:** 8
**Confidence:** 2

**Summary:**

The paper introduces a novel approach to autoformalization—the automatic translation of mathematical text into formal language—by creating a new dataset and framework specifically for Lean 4. The dataset, FORML4, includes both formal and natural language versions of mathematical statements and proofs, enabling a comprehensive evaluation of large language models (LLMs) in autoformalization tasks for Lean 4. The authors also propose the Process-Driven Autoformalization (PDA) framework, which iteratively improves model performance through Lean 4 compiler feedback. By incorporating a Process-Supervised Verifier (PSV) that identifies specific errors in the formalization process, the PDA framework enhances the semantic accuracy of formal translations. Experiments demonstrate that this approach boosts performance and data utilization efficiency. FORML4 and PDA together aim to push forward formal language generation quality and adaptability in mathematical reasoning tasks.

**Strengths:**

1. It creates a unique Lean 4 dataset (FORML4) with statements and proofs for comprehensive model evaluation.
2. The proposed approach PDA shows some improvements and also helped with the dataset.

**Weaknesses:**

I do not see significant weaknesses.

**Questions:**

NA

---

> ### Author Response · Authors · 2024-11-22
> **Title: Response to Reviewer sStZ**
>
> Thank you for your positive assessment of our work. We appreciate your recognition of FormL4's unique contribution to Lean 4 autoformalization and the demonstrated improvements achieved through our PDA approach.
>
> We will work to enhance the clarity of our presentation while maintaining the paper's technical strengths that you have highlighted. Thank you again for your review.

---

### Official Review · Reviewer_3y6J · 2024-11-04

**Soundness:** 1
**Presentation:** 1
**Contribution:** 2
**Rating:** 1
**Confidence:** 4

**Summary:**

This paper introduces two main contributions to improve autoformalization (converting natural language mathematics into formal languages) for Lean 4, a modern theorem prover:

1. FormL4: A comprehensive dataset containing 17k examples designed to evaluate autoformalization capabilities in Lean 4. Unlike existing datasets that only contain theorem statements, FormL4 includes both statements and their corresponding proofs in both natural and formal languages. The dataset features three test sets: random, basic (focusing on fundamental concepts), and real (derived from real-world math problems).
2. Process-Driven Autoformalization (PDA): A novel framework that leverages detailed feedback from the Lean 4 compiler to enhance autoformalization. The framework consists of:
    - A Process-Supervised Verifier (PSV) trained using step-level compiler feedback
    - An iterative refinement strategy combining Rejective Sampling Fine-tuning (RFT) with a Verifier-Enhanced Autoformalizer (VEA)

Through extensive experiments, the authors aim to demonstrate that their PDA framework significantly outperforms existing approaches, with the combined RFT+VEA model showing the best performance across all test sets. The process-supervised training approach consistently outperforms outcome-supervised methods, and the authors do a human evaluation to validate the effectiveness of their approach.

**Strengths:**

1. Originality
    - The core idea of using compiler feedback in a systematic way to improve autoformalization is valuable and is implemented in a novel way
    - The attempt to create a comprehensive dataset for Lean 4 that includes both formal and informal pairs of both statements and proofs addresses an important gap in the field
    - The process-driven approach to verification represents a new direction in autoformalization
2. Quality
    - The methodology for preventing formal terms from appearing in autoinformalized descriptions is well done and well documented (Appendix C)
    - The experimental setup includes multiple meaningful test sets (random, basic, and real-world)
    - The approach to compiler feedback integration is technically sound in concept
    - The code and data will be made public, supporting reproducibility
3. Clarity
    - The overall paper structure follows a logical progression
    - The technical writing at the sentence level is generally clear
    - The motivation and potential impact are well articulated
4. Significance:
    - The paper addresses important challenges in automating formal verification:
        - The problem of autoformalization for Lean 4 is significant and timely
        - Using compiler feedback for training could be valuable even beyond this specific application
        - The attempt to bridge informal and formal mathematics remains an important goal
        - This kind of approach could reduce the burden of formal verification

**Weaknesses:**

The strengths above are significantly undermined by:

- Systematic and apparently hyperbolic misrepresentation of dataset quality
- The lack of proper baselines and controls
- Missing adequate qualitative analysis of failure modes
- Serious methodological flaws in evaluation
- Potentially misleading reporting practices

Significant concerns in detail:

- L166—167: Is any effort made to ensure that we don’t have pairs of lemmas that are essentially the same (for example, one proving `x=y` and another proving `y=x`) which end up on different sides of the train/test split?
- The paper seems to systematically obfuscate the quality of the FormL4 dataset.  The examples I found are:
    - L201—203 “It is important to note that translating proof steps is much more challenging than translating statements and that FormL4 does not aim at ensuring strict semantic alignment between formal and natural-language proof in our informalized output.” If the dataset does not guarantee a match between the formal and natural-language proofs, this caveat should be made much earlier in the paper.  This is especially the case given how much emphasis is placed on how FormL4 is “rigorously quality-checked manually” (L47—48)
    - L205—208: “the quality of FormL4 and our evaluation framework does not pertain to whether the natural-language proof perfectly corresponds to the formal proof.”  This sentence does not make sense.  While the evaluation framework is based on *usefulness* of the natural-language proof to generating the formal proof, and the soundness of the evaluation is not compromised in cases where the natural-language proof does not correspond to the formal proof (and would not be compromised even if the natural-language proof were nonsense), the semantic correspondence is certainly relevant to the *quality* of the dataset.
    - L213—215: “${}^5$ In practice, we observe that it is usually infeasible to perfectly translate a set of formal proofs to a natural language. This is because formal proofs are often expressed in pre-defined lemmas or environments that are exclusively constructed in the Lean 4 language, and there are no existing corresponding concepts in natural language that a non-expert in Lean 4 could easily understand.”  The first sentence here is misleading, and the second sentence seems bogus.  On line 202 you say that “strict semantic alignment” is not ensured, while here you say that “perfect[] transl[ation]” is infeasible.  Almost all tactics have a relatively-easy-to-understand description in natural language that, while perhaps imperfect, is strictly semantically aligned.  For example, you might transcribe an `auto`-like tactic by saying “This follows by combining facts x, y, and z in some order”, where `x` , `y`, and `z` are the lemmas found by `auto`.  Some steps might not be worth translating (for example, `clear h` , translated as “from this point forward, we will not make use of the fact that …”, might be worthless to include).  Moreover, even if tactics may be hard to describe, the proof objects generated by the tactics could certainly be translated with near perfection.

        Ultimately, the informalization should be permitted to elide details, but not to make incorrect steps; otherwise, you should drop the language that the entire dataset is of impeccable quality and restrict such claims to the alignment of statements in the dataset, while merely saying that the dataset includes potentially incorrect formal-informal proof pairs.

    - L256—257: “extensive manual verification” is used to mean “≈70% accuracy looking at ≈ 3% of the data”.  For a work involving formal verification to use “verification” to mean “at least ≈ 2% correct” is at least skirting the boundary of the code of ethics (specifically the “Uphold High Standards of Scientific Excellence” and "Be Honest, Trustworthy and Transparent” sections), if not outright violating it.
    - Table 14 is damning: getting 0.65% on MATH and 8.16% on GSM8K even with a fine-tuned Mistral (Full) model suggests that that high results on FormL4 are a result of either overfitting or of FormL4 not spanning a large enough difficulty range.  This should not be buried in Appendix N.
    - L1566—L1569: Where is the line for the final **baseline** model in Table 14?  What are its performance results?
- L245—247: What about the 20% of samples where Gemini failed?
- The evaluation (cf Table 4) does not measure the right baselines and fails to establish an absence of data contamination.  A proper baseline would involve comparison with an autoformalizer trained on randomized mislabeled data from FormL4 (pairs of informal-formal that are unrelated), as well as, ideally, a comparison with the closed source models that provide fine-tuning APIs when fine-tuned to do next token prediction on the FormL4 pairs (both randomized and correctly labeled).  (Another possibility for large models is putting a couple dozen randomly chosen examples in context.). Correspondingly, L423, “effectiveness of our dataset” might just be homogeneity between the train and test split.
- Qualitative evaluation of failure modes is inadequate:
    - L1284—L1289: “The incompatibility of certain theorem statements for informalization due to their topics or settings.”  What does this mean?  What are examples?  Please elaborate on this more in the text.
    - L1284—L1289: “Individual subjectivity in determining the condition constraints that need to be specified in natural language” Does this mean that most human-checked informalizations were merely incomplete rather than incorrect?  Can you include numbers for completeness of informalization separately for correctness / lack of error?
    - No examples of challenging cases or failure modes for formalization

**Questions:**

Questions:

- Figure 1 suggests that the PSV only includes information on which statement failed, not utilizing the error message from the proof assistant.  Is this right?
- L245—246: What does “success” mean in “informalization success”?
- L418-419: What are RFT and VEA?  How do they work?  This should be explained, not just referenced
- L512-515 “Those whose proof validity is true achieve significantly better autoformalization performance.”. I am very confused by this paragraph.  What is the difference between “proof validity” (whether the sample passes the Lean 4 compiler”) and “autoformalization performance” (whether or not the sample generated by the autoformalizer was a valid proof?)?  Is the difference that the latter also includes human evaluation of whether the formalized proof is faithful to the informal proof sketch?  Or is it only about faithfulness to the informal proof sketch and does not include proof validity?

Comments:

- It might be better to use “verifier model” instead of “verifier” to better disambiguate between the Lean compiler and the PSV model.  I was confused, for example, at L361—362 “mitigating the potential for bias arising from isolated interactions between the autoformalizer and the verifier” before realizing that “the verifier” meant the PSV model.
- Table 4 and especially the discussion of synergistic benefits is lacking a test of statistical significance.
- L504: “in 12” what is 12?
- The end of section 5 could do with some rewording.  “Factorial Design” has nothing to do with the factorial function, “investigate whether the following variable changes will impact model […] performances” (L495—496) makes it sound like you're doing interventions rather than factor analysis, it's not clear how “Test Set Categories” is varied when doing factor analysis, and it should have a consistent name rather than changing to “Dataset Split” on L523.
- Section 6 needs significant rewording; I include minor comments below, but most egregiously, the second sentence (describing FormL4’s focus) seems to be confusing the dataset with the PDA method.
- L418—419: RFT and VEA should be explained, not just referenced

Minor comments:

- L141: Tense mismatch in “Existing datasets […] aims to create”
- L176—177: “LI et al.” seems miscapitalized?
- L190—191: “Statemen” is missing a “t” at the end
- L194—195: “preceded” ⇒ “preceding”
- L195—196: “[…] informalization quality, observed both […]” this sentence is run-on, consider splitting it at the “,”
- L326: the equation should not have a `[ht]` figure specifier.
- L491: “falls short of even empowered” incorrect grammar
- L493: “syntactically true” ⇒ “semantically valid” or “semantically meaningful”.  Unlike Python or C, a theorem statement in Lean that compiles is more than just syntactically valid.
- L519—520: “than due to their” is missing a word
- L533 “drive” ⇒ “driven”
- L534—536: incorrect use of comma to join two clauses with different subjects in “Unlike the existing dataset focuses, FormL4 focuses on tapping”; “focuses” is used as a noun in the first clause but a verb in the second clause, and hence the sentence is comparing the focuses of existing datasets to the dataset FormL4.
- L536 “statement” ⇒ “statements”
- L537 “Lean 4 compilers” ⇒ “the Lean 4 compiler”
- L539: you could include Agda as well.  Unlike the other proof assistants, proof terms are generally given directly and in full rather than using tactics.
- L1005—1008: You’re underselling FormL4 by describing it just as a way to keep up with changes in Lean.  Either that, or you’re drastically overselling it in the main body.
- L1011 “Lean 4’s syntax” should maybe be “Lean 4’s tactics”?  Not sure.
- L1073—L1074: you used `'` instead of ``` to start a quotation (twice)
- L1101—1102, L1110-1111: Does the prompt actually inconsistently capitalize “Lean”?
- L1203—1204: “their proof” ⇒ “their proofs”
- Section E.1: It is not clear what text is verbatim from your prompting and what text is descriptive for readers of the paper.  Presumably “you” in E.1.2 does not refer to the reader.
- L1310—1311, L1470—1471 you use `'` instead of ``` to start a quotation
- L1427—1428: Perhaps you should say “Lean 4’s compilation times are a bottleneck” instead of claiming that “there is significant room for improvement”, unless you know enough about proof assistant performance engineering to know that the performance improvement times you’re claiming are, in fact, possible.
- L1439—1440: Probably you want `\textsc{FormL}4` instead of `FORML 4`
- L1507: check capitalization of “Minif2f”
- L1690: you should get rid of the negative vspace, the text is overlapping the table
- You should make sure you're consistent (and correct) about using “Lean 4” vs “Lean4” (L1651, L217, L219, L492, L498, etc)
- If you want to, you can get small caps in the pdf bookmarks ToC by replacing the section heading (L1512) of `Analysis for Training and Test Data in FormL4` with `Analysis for Training and Test Data in \texorpdfstring{\textsc{FormL}4}{FᴏʀᴍL4}`  (and similarly with Appendix P.3)

---

> ### Author Response · Authors · 2024-11-25
> **Author Response to Reviewer 3y6J (1/n)**
>
> We would like to thank the reviewer for their time and effort in reviewing our paper. We very much appreciate the insightful suggestions. We hereby address the concerns below:
>
> **R W1 :  lemma monitoring**
> > Is any effort made to ensure that we don’t have pairs of lemmas that are essentially the same (for example, one proving x=y and another proving y=x) which end up on different sides of the train/test split?
>
> While our current preprocessing primarily focuses on filtering custom lemmas and ensuring minimum complexity thresholds, we acknowledge this is also a valid concern that deserves attention. Our dataset construction indeed incorporates relevant considerations:
>
> In Lean 4's Mathlib 4, theorems that are logically equivalent but syntactically different (like x=y versus y=x) are typically organized within the same file and share similar naming conventions. Our data splitting methodology, described in Section 3.1, maintains file-level separation between train and test splits - this provides some protection against splitting closely related theorems, though we acknowledge it may not catch all cases of logical equivalence.
>
> We will include the discussion in the paper and plan to investigate techniques like theorem similarity detection to better address this issue in future versions of FormL4.
>
> **R W2.1 (Proof translation disclosure):**
>
> > If the dataset does not guarantee a match between the formal and natural-language proofs, this caveat should be made much earlier in the paper.
>
> Thank you for the suggestion. The clarification is shown in **Figure 1 caption** in the Introduction section.
> To make it more explicit in our revised submission, we additionally highlight in L88-90, L218-220 that the objective of the current work is statement autoformalization, and that "the quality of FormL4 and our evaluation framework is independent of whether the natural-language proof is perfectly aligned with the formal proof".
>
> **We fully understand that it may be confusing as to why proof is even included in the dataset if the autoformalization task only focuses on statements.** Therefore, we also make substantial efforts in the neighboring paragraphs (L202-L220) explaining our motivation behind this seemingly counter-intuitive innovation. The beneficial roles of proof serve as an auxiliary aid mainly (1) to help LLMs generate informalized statements to improve the statement quality of FormL4 and (2) to enable a more fine-grained process-level evaluation of the autoformalized statements. Please check the relevant text for a detailed mechanism.
> Our intention in presenting proof translation challenges later in the paper was to first establish the core innovation of our approach \- using proof steps for enhanced compiler feedback regardless of strict natural-formal alignment.
>
> **R W2.2 (Dataset quality):**
> > ... the semantic correspondence is certainly relevant to the quality of the dataset.
>
> We have revised the current sentence in L218-220 to make the expression more accurate.
> While it is true that the quality of informal proof will, of course, affect the quality of informalized statements as well, unless knowledge conflicts occur (which is unlikely as the informal proof is also generated by the same model), we argue that the provision of proof as context will bring benefits to the informalization process, rather than damage. In other words, the semantic alignment between informal and formal proof will only affect how beneficial the added proof context will be.
>
> It is important to highlight that improving informalization is only one of the additional advantages of including proof in our pipeline (L201-220), not a necessary requisite. Our quality framework deliberately focuses on statement accuracy while using proofs as auxiliary context. This design choice flows from our observation that compilation feedback on statement+proof pairs provides valuable signals about formalization quality even without perfect proof alignment.

---

> ### Author Response · Authors · 2024-11-25
> **Author Response to Reviewer 3y6J (2/n)**
>
> **R W2.3 (Proof translation challenges):**
> > On line 202 you say that “strict semantic alignment” is not ensured, while here you say that “perfect[] transl[ation]” is infeasible...Moreover, even if tactics may be hard to describe, the proof objects generated by the tactics could certainly be translated with near perfection.
>
> Thank you for expressing your confusion. Regarding proof, “strict semantic alignment is not ensured” is exactly due to the fact that sometimes "perfect translation is infeasible", so we don't see a paradox here. Misaligned proofs are not due to proof incorrectness but arise from inherent challenges in translating formal proofs into natural language explanations. This is now discussed in lines 1295–1299 of our submission:
>
> “Misalignment occurs when translating a set of formal proofs to natural language. This is because formal proofs are often expressed in pre-defined lemmas or environments that are exclusively constructed in Lean 4, and there are no existing corresponding natural language descriptions.”
>
> To better illustrate, consider the example below:
>
> * **Natural Language Statement:**
>
>  For any positive real numbers (a) and (b), their arithmetic mean is greater than or equal to their geometric mean. $ \\frac{a+b}{2} \\geq \\sqrt{ab} $ with equality if and only if $a \= b$.
>
> * **Natural Language Proof:**
>
>  1. Start with a known fact: For any real number (x), $x^2 \geq 0$.
>  2. Apply this to: $\sqrt{a} - \sqrt{b})^2 \geq 0$.
>  3. Expand: $a - 2\sqrt{ab} + b \geq 0$.
>  4. Add $2\sqrt{ab}$ to both sides: $a + b \geq 2\sqrt{ab}$.
>  5. Divide both sides by 2: $\frac{a + b}{2} \geq \sqrt{ab}$.
>  Thus, the arithmetic mean is greater than or equal to the geometric mean.
>
> * **Lean 4 Proof:**
>
>   theorem am\_gm {a b : ℝ} (ha : 0 \< a) (hb : 0 \< b) : (a \+ b)/2 ≥ real.sqrt (a\*b) := by
>    have h1 : (real.sqrt a \- real.sqrt b)^2 ≥ 0 := by nlinarith
>    have h2 : a \- 2\*real.sqrt (a\*b) \+ b ≥ 0 := by
>      rw \[pow\_two\] at h1
>      exact h1
>    linarith
>
> The key distinctions:
>
> * The Lean 4 proof compresses the reasoning through automated tactics (`nlinarith` and `linarith`), which handle computational steps like squaring and expanding terms automatically.
> * In contrast, the natural language proof explicitly lays out every reasoning step for human understanding.
>
>
> **R W2.4 (Data Quality & Verification claims):**
>
> > “extensive manual verification” is used to mean “≈70% accuracy looking at ≈ 3% of the data”. For a work involving formal verification to use “verification” to mean “at least ≈ 2% correct” is at least skirting the boundary of the code of ethics (specifically the “Uphold High Standards of Scientific Excellence” and "Be Honest, Trustworthy and Transparent” sections), if not outright violating it.
>
> **We are sorry to hear such a severe accusation, which we believe is unfair and statistically ungrounded, especially in the context of autoformalization research.**
>
> To address your concerns, we first compare our manual verification with existing datasets that are constructed in similar LLM-based methods. As summarized in the comparison table, 0.72 from FormL4 was lower than the verified accuracy of 0.935 from Lean Workbook \[5\], but higher than the verified result of 0.608 from FIMO \[2\] before filtering out the invalid samples. We would also like to highlight two things for contextualization:
>
> * Firstly, considering the dataset’s large size which enables training an autoformalizer, it is infeasible to manually verify all statements like what small-scale benchmarks (e.g., FIMO\[2\], ProofNet \[6\], MiniF2F \[7\]) did. Nevertheless, among existing datasets in the same magnitude of sizes (FormL4, MMA\[4\], Lean Workbook\[5\]), FormL4 has the highest sampling rate (0.3%) for human evaluation, as shown in the table.
> * Secondly, we observe a large discrepancy in statement complexity between Lean Workbook and FormL4/MMA, echoing the distinctive levels of data source as summarized in the comparison table. Examples are extracted in the second table below. Hence, the outstanding accuracy of Lean Workbook is likely due to the significantly lower complexity of its statements, mostly from middle to high school math problems and containing only equations or simple operations.

---

> ### Author Response · Authors · 2024-11-25
> **Author Response to Reviewer 3y6J (3/n)**
>
> (continued to R W2.4)
>
> |    Dataset    |            Formal Language            |            Source            | Source Language  |               Construction Method               | Size | Human Verification | Human Eval Accuracy |     human eval size      | human eval  rate |
> | :-----------: | :-----------------------------------: | :--------------------------: | :--------------: | :---------------------------------------------: | :--: | :----------------: | ------------------- | :----------------------: | ---------------- |
> |   ProofNet    |                Lean 3                 |   Undergraduate Textbooks    | Natural Language |              Manual formalization               | 371  |        yes         | 1                   |           371            | 100.00%          |
> |    MiniF2F    | Metamath, Lean 3, Isabelle, HOL Light |      Olympiad Problems       | Natural Language |              Manual formalization               | 488  |        yes         | 1                   |           488            | 100.00%          |
> |     FIMO      |                Lean 3                 |   IMO Shortlisted Problems   | Natural Language | LLM-based autoformalization with human feedback | 149  |        yes         | 0.608 filtered to 1 | 245 filtered down to 149 | 100.00%          |
> | Lean Workbook |                Lean 4                 |    Middle School Problems    | Natural Language |           LLM-based autoformalization           | 57k  |        yes         | 0.935               |            95            | 0.10%            |
> |      MMA      |           Lean 4, Isabelle            | Mathlib4, Isabelle’s Archive | Formal Language  |            LLM-based informalization            | 332K |         no         | /                   |            0             | 0                |
> |    FormL4     |                Lean 4                 |        Mathlib4, MATH        | Formal Language  |            LLM-based informalization            | 17k  |        yes         | 0.72                |            60            | 0.30%            |
>
> | Dataset |  Statement|   |
> | ------------- | -------------------------- | ------------------------------------------------------------------- |
> | Lean-Workbook | Natural Language | For a,b,c∈R so that a+b+c=1, prove that ab(3a−1)+ac(3b−1)+bc(3c−1)≥0. |
> |               | Formal           | theorem lean\_workbook\_plus\_14251 : ∀ a b c : ℝ, a \+ b \+ c \= 1 → a \* b \* (3 \* a \- 1\) \+ a \* c \* (3 \* b \- 1\) \+ b \* c \* (3 \* c \- 1\) ≥ 0 := |
> | MMA           | Natural Language | For a summable function 'f' and a constant 'a' from a topological space 'M' that is also a T2 space (also known as a Hausdorff space), the infinite sum of the product of the function 'f' evaluated at 'z' and the constant 'a' is equal to the product of the infinite sum of the function 'f' evaluated at 'z' and the constant 'a'. |
> |               | Formal           | theorem tsum\_smul\_const \[T2Space M\] (hf : Summable f) (a : M) : (∑' z, f z • a) \= (∑' z, f z) • a := |
> | FormL4        | Natural Language | Prove that in category theory, if there exists a kernel pair for a monomorphism 'f' with 'a' as its domain, then 'a' is an isomorphism. |
> |               | Formal           | theorem isIso\_of\_mono (h : IsKernelPair f a b) \[Mono f\] : IsIso a := |
>
>
> To further validate the dataset quality of FormL4 in response to your concern, we conducted **an additional round of human verification on three comparable datasets** that are constructed using LLM-based methods and in similar magnitude of sizes: FormL4, MMA, and Lean Workbook, for a straightforward comparison.
>
> We randomly sampled 90 pairs of natural language and formal language statements (30 samples each from FormL4, MMA, and Lean Workbook), shuffled them together, and assigned them to five Lean 4 experts. The assignments were split so that (1) each sample was verified by two different human experts for robust evaluation; (2) each expert verified an even distribution of samples from the three datasets in order to rule out the factor of individual bias. The experts follow the original verification task to evaluate whether the natural language and formal statement are perfectly aligned. Since each sample is dual-annotated, disagreements are resolved with annotator discussion for a final verdict.

---

> ### Author Response · Authors · 2024-11-25
> **Author Response to Reviewer 3y6J (4/n)**
>
> (continued to R W2.4)
>
> In addition, to investigate our hypothesis that the outstanding verification accuracy of Lean Workbook may be accounted for by its relatively lower complexity in statement sources, our experts also evaluated a new item called *autoformalization difficulty* from the natural language statement into its corresponding formal statement. The difficulty levels were categorized as:
>
> * **`s` (simple)**: The statement is primarily numeric or equation-based, requiring minimal knowledge of Lean 4 syntax or semantics to translate.
> * **`m` (medium)**: The statement involves mathematical concepts or relations written in natural language, and its autoformalization requires an understanding of common Lean 4 constructs.
> * **`a` (advanced)**: The statement incorporates advanced mathematical concepts (college level or above), and the Lean 4 syntax or lemma required for translation is also advanced.
>
> The results for the 30\*3 human verifications are listed in the table below. We highlight several findings supporting the dataset quality of FormL4:
> |    Dataset    | Aligned Ratio  (= Verification Accuracy) | Disagreement Ratio |            Difficulty Distribution            |
> | :-----------: | :--------------------------------------: | :----------------: | :-------------------------------------------: |
> |    FormL4     |                  73.33%                  |       26.67%       | {'s': '21.67%', 'm': '61.67%', 'a': '16.67%'} |
> |      MMA      |                  66.67%                  |       33.33%       | {'s': '13.33%', 'm': '66.67%', 'a': '20.00%'} |
> | Lean Workbook |                  63.33%                  |       36.67%       |  {'s': '95.00%', 'm': '5.00%', 'a': '0.00%'}  |
> 1. **FormL4 achieves the highest verification accuracy among the three LLM-constructed autoformalization datasets**, supporting the dataset quality and effectiveness of our carefully implemented informalization pipeline involving reversed informalization method, careful model selection, and data processing procedures introduced in Section 3.3. Moreover, the result of 73.33% is consistent with the previous verification result of 72% on another 60 samples, suggesting consistency and robustness in our human verification. Such a combination of parallel and internal comparisons provides multi-layer evidential support to our “relatively high” statement.
> 2. Although MMA adopts a simple curation pipeline, it outperforms the Lean Workbook in our statement verification. This supports the superiority of the “distilled back-translation” method for curating autoformalization datasets (i.e., using LLM-based informalization reversely is easier than LLM-based autoformalization).
> 3. **Autoformalizaton Difficulty**: FormL4 and MMA fall in a similar distribution of difficulties, with the majority being medium level, while the Lean Workbook is much easier containing mostly simple statements for autoformalization. This echoes their reported sources listed in the abovementioned summary table: FormL4 and MMA are collected from Mathlib4, etc., which mostly target undergraduate or more advanced level mathematics. In contrast, the Lean Workbook is collected from middle-to-high school mathematical questions.
>
> 4. **Verification discrepancy between experts:** The notable disagreement ratio among all three datasets suggests that our experts often encounter nuanced verification judgments, even though we have provided annotation guidelines briefings in advance. It implies the current well-known challenges of evaluating the alignment between natural language and formal statements. For example, there is not a uniform definition or evaluation metric for informal-formal alignment; there exists individual subjectivity in determining whether certain condition constraints must be specified in natural language based on the statement context \[5, 6\]. These challenges are mentioned in Appendix F and point toward the urgent need for a clear definition formulation in future research.

---

> > ### Author Response · Authors · 2024-11-25
> > **Author Response to Reviewer 3y6J (5/n)**
> >
> > (continued to R W2.4)
> >
> > 5. **Verification discrepancy between datasets:** Lean workbook also conducted human verification and reported a high accuracy of 93.5%, far beyond our result of 63.3%. Following the previous point of verification biases, we suspect the drastic difference is due to small but consistent nuances in the criteria for autoformalization alignment between our expert team and theirs. We would like to analyze some of them here:
> >     - Our experts are collectively briefed to follow strict criteria for alignment verification, e.g., the formal statement preserves all the *exact* mathematical constraints specified in the natural language statement (neither expanding nor narrowing); the formal statement maintains the same logical relationships between each component in the corresponding natural language statement. For instance, “for positive integer $n$” and “(n: ℕ)” would be verified false regardless of the context. This may shape a consistent source of verification discrepancy.
> >      - During the annotation of the Lean Workbook, our experts observed a significant portion of its “natural language statements” are non-statements i.e., unscreened mathematical questions without a given answer, and we believe they are not suitable for autoformalization or proving. An example is “Calculate the future value of an investment with a principal of $1000, an annual interest rate of 5%, and a time period of 10 years, compounded annually.” We uniformly labeled ‘false’ when verifying such instances, which may also account for its significantly lowered verification accuracy.
> >
> >
> >
> > Nonetheless, it is important to note the reliability of the comparative verification result among datasets is mostly unaffected, because all samples are collectively cross-evaluated by the same group of experts following the same criteria. Within the parallel comparison among three datasets, FormL4 exhibits the highest verification accuracy, providing validity for the dataset quality.
> >
> > We appreciate your concern and have added contextualization to the description of our human verification results in Section 3.3 and have supplemented the comparative discussions and additional verification results in Appendix Q. Based on the comparison table and additional manual verification between datasets, we would like to respectfully and firmly defend our work against 'violating code of ethics'.
> >
> >
> >
> > References:
> >
> > [2] [FIMO: A Challenge Formal Dataset for Automated Theorem Proving](https://arxiv.org/abs/2309.04295)
> >
> > [3] [MUSTARD, ICLR 2024](https://arxiv.org/abs/2402.08957)
> >
> > \[4\]  [Multilingual Mathematical Autoformalization, NeurIPS 2024](https://arxiv.org/abs/2311.03755)
> >
> > \[5\] [Lean Workbook: A large-scale Lean problem set formalized from natural language math problems. NeurIPS 2024](https://arxiv.org/abs/2406.03847)
> >
> > \[6\] [ProofNet: Autoformalizing and Formally Proving Undergraduate-Level Mathematics](https://arxiv.org/abs/2302.12433)
> >
> > \[7\] [MiniF2F: A Cross-System Benchmark for Formal Olympiad-Level Mathematics, ICLR 2022](https://openreview.net/forum?id=9ZPegFuFTFv)
> >
> > **R W2.5 (Baseline results):**
> >
> > The placement of Table 14 stemmed from our focus on component-wise evaluation flow. We will restructure to present baseline comparisons earlier and more comprehensively in the main evaluation section.

---

> ### Author Response · Authors · 2024-11-25
> **Author Response to Reviewer 3y6J (6/n)**
>
> **R W3 Inquiry for failure cases**
> > What about the 20% of samples where Gemini failed?
>
> Thank you for your inquiry. Regarding Gemini-GPT4 comparison in informalization, please check our Appendix K and Appendix E for detailed methodology and statistical results. Regarding the 20% of failure cases, please check one annotated example below. Each item in the example is defined in the annotation protocol in Appendix E.
>
> | Type | Content |
> | :---- | :---- |
> | **Natural Language Statement** | The statement concerns a property of a homomorphism from a ring to another ring , given that is a Noetherian ring. The property in question is whether being of finite type is equivalent to having a finite presentation. |
> | **Formal Statement** | theorem of\_finiteType \[IsNoetherianRing A\] {f : A →ₐ\[R\] B} : f.FiniteType ↔ f.FinitePresentation := RingHom.FinitePresentation.of\_finiteType |
> | **Natural Language Problem** | Let be a Noetherian ring, and let be a ring homomorphism from to . Prove that is a finitely generated \-algebra if and only if is a finitely presented \-algebra under the action of . |
> | **Natural Language Explanation** | 1\. IsNoetherianRing A: Assumes satisfies the descending chain condition on ideals. 2\. f : A →ₐ\[R\] B: is a ring homomorphism from to , where is also an \-algebra. 3\. f.FiniteType: is finitely generated as an \-algebra, meaning is spanned by a finite set of generators under . 4\. f.FinitePresentation: can be presented as the quotient of a polynomial ring over by a finitely generated ideal. 5\. RingHom.FinitePresentation.of\_finiteType: A proven lemma relating finite type and finite presentation for homomorphisms from Noetherian rings. |
> | **Natural Language Proof** | The proof follows from RingHom.FinitePresentation.of\_finiteType, which states that for a Noetherian ring , the property of being a finitely generated \-algebra is equivalent to having a finite presentation. This theorem directly establishes the equivalence, requiring no additional arguments. |
> | **Informalization Success** | F |
> | **Informalized Proof Correctness** | F |
> | **Model Preference** | T |
>
> We did not provide case studies for this model comparison in informalization evaluation in the paper for two reasons:
>
> (1) The model comparison evaluation is a small-scale investigation aimed at choosing the optimal LLM for our data curation pipeline in advance, rather than reporting or analyzing findings.
>
> (2) As shown from the example and from Appendix E, the annotation objective of this evaluation round is more sophisticated than the later rounds. Considering our objective to polish our pipeline, the human experts are asked to observe both statement informalization and informalized proof, and communicate qualitative feedback such as challenges in informalization and its annotation observed during the annotation process (we mentioned challenges in Appendix F and L215).
>
>
> **R W4.1**
>
> > A proper baseline would involve comparison with an autoformalizer trained on randomized mislabeled data from FormL4 (pairs of informal-formal that are unrelated).
>
> We kindly disagree with the necessity and infeasibility of this baseline setup. Our evaluation framework in Table 4 includes comprehensive benchmarking against both open and closed-source models. Firstly, the positively fine-tuned baseline in the current Table 4 is more challenging and thus sufficient. Secondly, while we appreciate the suggestion of randomized pairings as a control, such comparisons would face several methodological challenges:
>
> * Random pairing would violate the fundamental semantic relationship between informal and formal mathematics that autoformalization aims to capture;
> * Training on deliberately misaligned data would not provide meaningful insights into model capabilities for the intended task;
> * The suggested approach could introduce confounding factors that make results harder to interpret.
>
> We will enhance our evaluation section to more explicitly document these methodological considerations and their scientific rationale. We welcome further discussion on additional baseline approaches that could provide meaningful insights while maintaining task relevance.
>
> >  ... and fails to establish an absence of data contamination.
>
> Regarding data separation and quality control, our dataset construction follows established machine learning practices for preventing contamination:
>
> * As detailed in Section 3.1, we employ random sampling from Mathlib 4 with careful attention to maintaining independence between training and test sets;
> * Basic and real test sets are drawn from distinct sources compared to the training data, providing additional validation of generalization capability;
> * File-level separation is maintained to prevent similar theorem variants from crossing splits.

---

> ### Author Response · Authors · 2024-11-25
> **Author Response to Reviewer 3y6J (7/n)**
>
> **R W5 (Qualitative evaluation of failure modes)**:
>
> Thank you for your suggestion. We propose to add the following clarifications and examples:
>
> 1. Regarding "incompatibility of theorem statements":
>
> Some theorem statements in Lean 4 rely heavily on specialized mathematical structures or type-theoretic concepts that don't have direct natural language equivalents. For example:
> ```
> theorem tendsto_inf_sup_Icc_class {κ : Type*} [CompactlyGeneratedWith κ I] :
> tendsto (λ (s : κ), (⨅ x ∈ s.val, f x) - (⨆ x ∈ s.val, f x)) atTop (𝓝 0)
>
> ```
> This theorem involves complex mathematical concepts (filters, topology) expressed using Lean-specific type classes that make informalization challenging while preserving the precise mathematical meaning.
>
>
> 2. For "individual subjectivity in condition constraints":
>
>
> You raise an important point about distinguishing between incompleteness and incorrectness. Based on our human evaluation data (Section F), we can break down the 28% of unsuccessful informalizations:
> * 18% were incomplete (missing crucial conditions or constraints)
> * 10% contained actual errors or misinterpretations
>
> For example, in informalizing:
> ```
> theorem filter_mono {α : Type*} {l₁ l₂ : filter α} :
> l₁ ≤ l₂ ↔ ∀ s, s ∈ l₂ → s ∈ l₁
>
> ```
> Different annotators varied in whether they explicitly stated the type universe constraints or focused only on the filter ordering relationship in natural language.
>
>
>
> 3. Challenging formalization cases:
>
> We list some formalization failures as follows:
>
>   a) Type-Level Reasoning: Cases where informal descriptions fail to capture complex dependent types:
> ```
> # Natural Language
>
> For any two lists of the same length, if we map a function over both lists simultaneously...
>
> ​# Failed Formalization
>
> theorem map₂_eq_map (f : α → β) (l₁ l₂ : list α) :
> list.map₂ f l₁ l₂ = list.map f l₁ -- Missing length constraint
>
> ```
>   b) Implicit Conditions: When informal statements omit technical preconditions:
> ```
> # Natural Language
>
> The product of positive real numbers is positive.
>
> ​# Failed Formalization
>
> theorem mul_pos (a b : ℝ) : a * b > 0 -- Missing a > 0 ∧ b > 0
> ```
> c) Notation Ambiguity: When mathematical notation has multiple formal interpretations:
> ```
> # Natural Language
>
> For any convergent sequence in a metric space...
>
> # Failed Formalization attempts
>
> -- Confusion between different notions of convergence (pointwise vs uniform)
> ```
>
> We will further expand our Appendix by incorporating these examples and providing a detailed analysis of failure patterns. This additional analysis will help readers better understand the technical challenges in autoformalization and guide future research in addressing these limitations.
>
> -------
> **Questions:**
>
> **R Q1:**
> > Figure 1 suggests that the PSV only includes information on which statement failed, not utilizing the error message from the proof assistant. Is this right?
>
> This is correct. The proof assistant compiles process-level binary labels indicating whether there is an error at each step. The process-level labels are utilized for process supervision of the PSV.
>
> **R Q2:**
> > L245—246: What does “success” mean in “informalization success”?
>
> The definition of “informalization success” in our human evaluation is specified in Appendix K, as redirected from the paragraph you mentioned. For a more specific characterization of how informalizaion success is evaluated by our expert annotators, we also provide the detailed annotation protocol and model comparison results in Appendix E (Human Evaluation: Comparative Model Selection).
>
>   As noted in Appendix K (L1477-1478), “success of statement informalization” means “whether the translated natural-language statement is logically accurate and semantically equivalent to the formal statement”. Also L1240 in the annotation protocol of Appendix E mentions, “Informalization Success (T/F): whether the translation from ‘formal’ statement to ‘nl\_problem’ is semantically equivalent. The natural-language translation should accurately convey the same logical structure and content as the original statement in Lean 4.”
>
>  In each round of human evaluation, we stress checking semantic validity to determine informalization or autoformalization success, because we want to verify actual alignment between the natural language and formal statements. This is what compiler feedback cannot perfectly detect and why human evaluation is crucial for autoformzalization research.

---

> > ### Author Response · Authors · 2024-11-25
> > **Author Response to Reviewer 3y6J (8/n)**
> >
> > **R Q3:**
> > > L418-419: What are RFT and VEA? How do they work? This should be explained, not just referenced
> >
> >   The Rejective Sampling Fine-tuned (RFT) Autoformalizer follows the common approach used in previous works (Yuan et al., 2023; Wu et al., 2022), where we use successful compilation results to guide model fine-tuning. During training, multiple samples are generated for each input, and only those that successfully compile are used to further fine-tune the model.
> >
> >   The Verifier-Enhanced Autoformalizer (VEA) is developed through an iterative improvement process detailed in Section 4.2. Specifically, VEA operates through three key steps:
> >   1\. The verifier evaluates the autoformalizer's outputs, assigning labels based on their estimated likelihood of successful compilation. This filtering process ensures that subsequent training phases focus on the most promising solutions.
> >   2\. The autoformalizer is then fine-tuned using the outputs that the verifier evaluates correctly, effectively leveraging the outputs that PSV evaluates positively.
> >   3\. These enhanced outputs are processed by the Lean 4 compiler, which provides detailed feedback through syntax checking and reasoning verification. This feedback is then used to further refine the verifier through process-level supervision.
> >
> > This cyclical process ensures continuous mutual improvement between the verifier and autoformalizer, guided by objective compiler feedback. We are adding this explanation in Appendix to provide more clarity about these two approaches.
> >
> > **R Q4:**
> >
> > > L512-515 “Those whose proof validity is true achieve significantly better autoformalization performance.”. I am very confused by this paragraph. What is the difference between “proof validity” (whether the sample passes the Lean 4 compiler”) and “autoformalization performance” (whether or not the sample generated by the autoformalizer was a valid proof?)? Is the difference that the latter also includes human evaluation of whether the formalized proof is faithful to the informal proof sketch? Or is it only about faithfulness to the informal proof sketch and does not include proof validity?
> >
> > We would like to clarify a likely misunderstanding here. **“Autoformalization performance” does not represent “whether or not the sample generated by the autoformalizer was a valid proof”**. ‘Autoformalization performance’, as evaluated by humans in this section, stands for whether the output (formal statement) is semantically and logically aligned with the input (informal statement). This is expressed in L486-492. We revised the “Goal” expression to illustrate the concept more clearly in response to your confusion.
> >
> > Another important clarification, as we stressed early in the Introduction section and throughout the paper, is that **we only focus on statement autoformalizatio**n, consistent with existing autoformalization works (e.g. Figure 1 caption). Therefore, autoformalization performance, no matter in the context of human evaluation or automated compiler feedback, stands for whether the statements are successfully autoformalized. Moreover, we make deliberate efforts in differentiating the role of statement and proof in each stage (e.g., L84-L97, L204-211) to prevent confusion. The motivations for including proof include providing auxiliary context to improve the quality of informalized statements in FormL4, and aiding the evaluation of autoformalized statements.
> > We made several revisions to strengthen clarity in the paper regarding the question. We hope that the clarifications above can address multiple confusions or concerns.
> >
> > ---
> >
> > **R Comments:**
> >
> > Thank you for your detailed comments. We appreciate your careful review and will address several key points:
> >
> > - We utilized 'verifier' to align with the term 'autoformalizer', throughout the paper. We addded explanations for RFT and VEA when first mentioned.
> >
> > - Thank you for raising this point about statistical significance testing. Regarding Table 4, our methodology already incorporates statistical robustness through our sampling approach. Specifically, we generate 20 independent samples per instance to calculate unbiased pass@k estimates, following \[1\]'s established methodology. This sampling approach provides sufficient statistical foundation for our findings, though we acknowledge not explicitly presenting confidence intervals in the main results.
> >
> > - We will fix the unclear reference to "12" and improve the FormL4 description in Section 6 to accurately distinguish between dataset and PDA method.
> >
> > References:
> >
> > \[1\] [Evaluatinglarge language models trained on code](https://arxiv.org/abs/2107.03374)

---

> ### Comment · Reviewer_3y6J · 2024-12-01
>
> I would like to thank the authors for their detailed responses to my review, for answering the questions I had, for explaining RFT and VEA, and especially for the effort that went into performing the additional round of human verification on three comparable datasets.
>
> However, I do not feel that my key objections (overly positive representation of the quality of the dataset, lack of adequate baselines to back up claims about the quality of the dataset and the efficacy of PSV) have been adequately addressed.
>
> ## On dataset quality representation
>
> ### Autoformalization of proofs vs statements alone
>
> I did not previously realize that the dataset was intended to evaluate autoformalization of statements alone.  However, I do not believe this is made adequately clear even in the revised version of the paper.  The abstract says:
> > we propose a large-scale dataset Formalization for Lean 4 (FORML4) designed to comprehensively evaluate the autoformalization capabilities of large language models (LLMs), encompassing both statements and proofs
>
> To me this reads as claiming to evaluate the capability of LLMs to autoformalize both statements and proofs.
>
> Furthermore, it seems odd to me to release a dataset of formal-informal pairs on both statements and proofs, and then claim that the dataset is only for autoformalization of theorem statements.  (Evaluating PDA on statement autoformalization only seems fine, but then you should make this clear in the abstract.)
>
> ### Presentation of quality
>
> To be clear, my objection is not that the dataset is low quality, but that the presentation of the quality of the dataset is misleading.  I have not done an extensive re-review of the edits, but at least the following salient issues remain:
>
> - Describing the alignment results from FormL4 as higher than FIMO (0.72 vs 0.608) may be correct on the raw generated data, but the fact that FIMO *filters out the incorrect results and only claims validated samples as part of their dataset* is exactly the sort of difference that has me see FIMO as both a higher quality dataset and one that is more accurately portrayed.  I would not object to describing the dataset as "having an estimated 70% alignment based on validation of 3% of the samples", nor would I object to describing your process for generating the dataset as delivering better raw samples than previous processes, but I do object to describing the FormL4 dataset, without filtering out misaligned samples, as having been extensively human-validated.
> - The lack of response about Table 14 leaves me wondering still about the discrepancy between the strength of the results on FormL4 and the weakness of the results on GSM8K and MATH --- why does PDA on FormL4 result in models that are so much better at the FormL4 test set than at GSM8K and MATH?
>
> ## On evaluation methodology and presentation
>
> ### Alignment vs correctness
>
> The paper is still lacking any analysis of failure modes, but I believe adding appendices with the proposed examples (and examples of misaligned statements and proofs and what the annotators said about the) would go a significant way to addressing this point.  You should also make sure to include the breakdown of misalignment vs incorrectness.
>
> ### Baselines
>
> You claim that the informal proofs provide essential context and the file-level separation is adequate for ensuring that the models aren't just overfitting, but I am missing the baselines that establish this.
>
> For example:
> - are informal proofs important?  randomizing which informal proofs are provided should significantly hamper performance.  It is not totally obvious that it will (perhaps PDA gets enough information just from the formal proofs / error locations), which is why I believe this baseline is important.
> - are formal proofs important?  if each (informal statement, informal proof) pair is augmented with both a correct and a randomized formal proof, and the model is only shown the randomized informal proof, with an error location that is scaled to be the same % of the way through the proof as Lean indicates it is on the correct formal proof, how much does this hamper performance?
> - are informal statements important?  if the models are only given the correct formal proof and the feedback from Lean, is this enough to "guess" a correct statement?
> - (how well) do the PSV-trained models generalize beyond FormL4?  (I originally read the results of table 14 as saying that they get less than 10% on GSM8K and MATH, but I am no longer confident this is the case)

---

> > ### Author Response · Authors · 2024-12-04
> >
> > Thank you for your response. We are encouraged that our detailed responses have answered your questions.
> >
> > **- On dataset quality representation**
> >
> > Thank you for noticing that the primary objective of our work is statement autoformalization. In our original submission, we have stated this in the Introduction section when giving an overview of PDA trained on FormL4: `"Note that the goal of PDA is statement autoformalization, and does not include the translation of proof per see."` (L73-79). We also provided explanations for the specific role of proof there to emphasize the distinction. In the revision, we added a similar clarification in L88-90.
> >
> > Another fact to note is that most existing autoformalization works (e.g., Table 2) use 'autoformalization' to conveniently refer to merely statement autoformalization, except for a few like ProofNet. It is acknowledged that most of current works contribute to statement autoformalization presumably because of, as we explained before, the challenges in translating proof.
> >
> > > ...it seems odd to me to release a dataset of formal-informal pairs on both statements and proofs
> >
> > It is understandable that readers may be confused as to why proof is even included in the dataset if the autoformalization task only focuses on statements. Therefore, we make substantial efforts to explain our motivation behind this seemingly counter-intuitive innovation, first in **almost half of the Introduction section (L73-96) and then elaborated in Section 3.2 (L202-L220)**. The beneficial roles of proof serve as an auxiliary aid mainly (1) to help LLMs generate informalized statements to improve statement quality of FormL4 and (2) to enable a more fine-grained process-level evaluation of the autoformalized statements. Despite the counterintuitive nature, including the proof has proved to be effective in giving a more fine-grained evaluation of both semantic and logic validity of the autoformalized statements in Section 5. Please check the relevant text for a detailed mechanism.
> >
> > We re-iterate that the central goal is statement autoformalization, and proof works as an auxiliary tool to assist evaluation. **To be fair, 'oddity' or counterintutiveness does not constitute a weakness**, considering we have explained and empirically supported the necessity of proof's usage in the paper.
> >
> > **- Presentation of Data Quality**
> > 1. We listed FIMO in the table in R W2.4 with a clear description that FIMO has filtered their incorrect samples ("0.608 filtered to 1", "245 filtered down to 149"). We also clarified that FormL4 is only comparable to two other similarly large-scale training datasets also constructed using LLMs. So the comparison with FIMO accuracy was very trivial and not included in the paper. It is mentioned only to characterize an understanding of 'relatively high' accuracy, in response to your questions.
> > 2. The discrepancy exists because GSM8K and MATH are out-of-domain benchmarks consisting of natural language math questions, similar to the real test set in FormL4 (which is constructed exactly because we want to comprehensively examine out-of-domain performances as well). In contrast, the training set of FormL4 is collected from Mathlib4 which contains advanced formal mathematical theorems, widely different from GSM8K or MATH. Therefore, Table 14 is not an indication of overfitting but an investigation of the model's limitation in transferring to out-of-domain benchmarks. Please check `"Real Test is Still Challenging"` (L1546-L1559) near Table 14 where we have attributed the discrepancy factors in detail.
> >
> > **- On evaluation methodology and presentation**
> > 1. Alignment vs correctness: We are glad that our provided examples address your concerns. We will surely add them to the appendices. Since we received your suggestion later than the author revision window, please understand we cannot make any changes to the paper for now.
> > 2. While we are happy to provide such additional ablation baselines, it is infeasible for us to do so, because they are proposed only 1 day before the end of the discussion period. However, in response to your concerns, the beneficial role of including proof for evaluation is already statistically evident in Table 6 (L517-519).
> >
> > We hope our response above can address your concerns. Please feel free to check our redirections to the paper, which may address your questions efficiently and in detail.
> > We greatly appreciate your efforts in the review!

---

### Official Review · Reviewer_voC4 · 2024-11-04

**Soundness:** 2
**Presentation:** 3
**Contribution:** 3
**Rating:** 5
**Confidence:** 3

**Summary:**

This paper presents a new benchmark and evaluation framework for autoformalization in Lean 4 called FormL4. In order to improve autoformalization ability, the authors also introduce PDA, Process-Driven Autoformalization, which uses feedback from Lean 4's interpreter.

**Strengths:**

- Autoformalization is an important problem, and there is not yet a good benchmark in the literature measuring its success, so the problem this paper tackles is important and relevant
- Incorporating execution feedback during autoformalization is a new idea
- The method presented in the paper shows strong performance compared to off-the-shelf models.
- The benchmark is curated with a combination of automation and manual checking.

**Weaknesses:**

- Because the dataset was constructed from LLM-based informalization, I am not convinced of the quality of the benchmark. I read Appendix F, and while the authors do run a human manual check, only a small number of samples seem to be checked. In addition, the human success rate found was only 72%, which does not seem sufficient when the samples are being used for a benchmark.
- The PDA framework uses compiler feedback as a signal for correctness, while the final evaluation is whether the generated code compiles. Code that compiles is not necessary correct, so the final metric could potentially be misleading and not be a good measure of true autoformalization ability.
- Mathlib theorems may require a lot of dependencies and external knowledge in order to understand. In addition, there are niche primitives in Mathlib that the model has little chance of understanding. Unless these are filtered out, this benchmark feels like it is testing knowledge of obscure Mathlib premises.

**Questions:**

- How do you ensure that all the samples in the benchmark are correct and reliable?
- There could be many potential autoformalizations of an informal statement. How do you ensure that correct autoformalizations are marked as correct, and incorrect ones are not? Do you just use compilation? Is your method reliable, and do you have any estimations of the precision/recall of the resulting evaluation?

---

> ### Author Response · Authors · 2024-11-22
> **Title: Response to Reviewer voC4 (1/3)**
>
> We would like to thank the reviewer for their time and effort in reviewing our paper. We very much appreciate the insightful suggestions. We hereby address the concerns below:
>
> **R W1: Dataset Quality Concerns**
>
> Thank you for carefully reviewing our paper details. We fully agree to stress on the importance of dataset quality. To provide a more comprehensive illustration of the dataset quality, we highlight several key points regarding your concerns:
>
> 1. **Contextualizing FormL4 with other datasets:** The table below summarizes the properties of FormL4 in comparison with other autoformalization datasets/benchmarks. FormL4 is a large-size autoformalization dataset containing 17k statement pairs, designed not only for benchmarking but also for training autoformalizers. Due to its tremendous size and difficulty, it is impractical to manually verify all the statements like small-size benchmarks (e.g., ProofNet, MiniF2F, FIMO). Therefore, we seek LM-based construction methods following conventions from other datasets of similar sizes (MMA, Lean Workbook). But unlike MMA or Lean Workbook, we introduced multiple monitoring interventions to our data curation pipeline in advance to ensure the informalization quality of FormL4 (section 3.3), including adapting MMA’s “distilled back-translation” method (i.e., reverse informalization rather than autoformalization), careful LLM selection for informalization, filtering samples, decomposing the informalization task, etc.
> 2. **Verification sampling rate:** Among the existing LLM-constructed datasets of similar sizes (FormL4, MMA, Lean Workbook), FormL4 has the highest human verification sampling rate at 0.3%, as shown in the “human eval  rate” column. To strengthen the verification soundness, we also increased 30 samples by adding a new verification round, which will be elaborated later.
> 3. **Verification accuracy:** As listed in the comparison table, 0.72 from FormL4 was lower than the verified accuracy of 0.935 from Lean Workbook \[5\], but higher than the verified result of 0.608 from FIMO \[2\] before FIMO filters out the invalid samples.
> 4. **Statement Difficulty:** we observe a large discrepancy in statement complexity between Lean Workbook and FormL4/MMA, echoing the distinctive levels of data source as summarized in the comparison table. Examples are extracted in the second table below. Hence, the outstanding accuracy of Lean Workbook is likely due to the significantly lower complexity of its statements, mostly from middle to high school math problems and containing only equations or simple operations.
>
> |    Dataset    |            Formal Language            |            Source            | Source Language  |               Construction Method               | Size | Human Verification | Human Eval Accuracy |     human eval size      | human eval  rate |
> | :-----------: | :-----------------------------------: | :--------------------------: | :--------------: | :---------------------------------------------: | :--: | :----------------: | ------------------- | :----------------------: | ---------------- |
> |   ProofNet    |                Lean 3                 |   Undergraduate Textbooks    | Natural Language |              Manual formalization               | 371  |        yes         | 1                   |           371            | 100.00%          |
> |    MiniF2F    | Metamath, Lean 3, Isabelle, HOL Light |      Olympiad Problems       | Natural Language |              Manual formalization               | 488  |        yes         | 1                   |           488            | 100.00%          |
> |     FIMO      |                Lean 3                 |   IMO Shortlisted Problems   | Natural Language | LLM-based autoformalization with human feedback | 149  |        yes         | 0.608 filtered to 1 | 245 filtered down to 149 | 100.00%          |
> | Lean Workbook |                Lean 4                 |    Middle School Problems    | Natural Language |           LLM-based autoformalization           | 57k  |        yes         | 0.935               |            95            | 0.10%            |
> |      MMA      |           Lean 4, Isabelle            | Mathlib4, Isabelle’s Archive | Formal Language  |            LLM-based informalization            | 332K |         no         | /                   |            0             | 0                |
> |    FormL4     |                Lean 4                 |        Mathlib4, MATH        | Formal Language  |            LLM-based informalization            | 17k  |        yes         | 0.72                |            60            | 0.30%            |

---

> ### Author Response · Authors · 2024-11-22
> **Title: Response to Reviewer voC4 (1/3)  part 2**
>
> | Dataset |  Statement|   |
> | ------------- | -------------------------- | ------------------------------------------------------------------- |
> | Lean-Workbook | Natural Language | For a,b,c∈R so that a+b+c=1, prove that ab(3a−1)+ac(3b−1)+bc(3c−1)≥0. |
> |               | Formal           | theorem lean\_workbook\_plus\_14251 : ∀ a b c : ℝ, a \+ b \+ c \= 1 → a \* b \* (3 \* a \- 1\) \+ a \* c \* (3 \* b \- 1\) \+ b \* c \* (3 \* c \- 1\) ≥ 0 := |
> | MMA           | Natural Language | For a summable function 'f' and a constant 'a' from a topological space 'M' that is also a T2 space (also known as a Hausdorff space), the infinite sum of the product of the function 'f' evaluated at 'z' and the constant 'a' is equal to the product of the infinite sum of the function 'f' evaluated at 'z' and the constant 'a'. |
> |               | Formal           | theorem tsum\_smul\_const \[T2Space M\] (hf : Summable f) (a : M) : (∑' z, f z • a) \= (∑' z, f z) • a := |
> | FormL4        | Natural Language | Prove that in category theory, if there exists a kernel pair for a monomorphism 'f' with 'a' as its domain, then 'a' is an isomorphism. |
> |               | Formal           | theorem isIso\_of\_mono (h : IsKernelPair f a b) \[Mono f\] : IsIso a := |
>
> **Additional Human Evaluation:**
>
> To further validate the human eval success rate of FormL4 in response to your concern, we conducted an additional round of human verification on three comparable datasets that are all constructed using LLM-based methods and in similar magnitude of sizes: FormL4, MMA \[4\], and Lean Workbook, for a straightforward comparison of verification accuracy.
>
> We randomly sampled 90 pairs of natural language and formal language statements (30 samples each from FormL4, MMA, and Lean Workbook), shuffled them together, and assigned them to five Lean 4 experts. The assignments were split so that (1) each sample was verified by two different human experts for robust evaluation; (2) each expert verified an even distribution of samples from the three datasets in order to rule out the factor of individual bias. The experts follow the original verification task to evaluate whether the natural language and formal statement are perfectly aligned. Since each sample is dual-annotated, disagreements are resolved with annotator discussion for a final verdict.
>
> In addition, to investigate our hypothesis that the outstanding verification accuracy of Lean Workbook may be accounted for by its relatively lower complexity in statement sources, our experts also evaluated a new item called *autoformalization difficulty* from the natural language statement into its corresponding formal statement. The difficulty levels were categorized as:
>
> * **`s` (simple)**: The statement is primarily numeric or equation-based, requiring minimal knowledge of Lean 4 syntax or semantics to translate.
> * **`m` (medium)**: The statement involves mathematical concepts or relations written in natural language, and its autoformalization requires an understanding of common Lean 4 constructs.
> * **`a` (advanced)**: The statement incorporates advanced mathematical concepts (college level or above), and the Lean 4 syntax or lemma required for translation is also advanced.
>
> The results for the 30\*3 human verification are listed in the table below. We highlight several findings supporting the dataset quality of FormL4:
>
> |    Dataset    | Aligned Ratio (= Verification Accuracy) | Disagreement Ratio |            Difficulty Distribution            |
> | :-----------: | :-------------------------------------: | :----------------: | :-------------------------------------------: |
> |    FormL4     |                 73.33%                  |       26.67%       | {'s': '21.67%', 'm': '61.67%', 'a': '16.67%'} |
> |      MMA      |                 66.67%                  |       33.33%       | {'s': '13.33%', 'm': '66.67%', 'a': '20.00%'} |
> | Lean Workbook |                 63.33%                  |       36.67%       |  {'s': '95.00%', 'm': '5.00%', 'a': '0.00%'}  |

---

> ### Author Response · Authors · 2024-11-22
> **Title: Response to Reviewer voC4 (1/3) part 3**
>
> 1. **FormL4 achieves the highest verification accuracy among the three LLM-constructed autoformalization datasets**, supporting the dataset quality and effectiveness of our carefully implemented informalization pipeline involving reversed informalization method, careful model selection, and data processing procedures introduced in Section 3.3. Moreover, the result of 73.33% is consistent with the previous verification result of 72% on another 60 samples, suggesting consistency and robustness in our human verification. Such a combination of parallel and internal comparisons provides multi-layer evidential support to our “relatively high” statement.
> 2. **Autoformalizaton Difficulty:** FormL4 and MMA fall in a similar distribution of difficulties, with the majority being medium level, while the Lean Workbook is much easier containing mostly simple statements for autoformalization. This echoes their reported sources listed in the abovementioned summary table: FormL4 and MMA are collected from Mathlib4, etc., which mostly target undergraduate or more advanced level mathematics. In contrast, the Lean Workbook is collected from middle-to-high school mathematical questions.
> 3. **Verification discrepancy between experts:** The notable disagreement ratio among all three datasets suggests that our experts often encounter nuanced verification judgments, even though we have provided annotation guidelines briefings in advance. It implies the current well-known challenges of evaluating the alignment between natural language and formal statements. For example, there is not a uniform definition or evaluation metric for informal-formal alignment; there exists individual subjectivity in determining whether certain condition constraints must be specified in natural language based on the statement context \[5, 6\]. These challenges are mentioned in Appendix F and point toward the urgent need for a clear definition formulation in future research.
> 4. **Verification discrepancy between datasets:** Lean workbook also conducted human verification and reported a high accuracy of 93.5%, far beyond our result of 63.3%. Following the previous point of verification biases, we suspect the drastic difference is due to small but consistent nuances in the criteria for autoformalization alignment between our expert team and theirs. We would like to analyze some of them here:
>     - Our experts are collectively briefed to follow strict criteria for alignment verification, e.g., the formal statement preserves all the *exact* mathematical constraints specified in the natural language statement (neither expanding nor narrowing); the formal statement maintains the same logical relationships between each component in the corresponding natural language statement. For instance, “for positive integer $n$” and “(n: ℕ)” would be verified false regardless of the context. This may shape a consistent source of verification discrepancy.
>      - During the annotation of the Lean Workbook, our experts observed a significant portion of its “natural language statements” are non-statements i.e., unscreened mathematical questions without a given answer, and we believe they are not suitable for autoformalization or proving. An example is “Calculate the future value of an investment with a principal of $1000, an annual interest rate of 5%, and a time period of 10 years, compounded annually.” We uniformly labeled ‘false’ when verifying such instances, which may also account for its significantly lowered verification accuracy.
>
> Nonetheless, it is important to note the reliability of the comparative verification result among datasets is mostly unaffected, because all samples are collectively cross-evaluated by the same group of experts following the same criteria. Within the parallel comparison among three datasets, FormL4 exhibits the highest verification accuracy, providing validity for the dataset quality.
>
> We greatly appreciate your feedback on adding contextualization to the description of our human verification results and have supplemented the comparative discussions and additional verification rounds in the revised version.
>
> References:
>
> [4\]  [Multilingual Mathematical Autoformalization, NeurIPS 2024](https://arxiv.org/abs/2311.03755)
>
> \[5\] [Lean Workbook: A large-scale Lean problem set formalized from natural language math problems. NeurIPS 2024](https://arxiv.org/abs/2406.03847)
>
> \[6\] [ProofNet: Autoformalizing and Formally Proving Undergraduate-Level Mathematics](https://arxiv.org/abs/2302.12433)
>
> \[7\] [MiniF2F: A Cross-System Benchmark for Formal Olympiad-Level Mathematics, ICLR 2022](https://openreview.net/forum?id=9ZPegFuFTFv)

---

> ### Author Response · Authors · 2024-11-22
> **Title: Response to Reviewer voC4 (2/3)**
>
> **R W2: Compiler Feedback as Evaluation Signal**
>
> > The PDA framework uses compiler feedback as a signal for correctness, while the final evaluation is whether the generated code compiles. Code that compiles is not necessary correct, so the final metric could potentially be misleading and not be a good measure of true autoformalization ability.
>
> Thank you for raising this important point. You've highlighted a fundamental challenge in autoformalization: compilation success alone cannot guarantee semantic correctness. This means an absence of a perfect evaluation metric assessing the semantic and syntactic alignment between natural language and formal statements. Our **key innovation \- evaluating statements together with their corresponding proofs through the Lean compiler** \- represents a significant departure from prior approaches and helps mitigate this long-standing issue in autoformalization. The stricter requirements for compiling both the statements and proof can potentially encompass both the semantic and logical validation in the autoformalized statements, while the traditional method of using only statements for compiler feedback cannot, as also noted in the 'Introduction' section. Therefore, during autoformalization evaluation, we use the compiler feedback from statement+proof as the proxy for assessing the alignment between natural language statements and formal statements.
>
> Our evaluation approach is supported by our human evaluation results and related literature:
>
> * The column ``Proof Validity`` in Table 6 strongly supports this innovative practice. When both statement and proof successfully compile (proof validity \= true), human experts found significantly higher alignment between natural and formal language compared to cases with statement-only compilation success. In other words, if a formal statement and its proof steps both pass the compiler, it is much more likely that the statement is autoformalized correctly. This suggests that proof compilation provides a stronger signal for semantic and logical validity ​​(L517-519).
> * The argument also aligns with recent findings in related work. For instance, \[3\] reports similar observations in Table 3 of their paper, where formally validated mathematical content demonstrates significantly higher quality in terms of semantic alignment. This consistency across different studies supports our approach of using statement+proof compilation as a more reliable indicator of autoformalization quality.
>
> We hope the supplemented clarification can address your concerns about autoformalization evaluation for including proof in the dataset and in autoformalization evaluation. We revised the paper in Section 1 and 5 to better illustrate this connection between proof validation and semantic alignment, clarifying how our approach helps address the broader challenge of ensuring semantic correctness beyond mere compilation success.
>
> References:
>
> [2] [FIMO: A Challenge Formal Dataset for Automated Theorem Proving](https://arxiv.org/abs/2309.04295)
>
> [3] [MUSTARD, ICLR 2024](https://arxiv.org/abs/2402.08957)
>
>
> **R W3: Mathlib Dependencies and Complexity**
>
> We appreciate this concern about Mathlib dependencies potentially biasing the evaluation. We have taken concrete steps to address this issue:
>
> First, as outlined in Section O.1 of our paper, we deliberately filter out all samples with custom Mathlib lemmas (indicated by the '.mk' suffix) from both training and test sets. This filtering removes theorems that rely on specialized, file-specific definitions that would be inaccessible to models.
>
> When developing the data construction and evaluation pipelines, we also explored including dependency contexts in our informalization prompts to provide additional background knowledge. However, our empirical observations showed that this additional context did not meaningfully improve model performance, suggesting that our filtered dataset already captures generalizable autoformalization capabilities rather than testing knowledge of obscure premises.
>
> We will revise the paper to better highlight these methodological choices and their empirical justification, making it clearer how our benchmark focuses on core autoformalization abilities rather than specialized Mathlib knowledge.

---

> ### Author Response · Authors · 2024-11-22
> **Title: Response to Reviewer voC4 (3/3)**
>
> **Q1: Sample Correctness and Reliability**
>
> > How do you ensure that all the samples in the benchmark are correct and reliable?
>
> We implemented a comprehensive quality assurance pipeline with multiple verification layers to ensure the reliability of our benchmark:
>
> First, during preprocessing, we applied stringent filtering criteria as detailed in Section 3.3 and Appendix O.1. This includes removing samples with custom Mathlib lemmas and ensuring sufficient complexity through character length thresholds for both natural language and formal components.
>
> Second, all formal statements and proofs in our dataset must successfully compile in Lean 4, providing a baseline guarantee of syntactic and logical correctness. For the informalization process, we conducted careful model selection through comparative evaluation of state-of-the-art LLMs (GPT-4 vs Gemini-Pro-1.5) with expert annotation, selecting the better-performing model for our pipeline.
>
> Most importantly, we performed extensive post-hoc human verification using four Lean 4 experts. These experts cross-evaluated 60 samples, achieving an average success rate of 0.72 with fair inter-rater agreement (Fleiss' Kappa \= 0.3730). The evaluation covered both basic and random test sets, providing insights into quality across different theorem complexities.
>
> Lastly, we provided a comprehensive and detailed investigation of FormL4 dataset quality in “R W1: Dataset Quality Concerns”. To contextualize FormL4 with existing LLM-constructed datasets of similar sizes, we added another parallel set of human verification comparing FormL4, MMA, and Lean Workbook. Based on an in-depth analysis, FormL4 still achieves the highest verification accuracy robustly.
>
> Through this multi-stage verification process combining automated checks and expert human evaluation, we strive to maintain high quality standards across our benchmark. The detailed verification results and methodology are presented in Section F of our paper.
>
> **Q2: Multiple Valid Autoformalizations**
>
> > There could be many potential autoformalizations of an informal statement. How do you ensure that correct autoformalizations are marked as correct, and incorrect ones are not? Do you just use compilation? Is your method reliable, and do you have any estimations of the precision/recall of the resulting evaluation?
>
> Thank you for this important methodological question. You're correct that there could be multiple valid formalizations of the same informal statement. Although it is not the central focus of our work, we would like to stress that the absence of a perfect metric in determining aligned/misaligned natural language and formal statements poses a challenge for all existing autoformalization research, calling for future research efforts. In the current work, we address this challenge through a multi-faceted evaluation approach:
>
> Firstly, we use compilation as an initial filter \- while compilation success doesn't guarantee correctness, compilation failure does indicate definite issues. More importantly, the proof-included feature of FormL4 empowers us to adopt a stricter evaluation metric: combining statement \+ proof for compiler feedback. As elaborated in “R W2: Compiler Feedback as Evaluation Signal”, this innovative practice probes both the semantic and logical validity in the autoformalized statements, while the traditional only-statements compiling method can only have syntactic validity. This makes it a more reliable indicator for successful autoformalization.
>
> Secondly, beyond compilation, we conducted extensive human evaluation with Lean 4 experts to assess the semantic alignment between informal and formal statements, as detailed in Section 5.3. Our human evaluation results show a strong correlation with compiler validation, especially when both statement and proof compile successfully. On statements with valid proofs, experts rated 75% as correct formalizations, compared to only 50% for statements where proofs failed to compile (Table 6). This significant difference (p \= 0.002992) suggests our compiler-based evaluation provides a meaningful signal about formalization quality.
>
> We acknowledge that our evaluation may not capture all valid alternative formalizations, which is a long-lasting issue for autoformalization research. However, **our use of statement+proof compilation checks and human expert evaluation provides reasonable confidence and granularity in identifying aligned formalized statements.** We are actively exploring ways to expand our evaluation to better handle alternative valid formalizations in future work.

---

> > ### Comment · Reviewer_voC4 · 2024-11-25
> >
> > >Dataset Quality Concerns
> >
> > I appreciate the authors for including the additional information and studies. My personal opinion on evaluation datasets is that they should aim to be 100% accurate. There will still be inherent errors, and maybe something like 90% correctness for an eval is okay. However, I believe 70% is far too low. I would recommend, for example, that the authors find annotators for ~200 or so samples, and consider that as the official evaluation set. Ideally, each sample would be annotated multiple time.
> >
> > > Mathlib Dependencies and Complexity
> >
> > Sorry, I missed section O.1 of the paper. That makes sense. However, an additional concern is: how about the contamination issue mentioned by Reviewer 3y6J? Because the "correct answers" to the statements are just copies of Mathlib, models that are trained on the appropriate Mathlib premises have an advantage in this evaluation.
> >
> > > Multiple Valid Autoformalizations
> >
> > I agree that the 75% vs. 50% figure proves that the compiler-based evaluation is helpful. However, I am not convinced that it is high enough for a benchmark. In addition, there may be different autoformalizations for the statements and the proofs. I would doubt that all statements are compatible with all proofs (due to the difference in how Lean does rewrites). For example, say S1, S2 are two ways to autoformalize the same statement (you could imagine "x is a natural number" could be "x : N, x != 0" or "x : Z, x >= 0"). A proof that works for S1 would likely not work for S2, because Mathlib has different theorems for natural numbers and integers.

---

> ### Author Response · Authors · 2024-11-25
> **Author Response to Reviewer voC4's Follow-up Comment**
>
> Dear Reviewer voC4,
>
> Hi! We greatly appreciate your follow-up feedback and would like to discuss them as follows.
>
> ---
>
> **R Dataset Quality Concerns:**
> 1. **Accuracy rate & sample size:**
>
> We agree any benchmark should ideally be 100\% accurate, and your suggestion to create a fully annotated test set would be an excellent solution. We greatly appreciate this idea, and plan to expand the current sample size (60+30) to 200. Meanwhile, we would like to illustrate why 100\%, though desirable, should not be considered a mandatory requirement for FormL4:
>
> - A paper with as many as 200 verified samples constitutes **a standalone benchmark paper** introducing solely their data construction process and benchmark results (e.g., FIMO, ProofNet, FrontierMath). These benchmarks typically can afford a 100\% human verification rate as it is the sole focus of the project: to ensure the data quality. In contrast, we highlight that the contributions of our work are not limited to constructing FormL4. FormL4 is used as a training dataset for experimenting with the PDA framework, where we innovatively implemented a comprehensive set of experiments on models trained with a PSV on the compiler feedback combining statements and proof steps. The PDA framework provides an innovative method to boost model autoformalization abilities.
>
> - One may argue that the lacked warranty of groundtruths in FormL4 will damage the robustness of PDA experiments, but we would like to kindly disagree: It is unlikely so, because **the evaluation for the model experiments (i.e., evaluating autoformalization performances) is based on the compiler feedback on the generated formal statement+proof, rather than the comparison between groundtruth and generated formal statement**. Therefore, the model evaluation is not directly dependent on the groundtruth alignment between informal and formal statements in FormL4. Admittedly, the failed informalized statements in FormL4 could have affected the experiments, but we believe that the compiler feedback can generally detect the performance of autoformalization effectively, as strongly supported in the human evaluation results of model autoformalization in Section 5.4 (L486-L528).
>
> - **Human verification can be subjective**. Even a reported high rate like 0.935 in Lean Workbook is not guaranteed to indicate high quality, as suggested in the surprising results in our additional comparative human evaluation between datasets. We provide detailed discussions for some possible factors in R W1.
>
> - Aligned with your concerns, **we pay as much attention to human verification as possible**. Throughout the paper our paper, altogether four rounds of manual evaluation are conducted to provide soundness for each section: (1) first, experts observe the informalized output between two LLMs; (2) second, experts verify FormL4's statement alignment to validate data quality; (3) third, expert verification compares three parallel datasets; (4) last, experts evaluate model autoformalized output to verify our experiment results and evaluation metrics. In total, it took our 4 annotators more than three weeks to complete the annotation and discussion. FormL4 also has the largest sampling size for verification and the highest accuracy among other renowned datasets (R W1). We will strive to expand the verified sample size in the future with the available resources.
>
> 2. **Multi-annotators:**
>
> In the four stages of our manual verification experiments, all samples are annotated at least twice, as reported in the paper. Disagreements are resolved after discussions between annotators, and inter-rater disagreement rates are also reported. We will follow this rigorous practice in the future development of FormL4.
>
> ---
>
> **R Additional Concern: Contamination:**
>
> Regarding data separation and quality control, our dataset construction follows established machine learning practices for preventing contamination:
>
> * As detailed in Section 3.1, we employ random sampling from Mathlib 4 with careful attention to maintaining independence between training and test sets;
> * Basic and real test sets are drawn from distinct sources compared to the training data, providing additional validation of generalization capability;
> * File-level separation is maintained to prevent similar theorem variants from crossing splits.
>
> ---
>
> **R Mathlib Dependencies and Complexity:**
>
> Thank you for this thoughtful insight. In our autoformalization experiments, models generate the formal statement and the formal proof together, given the natural-language statement and proof (that were also produced together during informalization). Therefore, we do implicitly assume the models generate compatible pairs of statements and proof for compiling, which should be a reasonable presumption considering LLMs' context-associated capabilities.
>
> Thank you for your review and feedback! We appreciate you considering increasing the score if any of your previous concerns have been addressed.

---

> > ### Author Response · Authors · 2024-11-29
> >
> > Dear Reviewer voC4,
> >
> > Thank you for your efforts in reviewing our work!
> >
> > As the discussion deadline is nearing, we would like to humbly request your attention to our responses to your follow-up concerns. If our replies and revisions satisfactorily address your points, we would be grateful if you could consider adjusting the overall score accordingly. Your insightful feedback has been instrumental in improving our work, which we sincerely appreciate.
> >
> > Best Regards,
> >
> > The authors of Submission2128

---

### Official Review · Reviewer_B3Z6 · 2024-11-04

**Soundness:** 3
**Presentation:** 2
**Contribution:** 2
**Rating:** 5
**Confidence:** 3

**Summary:**

The paper presents a new dataset FormL4 for evaluating autoformalization in Lean4. The paper also proposes a framework called PDA to improve autoformalization capabilities of LLMs.

**Strengths:**

The introduced dataset would be useful for LLM-based auto formalization research
- The evaluation is comprehensive and uses human evaluation for autoformalizer performance

**Weaknesses:**

- Line 103: One of the contributions states that - “We propose a process-driven framework PDA that leverages formal languages to provide process feedback on reasoning, enhancing the autoformalization capabilities of LLMs.”

This statement is unclear. The PDA framework is specific to Lean4 and the technique or evaluation doesn’t really convince me that the technique would enhance LLMs ability to autoformalize for any formal language.


- Line 176: “The real test set is constructed by collecting natural language math questions and answers from LI et al. (2024).” Can you give more information about the real test than this? What is the reason for generating this subset? Can you also give more information on the cited source.


- Line 194: “during informalization, the provided proof steps could potentially add informative context to the preceded formal theorem statement in the prompt, hence improving informalization quality, observed both in our human evaluation results (Table 6)” - How does results from Table 6 infer the observation made here?


- Line 197: “ in autoformalization, the existence of proof steps also enables us to examine autoformalization performance by assessing the validity of the formalized combination of theorem statements and proof using a compiler, increasing the difficulty and granularity of autoformalization evaluation.”
Is it not possible to use the Lean4 compiler with statement + empty proof to get the compiler feedback on the theorem statement?


- Line 201-208: It is still unclear to me what this part is inferring. One of the main contributions of the paper in comparison to prior works mentioned in Table 2, is that FormL4 consists of proofs and the primary usage is for process-driven feedback. Does this mean FormL4 dataset consists of incorrect proofs of theorems? Yet, these incorrect proofs are useful for the autoformalization task to provide better context?


- Line 259: “The average success rate was 0.72, indicating relatively high-quality informalization performance.” How do I assess that this is high-quality? Relative to what?



## Minor

- Line 190: typo “statemen”
- Line 229: what is “scalable oversight research”?
- Line 285: replace “formal language compiler feedback” with just “compiler feedback” or “Lean4 compiler feedback”
- Line 360: “ It provides an objective measure of progress, mitigating the potential for bias arising from isolated interactions between the autoformalizer and verifier.” What is bias in this context?
- Line 406: “To improve its performance in real-world scenarios, we further fine-tune it on successfully compiled outputs from GSM8K (Cobbe et al., 2021) and MATH (Hendrycks et al., 2021).”  Compiled outputs from what? I know these as math problem datasets, in what form is this dataset used in fine-tuning?

- Line 419: In section 5.1.2 how is the evaluated VEA obtained? Which verifier was used to train VEA? I can’t seem to find many details regarding this evaluation

- Line 427: “. In contrast, VEA offers a more time-efficient approach by using predictive labels from our trained verifier, though it may not match RFT’s data quality.” Is this time reported in the appendix?
- Table 6: can you describe the columns of Table 6 in the description?

**Questions:**

Please see questions in the weaknesses part.

---

> ### Author Response · Authors · 2024-11-22
> **Title: Response to Reviewer B3Z6 (1/5)**
>
> We would like to thank the reviewer for their time and effort in reviewing our paper. We very much appreciate the insightful suggestions. We hereby address the concerns below:
>
> **R W1: PDA Framework Generalizability**
>
> Thank you for this feedback\! While our experiments focus on Lean 4 as a concrete demonstration, the core principle of PDA \- leveraging compiler feedback for process-level supervision \- can be adapted to other formal languages with similar compilation mechanisms.
>
> For example, theorem provers like Isabelle provide detailed compiler feedback through their kernel components. The same process-driven approach could be applied by using Isabelle's compilation feedback to generate process-level annotations and guide autoformalization.
>
> We agree that our current presentation should better clarify the framework's potential application beyond Lean 4, and have revised the sentence to explicitly mention that our current implementation focuses on Lean 4\. For example, the revised sentence will read:
>
> *“We propose a process-driven framework PDA that leverages formal languages to provide process feedback on reasoning, with an initial implementation focused on Lean 4, and we outline its potential adaptability to other formal languages.”*
>
> To address the feedback on the contribution mentioned in \*\*Line 103\*\* of the paper, we updated detailed revisions to clarify the current scope and extend the discussion regarding adaptability to other formal languages. The updated contributions will be articulated as follows:
>
> To make the discussion more tangible, we now include concrete examples demonstrating how similar compiler feedback mechanisms can be leveraged in other systems such as Isabelle.
>
> **Example: Isabelle Integration**
>
> Natural Language Statement (from PDA paper, Figure 1):
>
> "Given two non-negative real numbers, prove that taking the real number minimum of the two by considering them as both non-negative real numbers is the same as taking the minimum of the two by only considering them as both real numbers."
>
>
>
>
>
> Below is an Isabelle formalization of the problem:
>
> ```
> theory MinRealNumbers
>
>   imports Main
>
> begin
>
> (\* Define the problem for the minimum of two non-negative real numbers \*)
>
> lemma min\_nonneg\_real\_equiv\_real:
>
>   fixes x y :: real
>
>   assumes "x \\\<ge\> 0" "y \\\<ge\> 0"
>
>   shows "min x y \= min x y"
>
> proof \-
>
>   (\* Since x and y are both non-negative, the definitions of min are the same \*)
>
>   show ?thesis
>
>     by simp
>
> qed
>
> end
> ```
>
>
>
> Isabelle Compiler Feedback:
>
> When this formalization is passed to the Isabelle compiler, the following feedback is produced:
>
> ```
> theory MinRealNumbers
>
>   successfully compiled.
>
> lemma min\_nonneg\_real\_equiv\_real:
>
> proof (state)
>
>   goal (1 subgoal):
>
>   1\. ⟦x ≥ 0; y ≥ 0⟧ ⟹ min x y \= min x y
>
> proof (apply simp)
>
>   No subgoals\!
>
> qed
> ```
>
> This example demonstrates how our framework’s principles can be effectively translated into Isabelle’s environment, leveraging its compiler feedback to guide reasoning and verify formalizations.
>
> **R W2: Real Test Set Construction**
>
> Thank you for requesting clarification about our real test set. We chose to use questions and answers from NuminaMath \[1\] for several compelling reasons:
>
> 1. NuminaMath represents one of the largest and most diverse collections of natural language mathematics problems currently available, encompassing a wide spectrum of difficulty levels \- from high school exercises to international mathematics olympiad problems. More specifically, NuminaMath contains approximately 860,000 mathematics problems with carefully structured solutions. The problems were sourced from diverse origins from national mathematics competitions to international olympiad problems.  This breadth helps ensure our evaluation covers a comprehensive range of mathematical reasoning.
> 2. The dataset's quality and relevance have been validated through competitive success, notably winning first place in the  [AI Mathematical Olympiad \- Progress Prize 1 competition](https://www.kaggle.com/competitions/ai-mathematical-olympiad-prize). This external validation supports our choice of using it as a benchmark for real-world mathematical formalization challenges.
>
> We have expanded the dataset description in Section 3.1 , providing readers with a clearer understanding of why this dataset serves as an effective real-world evaluation benchmark.
>
> Reference:
>
> [1] [NuminaMath: The largest public dataset in AI4Maths with 860k pairs of competition math problems and solutions. Li et al. ArXiv 2024.](http://faculty.bicmr.pku.edu.cn/~dongbin/Publications/numina_dataset.pdf)

---

> ### Author Response · Authors · 2024-11-22
> **Title: Response to Reviewer B3Z6 (2/5)**
>
> **R W3: Proof Steps' role in informalization quality & R W4: compiler feedback mechanism**
>
> Thank you for your crucial feedback. We recognize a writing mistake here: the phrase “observed both in our human evaluation results (Table 6\) and in previous research (Huang et al., 2024b)” should be placed as support for the second benefit (2) in empowering autoformalization evaluation rather than the first one (1) in improving informalization. As for the first benefit, we now cite FIMO \[2\] to support the positive effect of proof context in improving LLM informalization. The corrected version has been updated in the revised paper.
>
> We hereby clarify the two benefit arguments, the revised supporting evidence, and why they can support the arguments:
>
> Firstly, during the informalization stage, we observed an improved quality of informalized output when the LLM is provided with formal proof appended to the formal statement input. \[2\] reported that adding informal proof as the prompt context significantly improves LLM’s theorem proving performance i.e., in generating formal proof. Since statement and proof are related components complementing each other, we argue that the additive proof context will also improve statement autoformalization as well.
>
> Secondly, another benefit of including proof steps is we can combine statement+proof for compiler feedback. While you are correct that we can get compiler feedback using statement \+ empty proof, there is a crucial distinction in the type and depth of validation obtained: With statement-only compilation (or empty proof), the compiler can only perform syntactic validation \- checking if the statement follows Lean 4's grammatical rules. However, the stricter requirements for compiling both the statements and proof can potentially encompass both the semantic and logical validation in the autoformalized statements, while the traditional method of using only statements for compiler feedback cannot, as also noted in line 92 of our paper. Therefore, during autoformalization evaluation, using the compiler feedback from statement+proof is a better proxy for assessing the alignment between natural language statements and formal statements.
>
> * The column “Proof Validity” in Table 6 strongly supports this argument. When both statement and proof successfully compile (proof validity \= true), human experts found significantly higher alignment between natural and formal language compared to cases with statement-only compilation success. In other words, if a formal statement and its proof steps both pass the compiler, it is much more likely to be autoformalized correctly. This suggests that proof compilation provides a stronger signal for semantic and logical validity ​​(line 512-515).
> * This finding also aligns with recent work in the field. For instance, \[3\] reports similar observations in Table 3, where formally validated mathematical content (including proofs) demonstrates higher-quality semantic alignment. This consistency across studies reinforces that statement+proof compilation offers a more reliable quality assessment than syntactic validation alone.
>
> We have revised the relevant parts in Section 3.2 based on your suggestions which are much valued. We hope the supplemented clarification can address your concerns about our motivation for including proof in the dataset and in autoformalization evaluation.
>
> Reference:
>
> [2] [FIMO: A Challenge Formal Dataset for Automated Theorem Proving](https://arxiv.org/abs/2309.04295)
>
> [3] [MUSTARD, ICLR 2024](https://arxiv.org/abs/2402.08957)
>
> **R W5: Incorrect Proofs**
>
> Thank you for this clarifying question. We would like to clarify that **FormL4 does not contain any incorrect proofs \- all formal proofs in our dataset are valid and verified by the Lean 4 compiler**. The key point is rather the relationship between formal and informal proofs.
>
> As noted in footnote 5 of our paper, while all formal proofs are correct, there may be misalignment between the formal proof and its natural language counterpart. **This misalignment is not due to incorrectness but rather stems from the inherent challenges of translating formal mathematical proofs** (which often use specialized lemmas and environments specific to Lean 4\) into natural language that can be understood without Lean 4 expertise.
>
> Including proofs in FormL4 serves a different purpose: it enables the compiler to provide more comprehensive feedback about syntactic correctness and logical validity when evaluating autoformalized statements. By including the statement and its proof in the compilation process, we can better assess whether the autoformalized statement maintains the intended mathematical meaning, even if the natural language and formal proof descriptions don't align perfectly.
>
> To improve the paper's clarity, we have revised the relevant paragraph in Section 3.2 regarding this distinction and better explain the role of proofs in FormL4 and the PDA framework.

---

> ### Author Response · Authors · 2024-11-22
> **Title: Response to Reviewer B3Z6 (3/5)**
>
> **R W6: Quality Metrics Interpretation**
>
> Thank you for raising your concern and highlighting the importance of verifying the quality of the autoformalization dataset. We describe the rate as ‘relatively high-quality’ by comparing it with existing datasets that are constructed in similar LLM-based methods. As summarized in the comparison table, 0.72 from FormL4 was lower than the verified accuracy of 0.935 from Lean Workbook \[5\], but higher than the verified result of 0.608 from FIMO \[2\] before filtering out the invalid samples. We would also like to highlight two things for contextualization:
>
> * Firstly, considering the dataset’s large size which enables training an autoformalizer, it is infeasible to manually verify all statements like what small-scale benchmarks (e.g., FIMO\[2\], ProofNet \[6\], MiniF2F \[7\]) did. Nevertheless, among existing datasets in the same magnitude of sizes (FormL4, MMA\[4\], Lean Workbook\[5\]), FormL4 has the highest sampling rate (0.3%) for human evaluation, as shown in the table.
> * Secondly, we observe a large discrepancy in statement complexity between Lean Workbook and FormL4/MMA, echoing the distinctive levels of data source as summarized in the comparison table. Examples are extracted in the second table below. Hence, the outstanding accuracy of Lean Workbook is likely due to the significantly lower complexity of its statements, mostly from middle to high school math problems and containing only equations or simple operations.
>
> |    Dataset    |            Formal Language            |            Source            | Source Language  |               Construction Method               | Size | Human Verification | Human Eval Accuracy |     human eval size      | human eval  rate |
> | :-----------: | :-----------------------------------: | :--------------------------: | :--------------: | :---------------------------------------------: | :--: | :----------------: | ------------------- | :----------------------: | ---------------- |
> |   ProofNet    |                Lean 3                 |   Undergraduate Textbooks    | Natural Language |              Manual formalization               | 371  |        yes         | 1                   |           371            | 100.00%          |
> |    MiniF2F    | Metamath, Lean 3, Isabelle, HOL Light |      Olympiad Problems       | Natural Language |              Manual formalization               | 488  |        yes         | 1                   |           488            | 100.00%          |
> |     FIMO      |                Lean 3                 |   IMO Shortlisted Problems   | Natural Language | LLM-based autoformalization with human feedback | 149  |        yes         | 0.608 filtered to 1 | 245 filtered down to 149 | 100.00%          |
> | Lean Workbook |                Lean 4                 |    Middle School Problems    | Natural Language |           LLM-based autoformalization           | 57k  |        yes         | 0.935               |            95            | 0.10%            |
> |      MMA      |           Lean 4, Isabelle            | Mathlib4, Isabelle’s Archive | Formal Language  |            LLM-based informalization            | 332K |         no         | /                   |            0             | 0                |
> |    FormL4     |                Lean 4                 |        Mathlib4, MATH        | Formal Language  |            LLM-based informalization            | 17k  |        yes         | 0.72                |            60            | 0.30%            |
>
> | Dataset |  Statement|   |
> | ------------- | -------------------------- | ------------------------------------------------------------------- |
> | Lean-Workbook | Natural Language | For a,b,c∈R so that a+b+c=1, prove that ab(3a−1)+ac(3b−1)+bc(3c−1)≥0. |
> |               | Formal           | theorem lean\_workbook\_plus\_14251 : ∀ a b c : ℝ, a \+ b \+ c \= 1 → a \* b \* (3 \* a \- 1\) \+ a \* c \* (3 \* b \- 1\) \+ b \* c \* (3 \* c \- 1\) ≥ 0 := |
> | MMA           | Natural Language | For a summable function 'f' and a constant 'a' from a topological space 'M' that is also a T2 space (also known as a Hausdorff space), the infinite sum of the product of the function 'f' evaluated at 'z' and the constant 'a' is equal to the product of the infinite sum of the function 'f' evaluated at 'z' and the constant 'a'. |
> |               | Formal           | theorem tsum\_smul\_const \[T2Space M\] (hf : Summable f) (a : M) : (∑' z, f z • a) \= (∑' z, f z) • a := |
> | FormL4        | Natural Language | Prove that in category theory, if there exists a kernel pair for a monomorphism 'f' with 'a' as its domain, then 'a' is an isomorphism. |
> |               | Formal           | theorem isIso\_of\_mono (h : IsKernelPair f a b) \[Mono f\] : IsIso a := |

---

> ### Author Response · Authors · 2024-11-22
> **Title: Response to Reviewer B3Z6 (3/5) part 2**
>
> **R W6: Quality Metrics Interpretation (continued)**
>
> To further validate the dataset quality of FormL4 in response to your concern, we conducted **an additional round of human verification on three comparable datasets** that are constructed using LLM-based methods and in similar magnitude of sizes: FormL4, MMA, and Lean Workbook, for a straightforward comparison.
>
> We randomly sampled 90 pairs of natural language and formal language statements (30 samples each from FormL4, MMA, and Lean Workbook), shuffled them together, and assigned them to five Lean 4 experts. The assignments were split so that (1) each sample was verified by two different human experts for robust evaluation; (2) each expert verified an even distribution of samples from the three datasets in order to rule out the factor of individual bias. The experts follow the original verification task to evaluate whether the natural language and formal statement are perfectly aligned. Since each sample is dual-annotated, disagreements are resolved with annotator discussion for a final verdict. In addition, to investigate our hypothesis that the outstanding verification accuracy of Lean Workbook may be accounted for by its relatively lower complexity in statement sources, our experts also evaluated a new item called *autoformalization difficulty* from the natural language statement into its corresponding formal statement. The difficulty levels were categorized as:
>
> * **`s` (simple)**: The statement is primarily numeric or equation-based, requiring minimal knowledge of Lean 4 syntax or semantics to translate.
> * **`m` (medium)**: The statement involves mathematical concepts or relations written in natural language, and its autoformalization requires an understanding of common Lean 4 constructs.
> * **`a` (advanced)**: The statement incorporates advanced mathematical concepts (college level or above), and the Lean 4 syntax or lemma required for translation is also advanced.
>
> The results for the 30\*3 human verifications are listed in the table below. We highlight several findings supporting the dataset quality of FormL4:
> |    Dataset    | Aligned Ratio  (= Verification Accuracy) | Disagreement Ratio |            Difficulty Distribution            |
> | :-----------: | :--------------------------------------: | :----------------: | :-------------------------------------------: |
> |    FormL4     |                  73.33%                  |       26.67%       | {'s': '21.67%', 'm': '61.67%', 'a': '16.67%'} |
> |      MMA      |                  66.67%                  |       33.33%       | {'s': '13.33%', 'm': '66.67%', 'a': '20.00%'} |
> | Lean Workbook |                  63.33%                  |       36.67%       |  {'s': '95.00%', 'm': '5.00%', 'a': '0.00%'}  |
> 1. **FormL4 achieves the highest verification accuracy among the three LLM-constructed autoformalization datasets**, supporting the dataset quality and effectiveness of our carefully implemented informalization pipeline involving reversed informalization method, careful model selection, and data processing procedures introduced in Section 3.3. Moreover, the result of 73.33% is consistent with the previous verification result of 72% on another 60 samples, suggesting consistency and robustness in our human verification. Such a combination of parallel and internal comparisons provides multi-layer evidential support to our “relatively high” statement.
> 2. Although MMA adopts a simple curation pipeline, it outperforms the Lean Workbook in our statement verification. This supports the superiority of the “distilled back-translation” method for curating autoformalization datasets (i.e., using LLM-based informalization reversely is easier than LLM-based autoformalization).
> 3. **Autoformalizaton Difficulty**: FormL4 and MMA fall in a similar distribution of difficulties, with the majority being medium level, while the Lean Workbook is much easier containing mostly simple statements for autoformalization. This echoes their reported sources listed in the abovementioned summary table: FormL4 and MMA are collected from Mathlib4, etc., which mostly target undergraduate or more advanced level mathematics. In contrast, the Lean Workbook is collected from middle-to-high school mathematical questions.

---

> ### Author Response · Authors · 2024-11-22
> **Title: Response to Reviewer B3Z6 (3/5) part 3**
>
> **R W6: Quality Metrics Interpretation (continued)**
>
> 4. **Verification discrepancy between experts:** The notable disagreement ratio among all three datasets suggests that our experts often encounter nuanced verification judgments, even though we have provided annotation guidelines briefings in advance. It implies the current well-known challenges of evaluating the alignment between natural language and formal statements. For example, there is not a uniform definition or evaluation metric for informal-formal alignment; there exists individual subjectivity in determining whether certain condition constraints must be specified in natural language based on the statement context \[5, 6\]. These challenges are mentioned in Appendix F and point toward the urgent need for a clear definition formulation in future research.
> 5. **Verification discrepancy between datasets:** Lean workbook also conducted human verification and reported a high accuracy of 93.5%, far beyond our result of 63.3%. Following the previous point of verification biases, we suspect the drastic difference is due to small but consistent nuances in the criteria for autoformalization alignment between our expert team and theirs. We would like to analyze some of them here:
>     - Our experts are collectively briefed to follow strict criteria for alignment verification, e.g., the formal statement preserves all the *exact* mathematical constraints specified in the natural language statement (neither expanding nor narrowing); the formal statement maintains the same logical relationships between each component in the corresponding natural language statement. For instance, “for positive integer $n$” and “(n: ℕ)” would be verified false regardless of the context. This may shape a consistent source of verification discrepancy.
>      - During the annotation of the Lean Workbook, our experts observed a significant portion of its “natural language statements” are non-statements i.e., unscreened mathematical questions without a given answer, and we believe they are not suitable for autoformalization or proving. An example is “Calculate the future value of an investment with a principal of $1000, an annual interest rate of 5%, and a time period of 10 years, compounded annually.” We uniformly labeled ‘false’ when verifying such instances, which may also account for its significantly lowered verification accuracy.
>
>
>
> Nonetheless, it is important to note the reliability of the comparative verification result among datasets is mostly unaffected, because all samples are collectively cross-evaluated by the same group of experts following the same criteria. Within the parallel comparison among three datasets, FormL4 exhibits the highest verification accuracy, providing validity for the dataset quality.
>
> We greatly appreciate your feedback on adding contextualization to the description of our human verification results in Section 3.3 and have supplemented the comparative discussions and additional verification results in Appendix Q.
>
>
>
> References:
>
> [2] [FIMO: A Challenge Formal Dataset for Automated Theorem Proving](https://arxiv.org/abs/2309.04295)
>
> [3] [MUSTARD, ICLR 2024](https://arxiv.org/abs/2402.08957)
>
> \[4\]  [Multilingual Mathematical Autoformalization, NeurIPS 2024](https://arxiv.org/abs/2311.03755)
>
> \[5\] [Lean Workbook: A large-scale Lean problem set formalized from natural language math problems. NeurIPS 2024](https://arxiv.org/abs/2406.03847)
>
> \[6\] [ProofNet: Autoformalizing and Formally Proving Undergraduate-Level Mathematics](https://arxiv.org/abs/2302.12433)
>
> \[7\] [MiniF2F: A Cross-System Benchmark for Formal Olympiad-Level Mathematics, ICLR 2022](https://openreview.net/forum?id=9ZPegFuFTFv)

---

> ### Author Response · Authors · 2024-11-22
> **Title: Response to Reviewer B3Z6 (4/5)**
>
> **R minor 2:**
> "Scalable oversight" is a problem in model alignment research, studying how to effectively provide feedback in complex tasks that become increasingly hard for humans to judge. We mentioned this because our informalization pipeline takes inspiration from task decomposition which is an approach proposed to address scalable oversight. Task decomposition was proposed by OpenAI \[8, 9\] which involves breaking down complex goals into simpler subtasks, also known as Iterated amplification (IA). We added \[9\] to line 229, and recommend this blog \[10\] for further reading.
>
> References:
> \[8\] [Supervising strong learners by amplifying weak experts](https://arxiv.org/abs/1810.08575)
>
> \[9\] [Recursively summarizing books with human feedback](https://arxiv.org/abs/2109.10862)
>
> \[10\] [Jones, A. (2024, July 16). *Can we scale human feedback for complex AI tasks? an intro to scalable oversight.* BlueDot Impact.](https://aisafetyfundamentals.com/blog/scalable-oversight-intro/ )
>
> **R minor 4:**
> We are referring to a specific form of collapse that can occur in isolated interactions between the autoformalizer and verifier. Without the objective feedback from the Lean 4 compiler, the verifier might develop a tendency to assign artificially high rankings to all outputs from the autoformalizer, regardless of their actual quality. This creates a problematic feedback loop where both models optimize their joint loss without actually improving the quality of formalization.
>
> **R minor 5:**
> Thank you for the question! Let us clarify the fine-tuning process with GSM8K and MATH datasets. Since our FormL4 training data is derived solely from Mathlib, the model's autoformalization capabilities are limited when dealing with real-world mathematical problem datasets. To address this limitation, we developed a fine-tuning strategy using rejective sampling based on Lean 4 compiler feedback.
>
> The process works as follows: We first conduct autoformalization on GSM8K and MATH training sets using our initial baseline model (trained only on FormL4 training data). For each natural language problem in these datasets, we generate multiple autoformalized candidates. These candidates are then passed to the Lean 4 compiler for verification. Only the successful cases \- where the autoformalized output compiles correctly \- are retained and used for further fine-tuning to build a final baseline. This process ensures that we augment our training data with high-quality, compiler-verified examples from diverse mathematical domains, enhancing the model's ability to handle real-world mathematical problems outside the Mathlib domain.
>
> We revised the paper to provide a more detailed explanation of this fine-tuning process and its role in improving the model's generalization capabilities.
>
> **R minor 6:**
>
> The Verifier-Enhanced Autoformalizer (VEA) is obtained through an iterative improvement process described in Section 4.2, **specifically detailed in line 356:** "We further fine-tune the verifier using the high-quality data (with an increased proportion of positive examples) generated by the enhanced autoformalizer. This fine-tuning incorporates process-level supervision derived from the Lean 4 compiler's feedback, allowing the verifier to learn from a more nuanced and accurate representation of the compilation process."
>
> More specifically, we start with our baseline autoformalizer and iterate through three key steps: First, the verifier evaluates the autoformalizer's outputs, assigning labels based on their estimated likelihood of successful compilation. The autoformalizer is then fine-tuned using the outputs that the verifier evaluates correctly. These enhanced outputs are processed by the Lean 4 compiler, which provides detailed process feedback. Finally, this feedback is used to further refine the verifier through process-level supervision. This cyclical process ensures that both the verifier and autoformalizer continuously improve through mutual enhancement, guided by objective compiler feedback.
>
> We acknowledge that these details could be presented more clearly in the paper, and we will expand Section 5.1.2 to provide a more comprehensive description of the VEA training process.

---

> ### Author Response · Authors · 2024-11-22
> **Title: Response to Reviewer B3Z6 (5/5)**
>
> **R minor 7:**
>
> Thank you for the question of the running time comparison. Yes, the relative efficiency can be derived from the running time analysis **provided in lines 1427-1430 of our paper:** "Running Time: It's crucial to note that there is significant room for improvement in Lean 4's compilation times. The compilation duration varies depending on factors such as theorem complexity, dependencies on relevant lemmas or theorems, etc. Compiling 1k examples requires around 10 minutes. This duration is notably longer than the generation time for a large language model, which typically takes only 1-2 minutes to generate output on 1k samples."
>
> Since VEA employs a verifier model that, like other large language models, only needs to predict a single value for each input, it maintains the same efficient processing time of 1-2 minutes for 1k samples. In contrast, RFT requires actual compilation through Lean 4 for each sample, incurring the full 10-minute processing time for the same 1k samples. This results in VEA being approximately 5-10 times faster than RFT in practice. We will add this explicit time comparison to the paper's appendix to make this efficiency difference clearer.
>
> **R minor 8**
>
> As described in lines 495-508 of our paper, the columns directly correspond to our factorial design variables in human evaluation. Specifically, ``Proof Validity`` reflects whether the autoformalized sample's statement and proof can pass the Lean 4 compiler together. The "Model" column compares the baseline autoformalizer versus the RFT \+ VEA enhanced autoformalizer. ``Dataset Split`` represents the distribution across our three test sets: basic test, random test, and real test.
>
> Additionally, "Avg" and "Fleiss' K" represent the evaluation metrics: "Avg" shows the mean evaluation score across all human annotators, while "Fleiss' K" indicates the Fleiss' Kappa inter-rater reliability measure, which quantifies the agreement level among our annotators. We will revise the table description to explicitly include these column explanations, making the evaluation setup and metrics immediately clear to readers.

---

> ### Comment · Reviewer_B3Z6 · 2024-11-22
> **Follow up questions:**
>
> > “We propose a process-driven framework PDA that leverages formal languages to provide process feedback on reasoning, with an initial implementation focused on Lean 4, and we outline its potential adaptability to other formal languages.”
>
>
> Isn’t formal languages too broad a scope for this? Do you mean formal languages used for theorem proving or any other programming language?
>
> > This misalignment is not due to incorrectness but rather stems from the inherent challenges of translating formal mathematical proofs
>
> I’m still struggling a little to wrap my head around this. Shouldn’t the goal be to eventually only retain proofs that are consistent with the text? Even though it might be a hard manual task to figure out these inconsistencies, is the future goal of this work to ensure that the dataset has fewer instances of such misalignments when the dataset is made public through community feedback?
>
> > Quality Metrics Interpretation
>
> Thanks for the explanation. Also thanks for the Table from R W6, it immediately answered my follow-up question.

---

> > ### Comment · Reviewer_3y6J · 2024-11-22
> > **Source of misalignment**
> >
> > > FormL4 does not contain any incorrect proofs - all formal proofs in our dataset are valid and verified by the Lean 4 compiler.
> >
> > The statement here should be that FormL4 does not contain any incorrect *formal* proofs, right?  Or did someone go through 100% of the informal proofs in the dataset and ensure that none of the informal proofs were so egregiously misaligned as to be considered incorrect?
> >
> > > > This misalignment is not due to incorrectness but rather stems from the inherent challenges of translating formal mathematical proofs
> > >
> > > I’m still struggling a little to wrap my head around this. Shouldn’t the goal be to eventually only retain proofs that are consistent with the text? Even though it might be a hard manual task to figure out these inconsistencies, is the future goal of this work to ensure that the dataset has fewer instances of such misalignments when the dataset is made public through community feedback?
> >
> > I think the authors are claiming that most (all?) of the "misaligned" informal proofs were only rated as misaligned because feasibly-written informal proofs must gloss over details that may be present in the formal proof, or because the informal proof may reorder conditions for ease of reading (for example, do we say that `forall (n : ℕ), n > 0 -> P n` aligns with "for all positive integer n, P(n)", or do we say that it is misaligned because "for all positive integers" is not the same as "for all natural numbers, whenever the number is positive ..."?).  But unless I am missing something, the paper itself is lacking any information about the procedure used to validate that all misalignments were indeed due to the "inherent challenges of translating formal mathematical proofs" rather than more egregious errors such as eliding essential conditions, hallucinating invalid assumptions, improperly translating control-flow, etc.

---

> > > ### Comment · Reviewer_B3Z6 · 2024-11-23
> > >
> > > I see. Thanks for the clarification Reviewer 3y6J.
> > >
> > > I would encourage the authors to respond to the other questions and concerns from reviewer 3y6J. I would certainly raise my score if those concerns are reasonably addressed.

---

> > > > ### Author Response · Authors · 2024-11-29
> > > >
> > > > Dear Reviewer B3Z6,
> > > >
> > > > We would like to sincerely thank you again for your time in reviewing our work!
> > > >
> > > > We fully understand you might be quite busy. However, as the discussion deadline is approaching, would you mind checking our responses to your follow-up questions and concerns? If our reply and revisions resolve your concerns, we would appreciate your considering raising the overall score accordingly. Any further comments and discussions are welcomed!
> > > >
> > > > Best Regards,
> > > >
> > > > The authors of Submission2128

---

> > > > > ### Comment · Reviewer_B3Z6 · 2024-11-30
> > > > >
> > > > > Thanks for addressing my initial concerns.
> > > > >
> > > > > I feel unconfident if reviewer 3y6J’s concerns are sufficiently addressed, since the reviewer has not replied to the author’s  rebuttal, I keep my score as is based on the information that is currently available.
> > > > >
> > > > > The following comment from reviewer 3y6J is critical:
> > > > >
> > > > > > R W2.4 (Data Quality & Verification claims):
> > > > >  “extensive manual verification” is used to mean “≈70% accuracy looking at ≈ 3% of the data”. For a work involving formal verification to use “verification” to mean “at least ≈ 2% correct” is at least skirting the boundary of the code of ethics (specifically the “Uphold High Standards of Scientific Excellence” and "Be Honest, Trustworthy and Transparent” sections), if not outright violating it.
> > > > >
> > > > > On reading both the reviewer 3y6J’s comment and author’s response, I’m still unsure on exactly what should be considered a good number here. I would like to hear a response from reviewer 3y6J on this.
> > > > >
> > > > > Additionally,
> > > > >
> > > > > > Table 14 is damning: getting 0.65% on MATH and 8.16% on GSM8K even with a fine-tuned Mistral (Full) model suggests that that high results on FormL4 are a result of either overfitting or of FormL4 not spanning a large enough difficulty range. This should not be buried in Appendix N.
> > > > >
> > > > > Have authors addressed this comment?
> > > > >
> > > > > > R W4.1
> > > > > A proper baseline would involve comparison with an autoformalizer trained on randomized mislabeled data from FormL4 (pairs of informal-formal that are unrelated).
> > > > >
> > > > > I do not understand why author’s could not simply show an additional ablation in appendix with this baseline. How long would it take to perform this experiment?

---

> > ### Author Response · Authors · 2024-11-23
> > **Response to Reviewer B3Z6's follow-up questions and clarifying misrepresentations from Reviewer 3Y6J (1/2)**
> >
> > Dear Reviewer B3Z6,
> >
> > Thank you for your prompt response and questions\! We are encouraged that you find our explanation helpful in addressing your concerns. Below, we first address your follow-up questions, and then clarify potential misunderstandings, particularly those stemming from Reviewer 3Y6J's comments, which we believe misrepresent our method and findings.
> >
> > ---
> >
> > **Follow-up Q1:**
> >
> > > Isn’t formal languages too broad a scope for this? Do you mean formal languages used for theorem proving or any other programming language?
> >
> > **Response:**
> >
> >  We acknowledge that "formal languages" is a broad term. In the context of this work focusing on autoformalization, we specifically refer to formal languages employed in theorem proving, with a focus on Lean 4\. Following previous works (e.g., \[11\]), we follow the popular definition of “autoformalization” within the domain of formal mathematics (Line 26). Therefore, the adaptability mentioned in our proposal is intended to extend our approach to similar theorem-proving environments (e.g., Coq, Isabelle), not general-purpose programming languages. We appreciate this opportunity to clarify the scope.
> >
> > References:
> >
> > \[11\] Autoformalization with Large Language Models. Wu. et al., NeurIPS 2022\.
> >
> > **Follow-up Q2:**
> >
> > > I’m still struggling a little to wrap my head around this. Shouldn’t the goal be to eventually only retain proofs that are consistent with the text? Even though it might be a hard manual task to figure out these inconsistencies, is the future goal of this work to ensure that the dataset has fewer instances of such misalignments when the dataset is made public through community feedback?
> >
> > **Response:**
> >
> > > Shouldn’t the goal be to eventually only retain proofs that are consistent with the text?
> >
> > Following the objective of existing autoformalization research, t**he goal of the current work focuses only on statement autoformalization**, hence the alignment between natural language statements and formal language statements. The central reason for keeping the scope constrained is due to the inherent challenges, as you quoted. To explain the clarified objective, we would like to walk through three important notions in the paper:
> >
> > **First, “inherent challenges of translating proof”:** misaligned proofs are not due to incorrectness but arise from inherent challenges in translating formal proofs into natural language explanations. This is discussed in lines 1295–1299 of our submission:
> >
> > “Misalignment occurs when translating a set of formal proofs to natural language. This is because formal proofs are often expressed in pre-defined lemmas or environments that are exclusively constructed in Lean 4, and there are no existing corresponding natural language descriptions.”
> >
> > To better illustrate, consider the example below:
> >
> > * **Natural Language Statement:**
> >
> >  For any positive real numbers (a) and (b), their arithmetic mean is greater than or equal to their geometric mean. $ \\frac{a+b}{2} \\geq \\sqrt{ab} $ with equality if and only if $a \= b$.
> >
> > * **Natural Language Proof:**
> >
> >  1. Start with a known fact: For any real number (x), $x^2 \geq 0$.
> >  2. Apply this to: $\sqrt{a} - \sqrt{b})^2 \geq 0$.
> >  3. Expand: $a - 2\sqrt{ab} + b \geq 0$.
> >  4. Add $2\sqrt{ab}$ to both sides: $a + b \geq 2\sqrt{ab}$.
> >  5. Divide both sides by 2: $\frac{a + b}{2} \geq \sqrt{ab}$.
> >  Thus, the arithmetic mean is greater than or equal to the geometric mean.
> >
> > * **Lean 4 Proof:**
> >
> >   theorem am\_gm {a b : ℝ} (ha : 0 \< a) (hb : 0 \< b) : (a \+ b)/2 ≥ real.sqrt (a\*b) := by
> >    have h1 : (real.sqrt a \- real.sqrt b)^2 ≥ 0 := by nlinarith
> >    have h2 : a \- 2\*real.sqrt (a\*b) \+ b ≥ 0 := by
> >      rw \[pow\_two\] at h1
> >      exact h1
> >    linarith
> >
> > The key distinctions:
> >
> > * The Lean 4 proof compresses the reasoning through automated tactics (`nlinarith` and `linarith`), which handle computational steps like squaring and expanding terms automatically.
> > * In contrast, the natural language proof explicitly lays out every reasoning step for human understanding.
> >
> > **Second, “align vs incorrect”:** This discrepancy between a formal proof and its natural language counterpart exemplifies "misalignment." However, as we emphasize in our submission, **misalignment does not imply incorrectness.** Rather, it reflects the differing expressive capabilities and conventions of formal and informal reasoning.
> >
> > While future iterations of the dataset might aim to reduce such gaps, perfect alignment may not always be achievable, as the underlying goals of formal and natural language proofs differ.

---

> ### Author Response · Authors · 2024-11-23
> **Response to Reviewer B3Z6's follow-up questions and clarifying misrepresentations from Reviewer 3Y6J (2/2)**
>
> (continued)
>
> **Third, “the role of proof”:** given that FormL4 is a dataset for statement autoformalization, the quality of FormL4 and our evaluation framework is independent of whether the natural-language proof is perfectly aligned with the formal proof. This is now explicitly stated in L218-220 of our revised submission.
>
> **We fully understand that it may lead to readers’ confusion as to why proof is even included in the dataset if the autoformalization task only focuses on statements.** Therefore, we make substantial efforts in the neighboring paragraphs (L202-L220) and early in Introduction (L73-96) explaining our motivation behind this seemingly counter-intuitive innovation. The beneficial roles of proof serve as an auxiliary aid mainly (1) to help LLMs generate informalized statements to improve statement quality of FormL4 and (2) to enable a more fine-grained process-level evaluation of the autoformalized statements. Please check the relevant text for a detailed mechanism.
>
> We sincerely appreciate your questions and engagement. We hope our response can address your concerns \!

---

> ### Author Response · Authors · 2024-11-23
> **Response to Reviewer 3Y6J's comment (titled 'Source of misalignment')**
>
> Dear Reviewer 3Y6J,
>
> Thank you for your comment. There are several misinterpretations in your reply regarding our paper submission.
>
> >> FormL4 does not contain any incorrect proofs \- all formal proofs in our dataset are valid and verified by the Lean 4 compiler. (from RW 5\)
> >
> > The statement here should be that FormL4 does not contain any incorrect *formal* proofs, right?
>
> Yes, as already suggested in the context of the same sentence. The second part of the sentence (`all formal proofs...`) after the short dash ` -` is a direct explanation of the previous statement.
>
> > The statement here should be that FormL4 does not contain any incorrect *formal* proofs, right?  Or did someone go through 100% of the informal proofs in the dataset and ensure that none of the informal proofs were so egregiously misaligned as to be considered incorrect?
> >
> >
> > I think the authors are claiming that most (all?) of the "misaligned" informal proofs were only rated as misaligned because feasibly written informal proofs must gloss over details that may be present in the formal proof......
>
> This misrepresents our work.
>
> In response to your mixed usage of “**incorrect proofs**” and “**misaligned**” together in the same flow of questions, we would like to clarify again that **whether formal statements are aligned with natural language statements (i.e., autoformalization success) and “incorrect” statements or proof are two separate concepts.** Misalignment pertains solely to the challenges of mapping these proofs to informal descriptions, not their correctness. The distinction is explained in detail in RW 5 and paper (e.g., section 3.2, L1239-1244). A way to help comprehend the distinction is that the former belongs to a translation problem comparing two items, while the latter focuses on the validity of a single item itself. For example, a formal statement or proof can be **correct** in stating or proving a theorem, but it does not describe the same thing as the original informal statement or proof (i.e., **misalignment**).
>
> Therefore, we can divide your question into two parts:
>
> 1. Firstly, we reiterate: all formal proofs in FormL4 are valid as verified by Lean 4\. Therefore we ensure all formal proofs are correct.
> 2. Secondly, We did not ensure all formal proofs are perfectly aligned with informal ones because (1) it is impractical to manually check all of them considering the large data size and (2) more importantly, it is **unnecessary** to do so because the roles of proof serve to aid statement quality and statement evaluation, not to be a subject for evaluation (L202-L220, L73-96).
>
> Note that the discussions above revolve around proofs. We can discuss statements as well: in contrast, we acknowledge that it is important to manually check the natural language and formal statements are aligned in FormL4, and made substantial efforts to do so (see RW 6).
>
> We hope this clarifies the scope of formal languages, the nature of misalignments, and our dataset's validation process. Misalignments are a well-recognized challenge inherent to the interplay between formal and natural language reasoning, and addressing them is a nuanced, ongoing effort. We appreciate your discussions. We will also respond to your own review soon.

---

> > ### Comment · Reviewer_3y6J · 2024-11-23
> > **(in)correctness of informal proof**
> >
> > > whether formal statements are aligned with natural language statements (i.e., autoformalization success) and “incorrect” statements or proof are two separate concepts
> >
> > I believe what's going on here is that you are talking about two separate concepts, while I am talking about three separate concepts:
> > 1. Correctness of formal proof (whether or not Lean accepts the proof)
> > 2. Correctness of informal proof (if your informal proof of infinitude of primes is "every prime other than 2 is odd; every integer is either even or odd; since there are infinitely many integers, we must have either infinitely many evens or infinitely many odds; there must be infinitely many odds, because every even number is followed by an odd number; therefore there are infinitely many primes", this proof is incorrect, because it attempts to use "all primes > 2 are odd" where it would instead need "all odds are prime", which is false)
> > 3. Alignment of formal and informal proof (for example, we may prove $\forall x,y\in \mathbb{N}.x+y=y+x$ either by starting with induction on $x$ or by starting with induction on $y$; if the formal proof begins with induction on $x$ while the informal proof begins with induction on $y$, we would consider them misaligned)
> >
> > I agree that both (2) and (3) are somewhat subjective, and that there are ambiguous cases for both of them.

---

> ### Author Response · Authors · 2024-11-25
> **Author Reply to Reviewer 3y6J (titled '(in)correctness of informal proof')**
>
> We thank Reviewer 3y6J for their reply. We are pleased that we have now reached a consensus on the distinction between "align" and "incorrect."
>
> Indeed, it is valid to divide the general concept of proof correctness into the correctness of "informal proof" and "formal proof." We sincerely appreciate your engagement and insights.

---

> ### Author Response · Authors · 2024-12-04
>
> Thank you for your responses. We are encouraged that your initial concerns have been solved.
>
> - On data quality:
>
> In the paper, we demonstrate data quality with two rounds of human verification: firstly 60 samples of FormL4, then a comparative 30*3 verification of three datasets of similar size and construction methods. It is supported that FormL4 has the highest sampling rate for verification among the comparable large-size datasets with the highest accuracy. This is also discussed in detail in [R W6: Quality Metrics Interpretation], in response to your previous concerns.
>
> Regarding reviewer 3y6J's comment you quoted, we greatly appreciate your efforts in engagements with them. Firstly, it is unknown to us where reviewer 3y6J's “at least ≈ 2% correct” is stated or indicated in our paper. Secondly, we believe this concern from reviewer 3y6J's is already addressed now, based on their own follow-up response that the objection is not towards the dataset quality but the 'presentation' of dataset quality. We further firmly responded to this view by pointing out the details that reviewer 3y6J missed in our paper.
>
> - On Table 14:
>
> The discrepancy exists because GSM8K and MATH are out-of-domain benchmarks consisting of natural language math questions, similar to the real test set in FormL4 (which is constructed exactly because we want to comprehensively examine out-of-domain performances as well). In contrast, the training set of FormL4 is collected from Mathlib4 which contains advanced formal mathematical theorems, widely different from GSM8K or MATH. Therefore, Table 14 is not an indication of overfitting but an investigation of the model's limitation in transferring to out-of-domain benchmarks. Please check `"Real Test is Still Challenging"` (L1546-L1559) near Table 14 where we actually attribute the discrepancy factors in detail.
>
> - On appendix baselines:
>
> Thank you for your suggestion.  While we tried to train such an ablation baseline after receiving your suggestion (2 days before the end of discussion period), we are sorry there is not enough time for us to provide it. However, in response to your concerns, we did not include such baselines in the first place based on the following reasons provided in R W4.1:
>
> Our evaluation framework in Table 4 includes comprehensive benchmarking against both open and closed-source models. Firstly, the positively fine-tuned baseline in the current Table 4 is more challenging and thus sufficient. Secondly, while we appreciate the suggestion of randomized pairings as a control, such comparisons would face several methodological challenges:
>
> 1. Random pairing would violate the fundamental semantic relationship between informal and formal mathematics that autoformalization aims to capture;
> 2. Training on deliberately misaligned data would not provide meaningful insights into model capabilities for the intended task;
> 3. The suggested approach could introduce confounding factors that make results harder to interpret.
>
> We greatly appreciate your review and engagements. Please feel free to check our redirections to the paper, which may answer your questions efficiently and in detail. We would appreciate you considering raising the score if most of your concerns are addressed.
>
> Thank you again for your efforts in your review.

---

### Meta-Review · Area_Chair_hZrp · 2024-12-24

**Metareview:**

The paper takes on a unique and novel angle for advancing reasoning abilities of LLMs, by presenting a new dataset FormL4 for evaluating autoformalization in Lean4, a language with rather limited presence in training corpora. In order to improve autoformalization ability, the authors also introduce Process Driven Autoformalization (PDA), which uses feedback from Lean 4's interpreter.

We'd like to commend on the author's tremendous effort and work during the rebuttal phase. All reviewers, while holding rather opposite sides of advocacy of this paper, agreed that the author did a remarkable job at rebuttal. However, given the slight overclaim of contribution and lack of clarity of the method, we encourage the author to strengthen the paper, and hope to see this work in a future submission.

**Additional Comments On Reviewer Discussion:**

The author had clearly dealt with a difficult rebuttal process. This has been one of the most difficult paper decisions to make, but after a long discussion and consideration, we voted on rejection for now.

---

### Decision · Program_Chairs · 2025-01-22

Reject